# SAMPLE-EFFICIENT LEARNING OF INFINITE-HORIZON AVERAGE-REWARD MDPS WITH GENERAL FUNCTION APPROXIMATION

**Jianliang He**
Department of Statistics and Data Science
Fudan University
hejl@fudan.edu.cn

**Han Zhong**
Center for Data Science
Peking University
hanzhong@stu.pku.edu.cn

**Zhuoran Yang**
Department of Statistics and Data Science
Yale University
zhuoran.yang@yale.edu

## ABSTRACT

We study infinite-horizon average-reward Markov decision processes (AMDPs) in the context of general function approximation. Specifically, we propose a novel algorithmic framework named Local-fitted Optimization with OPtimism (LOOP), which incorporates both model-based and value-based incarnations. In particular, LOOP features a novel construction of confidence sets and a low-switching policy updating scheme, which are tailored to the average-reward and function approximation setting. Moreover, for AMDPs, we propose a novel complexity measure — average-reward generalized eluder coefficient (AGEC) — which captures the challenge of exploration in AMDPs with general function approximation. Such a complexity measure encompasses almost all previously known tractable AMDP models, such as linear AMDPs and linear mixture AMDPs, and also includes newly identified cases such as kernel AMDPs and AMDPs with Bellman eluder dimensions. Using AGEC, we prove that LOOP achieves a sublinear $\tilde{\mathcal{O}}(\mathrm{poly}(d, \mathrm{sp}(V^*))\sqrt{T\beta})$ regret, where $d$ and $\beta$ correspond to AGEC and log-covering number of the hypothesis class respectively, $\mathrm{sp}(V^*)$ is the span of the optimal state bias function, $T$ denotes the number of steps, and $\tilde{\mathcal{O}}(\cdot)$ omits logarithmic factors. When specialized to concrete AMDP models, our regret bounds are comparable to those established by the existing algorithms designed specifically for these special cases. To the best of our knowledge, this paper presents the first comprehensive theoretical framework capable of handling nearly all AMDPs.

## 1 INTRODUCTION

Reinforcement learning (RL) (Sutton & Barto, 2018) is a powerful tool for addressing intricate sequential decision-making problems. In this context, Markov decision processes (MDPs) (Puterman, 2014; Sutton & Barto, 2018) frequently serve as a fundamental model for modeling such decision-making scenarios. Motivated by different feedback structures in real applications, MDPs consist of three subclasses — finite-horizon MDPs, infinite-horizon discounted MDPs, and infinite-horizon average-reward MDPs. Each of these MDP variants is of paramount significance and operates in a parallel fashion, with none being amenable to complete reduction into another. Of these three MDP subclasses, finite-horizon MDPs have received significant research efforts in understanding their exploration challenge, especially in the presence of large state spaces which necessitates function approximation tools. Existing works on finite-horizon MDPs have proposed numerous structural conditions on the MDP model that empower sample-efficient learning. These structural conditions include but are not limited to linear function approximation (Jin et al., 2020), Bellman rank (Jiang et al., 2017), eluder dimension (Wang et al., 2020), Bellman eluder dimension (Jin et al., 2021), bilinear class (Du et al., 2021), decision estimation coefficient (Foster et al., 2021), and generalized

eluder coefficient (Zhong et al., 2022). Moreover, these works have designed various model-based and value-based algorithms to address finite-horizon MDPs governed by these structural conditions.

In contrast to the rich literature devoted to finite-horizon MDPs, the study of sample-efficient exploration in infinite-horizon MDPs has hitherto been relatively limited. Importantly, it remains elusive how to design in a principled fashion a sample-efficient RL algorithm in the online setting with general function approximation. To this end, we focus on infinite-horizon average-reward MDPs (AMDPs), which offer a suitable framework for addressing real-world decision-making scenarios that prioritize long-term returns, such as product delivery (Proper & Tadepalli, 2006). Our work endeavors to provide a unified theoretical foundation for understanding infinite-horizon average-reward MDPs from the perspective of general function approximation, akin to the comprehensive investigations conducted in the domain of finite-horizon MDPs. To pursue this overarching objective, we have delineated two subsidiary goals that form the crux of our research endeavor.

- **Development of a Novel Structural Condition/Complexity Measure.** Existing works are restricted to tabular AMDPs (Bartlett & Tewari, 2012; Jaksch et al., 2010) and AMDPs with linear function approximation (Wu et al., 2022; Wei et al., 2021), with Chen et al. (2022a) as the only exception (to our best knowledge). While Chen et al. (2022b) does extend the eluder dimension for finite-horizon MDPs (Ayoub et al., 2020) to the infinite-horizon average-reward context, their complexity measure seems to be only slightly more general than the linear mixture AMDPs (Wu et al., 2022) and falls short in capturing other fundamental models such as linear AMDPs (Wei et al., 2021). Hence, our first subgoal is proposing a new structural condition. This condition is envisioned to be sufficiently versatile to encompass all known tractable AMDPs, while also potentially introducing innovative and tractable models into the framework.
- **Algorithmic Framework for Addressing Identified Structural Condition.** The second subgoal is anchored in the development of sample-efficient algorithms for AMDPs characterized by the structural condition proposed in our work. Our aspiration is to devise an algorithmic framework that can be flexibly implemented in both model-based and value-based paradigms, depending on the nature of the problem at hand. This adaptability guarantees that our algorithms possess the ability to effectively address a wide range of AMDPs.

Our work attains these two pivotal subgoals through the introduction of (i) a novel complexity measure — Average-reward Generalized Eluder Coefficient (AGEC), and (ii) a corresponding algorithmic framework dubbed as Local-fitted Optimization with OPtimism (LOOP). Our primary contributions and novelties are summarized below:

- **AGEC Complexity Measure.** Our complexity measure AGEC extends the generalized eluder coefficient (GEC) in Zhong et al. (2022) to the infinite-horizon average-reward setting. However, it incorporates significant modifications. AGEC not only establishes a connection between the Bellman error and the training error, akin to GEC but also imposes certain constraints on transferability (see Definition 3 for details). This modification proves instrumental in attaining sample efficiency in the realm of AMDPs (see Section 5 for detailed discussion). We demonstrate that AGEC not only encompasses all previously recognized tractable AMDPs, including tabular AMDPs (Bartlett & Tewari, 2012; Jaksch et al., 2010), linear AMDPs (Wei et al., 2021), linear mixture MDPs (Wu et al., 2022), AMDPs with low eluder dimension (Chen et al., 2022a), but also captures some new identified models like linear $Q^*/V^*$ AMDPs (see Definition 14), kernel AMDPs (see Proposition 7), and AMDPs with low Bellman eluder dimension (see Definition 8).
- **LOOP Algorithmic Framework.** Our algorithm LOOP is based on the optimism principle and features a novel construction of confidence sets along with a low-switching updating scheme. Remarkably, LOOP offers the flexibility to be implemented either in the model-based or value-based paradigm, depending on the problem type.
- **Unified Theoretical Results.** From the theoretical side, we prove that LOOP enjoys the regret of $\tilde{\mathcal{O}}(\text{poly}(d, \text{sp}(V^*))\sqrt{T\beta})$, where $d$ and $\beta$ correspond to the AGEC and the log-covering number of the hypothesis class respectively, $\text{sp}(V^*)$ denotes the span of the optimal state bias function, $T$ is the number of steps, and $\tilde{\mathcal{O}}$ hides logarithmic factors. This result shows that LOOP is capable of solving all AMDPs with low AGEC.

In summary, we provide a unified theoretical understanding of infinite-horizon AMDPs with general function approximation. **Further elaboration on our contributions and technical novelties are**

| Algorithm | Assumption | Type | Tabular | Linear Mixture | Linear | Eluder | ABE | Kernel | AGEC |
|-----------|-----------|------|---------|----------------|--------|--------|-----|--------|------|
| **LOOP (Ours)** | Bellman optimality (finite span) | Model-based & Value-based | ✓ | ✓ | ✓ | ✓ | ✓ | ✓ | ✓ |
| **SIM-TO-REAL** (Chen et al., 2022a) | Communicating AMDP (finite diameter) | Model-based | ✓ | ✓ | ✗ | ✓ | ✗ | ✗ | ✗ |
| **UCRL2-VTR** (Wu et al., 2022) | Communicating AMDP (finite diameter) | Model-based | ✓ | ✓ | ✗ | ✗ | ✗ | ✗ | ✗ |
| **FOPO** (Wei et al., 2021) | Bellman optimality (finite span) | Value-based | ✓ | ✗ | ✓ | ✗ | ✗ | ✗ | ✗ |
| **UCRL2** (Jaksch et al., 2010) | Communicating AMDP (finite diameter) | Model-based | ✓ | ✗ | ✗ | ✗ | ✗ | ✗ | ✗ |

Table 1: A comparison with the most related algorithms on AMDPs. We remark that our assumption is weaker since the communicating MDP satisfies the Bellman optimality and the diameter is bound by the span. Besides, average-reward Bellman eluder dimension (ABE), kernel AMDPs, and AGEC are new complexity measures proposed by our work. In particular, AGEC serves as a unifying complexity measure capable of encompassing all other established complexity measures.

**provided in Appendix A.** Due to space limits, we only provide a comparison between our results and mostly related works on AMDPs in Table 1. More related works are deferred to Appendix A.

## 2 PRELIMINARIES

An infinite-horizon average-reward Markov Dependent Process (AMDPs) is characterized by $\mathcal{M} = (\mathcal{S}, \mathcal{A}, r, \mathbb{P})$, where $\mathcal{S}$ is a Borel state space with a possibly infinite number of elements, $\mathcal{A}$ is a finite set of actions, $r : \mathcal{S} \times \mathcal{A} \mapsto [-1, 1]$ is an unknown reward function[1] and $\mathbb{P}(\cdot|s, a)$ is the unknown transition kernel. The learning protocol for infinite-horizon average-reward RL is as follows: the agent interacts with $\mathcal{M}$ over a fixed number of $T$ steps, starting from a pre-determined initial state $s_1 \in \mathcal{S}$. At each step $t \in [T]$, the agent observe a state $s_t \in \mathcal{S}$ and takes an action $a_t \in \mathcal{A}$, receives a reward $r(s_t, a_t)$ and transits to the next state $s_{t+1}$ drawn from $\mathbb{P}(\cdot|s_t, a_t)$.

Denote $\Delta(\mathcal{A})$ be the probability simplex over the action space $\mathcal{A}$. Specifically, the stationary policy $\pi$ is a mapping $\pi : \mathcal{S} \mapsto \Delta(\mathcal{A})$ with $\pi(a|s)$ specifying the probability of taking action $a$ at state $s$. Given a stationary policy $\pi$, the long-term average reward starting is defined as

$$J^\pi(s) := \liminf_{T \mapsto \infty} \frac{1}{T} \mathbb{E}\left[\sum_{t=1}^{T} r(s_t, a_t)|s_1 = s\right], \qquad \forall s \in \mathcal{S},$$

where the expectation is taken with respect to the policy, i.e., $a_t \sim \pi(\cdot|s_t)$ and the transition, i.e., $s_{t+1} \sim \mathbb{P}(\cdot|s_t, a_t)$. In infinite-horizon average-reward RL, existing works mostly rely on additional assumptions to achieve sample efficiency. The necessity arises from the absence of a natural counterpart to the celebrated Bellman optimality equation in the average-reward RL that is self-evident and crucial within episodic and discounted settings (Puterman, 2014). To this end, we consider a broad subclass where a modified Bellman optimality equation holds (Hernández-Lerma, 2012).

**Assumption 1** (Bellman optimality equation). There exists bounded measurable function $Q^* : \mathcal{S} \times \mathcal{A} \mapsto \mathbb{R}$, $V^* : \mathcal{S} \mapsto \mathbb{R}$ and unique constant $J^* \in [-1, 1]$ such that for all $(s, a) \in \mathcal{S} \times \mathcal{A}$, it holds

$$J^* + Q^*(s, a) = r(s, a) + \mathbb{E}_{s' \sim \mathbb{P}(\cdot|s,a)}[V^*(s')], \quad V^*(s) = \max_{a \in \mathcal{A}} Q^*(s, a). \qquad (2.1)$$

The Bellman optimality equation, adapted for average-reward RL, posits that for any initial states $s_1 \in \mathcal{S}$, the optimal reward is independent such that $J^*(s_1) = J^*$ under a deterministic optimal policy following $\pi^*(\cdot|s) = \operatorname{argmax}_{a \in \mathcal{A}} Q^*(\cdot, a)$. The justification is presented in Wei et al. (2021).

Note that functions $V^*(s)$ and $Q^*(s, a)$ reveal the relative advantage of starting from state $s$ and state-action pair $(s, a)$ under the optimal policy, and are respectively called the optimal state and

---

[1]Throughout this paper, we consider the deterministic reward for notational simplicity and all results are readily generalized to the stochastic setting. Also, we assume reward lies in $[-1, 1]$ without loss of generality.

state-action bias function (Wei et al., 2021). Denote $\mathrm{sp}(V) = \sup_{s,s' \in \mathcal{S}} |V(s) - V(s')|$ as the span of any bounded measurable function. Note that for any solution pair $(V^*, Q^*)$ satisfying the Bellman optimality equation in (2.1), the shifted pair $(V^* - c, Q^* - c)$ for any constant $c$ is still a solution. Thus, without loss of generality, we can focus on the unique centralized solution such that $\|V^*\|_\infty \leq \frac{1}{2}\mathrm{sp}(V^*)$. Following the tradition in the average-reward RL (Wei et al., 2020; 2021; Wang et al., 2022; Zhang & Xie, 2023), the span $\mathrm{sp}(V^*)$ is assumed to be known.

As aforementioned in the paper, distinct assumptions have been employed in average-reward RL research to ensure the explorability of the problem, which includes ergodic AMDPs (Wei et al., 2020; Hao et al., 2021; Zhang & Xie, 2023), communicating AMDPs (Chen et al., 2022a; Wang et al., 2022; Wu et al., 2022) and the Bellman optimality equation (Wei et al., 2021). Among these widely adopted assumptions, we remark that the Bellman optimality equation is the least stringent one. Note that the ergodic MDP suggests the existence of bias functions for each $\pi \in \Pi$, while the latter two only require the existence of bias functions for the optimal policy. As for weak communicating assumption, a weaker form of communicating MDP (Wang et al., 2022), it directly implies the existence of the Bellman optimality equation and thus is stronger (Hernández-Lerma, 2012). Given the Bellman optimality assumption in (2.1), we introduce the average-reward Bellman operator below:

$$(\mathcal{T}_J F)(s,a) := r(s,a) + \mathbb{E}_{s' \sim \mathbb{P}(\cdot|s,a)}\left[\max_{a' \in \mathcal{A}} F(s',a')\right] - J, \qquad \forall (s,a) \in \mathcal{S} \times \mathcal{A}, \qquad (2.2)$$

for any bounded function $F : \mathcal{S} \times \mathcal{A} \mapsto \mathbb{R}$ and constant $J \in [-1, 1]$. Then, the Bellman optimality equation in (2.1) can be written as $\mathcal{T}_{J^*} Q^* = Q^*$. Moreover, we define the Bellman error:

$$\mathcal{E}(F, J)(s,a) := F(s,a) - (\mathcal{T}_J F)(s,a), \qquad \forall (s,a) \in \mathcal{S} \times \mathcal{A}. \qquad (2.3)$$

**Learning Objective** Under the framework of online learning for AMDPs, the agent aims to learn the optimal policy by interacting with the environment over potentially infinite steps. The regret measures the cumulative difference between the optimal average-reward and the reward achieved after interacting for $T$ steps, formally defined as $\mathrm{Reg}(T) = \sum_{t=1}^{T} (J^* - r(s_t, a_t))$.

## 3 GENERAL FUNCTION APPROXIMATION

To capture both model-free and model-based problems with function approximation, we consider a general hypotheses class $\mathcal{H}$ which contains a class of functions. We consider two kinds of hypothesis classes, targeting at *value-based* problems and *model-based* problems respectively.

**Definition 1** (Value-based hypothesis)**.** We say $\mathcal{H}$ is a value-based hypotheses class if all hypothesis $f \in \mathcal{H}$ is defined over state-action bias function $Q$ and average-reward $J$ such that $f = (Q_f, J_f) \in \mathcal{H}$. Let $V_f(\cdot) = \max_{a \in \mathcal{A}} Q_f(\cdot, a)$ and $\pi_f(\cdot) = \operatorname{argmax}_{a \in \mathcal{A}} Q_f(\cdot, a)$ be the greedy bias function and policy induced from hypothesis $f \in \mathcal{H}$. Denote $f^*$ be the optimal hypothesis under true model $\mathcal{M}$.

**Definition 2** (Model-based hypothesis)**.** We say $\mathcal{H}$ is a model-based hypotheses class if all hypothesis $f \in \mathcal{H}$ is defined over the transition kernel $\mathbb{P}$ and reward function $r$ such that $f = (\mathbb{P}_f, r_f) \in \mathcal{H}$. Let $Q_f$, $V_f$, $J_f$, and $\pi_f$ respectively be the optimal bias functions, average-reward and policy induced from hypothesis $f \in \mathcal{H}$, which satisfies the Bellman optimality equation such that

$$Q_f(s,a) + J_f = (r_f + \mathbb{P}_f V_f)(s,a), \quad \mathbb{P}_{f'} V_f(s,a) := \mathbb{E}_{s' \sim \mathbb{P}_{f'}(\cdot|s,a)}[V_f(s')],$$

for all $s \in \mathcal{S}$ and $a \in \mathcal{A}$. Denote $f^*$ as the hypothesis concerning the true model $\mathcal{M}$.

The definition of hypotheses class $\mathcal{H}$ over the value-based (see Definition 1) and the model-based (see Definition 2) problems in AMDP is different from the episodic setting (Du et al., 2021; Zhong et al., 2022). The most significant difference is that the Bellman equation has a different form. As a result, in the value-based scenario, instead of using a single state-action value function $Q_f$ in episodic setting, the paired hypothesis $(Q_f, J_f)$ is introduced to fully capture the average-reward structure. Besides, we retain the definition of hypothesis over model-based problems, augmenting it with an additional average-reward term $J_f$ induced from $(\mathbb{P}_f, r_f)$. Since we do not impose any specific structural form to the hypothesis class, we stay in the realm of *general function approximation*.

As function approximation is challenging without further assumptions (Krishnamurthy et al., 2016), we introduce the realizability assumption, which is widely adopted in literatures (Jin et al., 2021).

**Assumption 2** (Realizablity). We assume that $f^* \in \mathcal{H}$.

Moreover, we establish the fundamental distribution families over the state-action pair upon which the metric is built. Considering the learning goal defined over the empirical regret, throughout the paper we focus on the point-wise distribution family $\mathcal{D}_\Delta = \{\delta_{s,a}(\cdot)|(s,a) \in \mathcal{S} \times \mathcal{A}\}$, which includes collections of Dirac probability measure over $\mathcal{S} \times \mathcal{A}$. Discussions are deferred to Appendix A.

### 3.1 AVERAGE-REWARD GENERALIZED ELUDER COEFFICIENTS

In this subsection, we are going to introduce a novel metric — average-reward generalized eluder coefficients (AGEC), to capture the complexity of hypothesis class $\mathcal{H}$ for AMDP. Extended from the generalized Eluder coefficients (GEC; Zhong et al., 2022) for finite-horizon MDPs, AGEC is a variant to fit the infinite-horizon learning with average reward, and imposes an additional structural constraint—transferability, motivated by Eluder condition (EC; Xiong et al., 2023) and proved mild in Section 3.2, to ensure the tractability of the problems.

**Definition 3** (AGEC). Given hypothesis class $\mathcal{H}$, discrepancy function set $\{l_f\}_{f \in \mathcal{H}}$ and constant $\epsilon > 0$, the average-reward generalized eluder coefficients $\mathrm{AGEC}(\mathcal{H}, \{l_f\}, \epsilon)$ is defined as the smallest coefficients $\kappa_G$ and $d_G$, such that following two conditions hold with absolute constants $C_1, C_2 > 0$:

(i) (Bellman dominance) There exists constant $d_G > 0$ such that

$$\underbrace{\sum_{t=1}^T \mathcal{E}(f_t)(s_t, a_t)}_{\text{Bellman error}} \leq \left[ d_G \cdot \underbrace{\sum_{t=1}^T \sum_{i=1}^{t-1} \|\mathbb{E}_{\zeta_i}[l_{f_i}(f_t, f_t, \zeta_i)]\|_2^2}_{\text{In-sample training error}} \right]^{1/2} + \underbrace{C_1 \cdot \mathrm{sp}(V^*) \min\{d_G, T\} + T\epsilon}_{\text{Burn-in cost}}.$$

(ii) (Transferability) There exists constant $\kappa_G > 0$ such that for hypotheses $f_1, \dots, f_T \in \mathcal{H}$, if $\sum_{i=1}^{t-1} \|\mathbb{E}_{\zeta_i}[l_{f_i}(f_t, f_t, \zeta_i)]\|_2^2 \leq \beta$ holds for all $t \in [T]$, then we have

$$\underbrace{\sum_{t=1}^T \|\mathbb{E}_{\zeta_t}[l_{f_t}(f_t, f_t, \zeta_t)]\|_2^2}_{\text{Out-sample training error}} \leq \kappa_G \cdot \beta \log T + \underbrace{C_2 \cdot \mathrm{sp}(V^*)^2 \min\{\kappa_G, T\} + 2T\epsilon^2}_{\text{Burn-in cost}}.$$

In the definition above, $\zeta_i$ is a subset of trajectory with varying meaning concerning the specific choice of discrepancy function, and the expectation is taken over it; $C_1, C_2$ are absolute constants related to span $\mathrm{sp}(V^*)$. To simplify the notation, we denote $\mathcal{E}(f_t)(s, a) := \mathcal{E}(Q_{f_t}, J_{f_t})(s, a)$ for all $(s, a) \in \mathcal{S} \times \mathcal{A}$. Besides, the Burn-in cost is taken at the worst case and it varies across different settings but usually non-dominating. The intuition behind the metric is that, on average, if hypotheses have small in-sample training error on the well-explored dataset, then the prediction error on a different trajectory is expected to maintain a consistently low level (Zhong et al., 2022). In specific, the dominance coefficient $d_G$ encapsulates the challenge inherent in assessing the performance of prediction, specifically the Bellman error, given the consistently controlled in-sample training error within the designated function class $\mathcal{H}$. Moreover, due to the unique challenge of infinite-horizon average-reward setting, we introduce the transferability coefficient $\kappa_G$ to quantify the transferability from the in-sample training error to the out-of-sample ones. Despite this additional structural condition, we can verify that nearly all tractable AMDPs admit a low AGEC value (see Section 3.2). Moreover, in Section 5, we will demonstrate the importance of such additional structural conditions for achieving sample efficiency in AMDPs from the theoretical perspective.

Moreover, to facilitate further theoretical analysis, we make further assumptions on the discrepancy function and hypothesis class as Chen et al. (2022b); Zhong et al. (2022).

**Assumption 3** (Boundedness). Given any $f \in \mathcal{H}$, it holds that $\|l_f\|_\infty \leq C_\ell \cdot \mathrm{sp}(V^*)$ with $C_\ell > 0$.

The boundedness assumption is reasonable and uniformly satisfied, as in most cases, it takes the Bellman discrepancy, defined as: for all $\zeta_t = \{s_t, a_t, r_t, s_{t+1}\} \in \mathcal{S} \times \mathcal{A} \times \mathbb{R} \times \mathcal{S}$, we have

$$l_{f'}(f, g, \zeta_t) = Q_g(s_t, a_t) - r(s_t, a_t) - V_f(s_{t+1}) + J_g, \tag{3.1}$$

or other natural derivatives, so that the discrepancy is generally upper bounded by $\mathcal{O}(\mathrm{sp}(V^*))$.

**Assumption 4** (Generalized completeness). Let $\mathcal{G}$ be an auxiliary function class and there exists a functional operator $\mathcal{P} : \mathcal{H} \mapsto \mathcal{G}$, we say that $\mathcal{H}$ satisfies generalized completeness in $\mathcal{G}$ concerning discrepancy function $l_{f'}$ if for any $(f, g) \in \mathcal{H} \times (\mathcal{H} \cup \mathcal{G})$, it holds that

$$l_{f'}(f, g, \zeta) - l_{f'}(f, \mathcal{P}(f), \zeta) = \mathbb{E}_{\zeta}\big[l_{f'}(f, g, \zeta)\big], \tag{3.2}$$

where the expectation is taken over trajectory $\zeta$. Besides, the operator satisfies that $\mathcal{P}(f^*) = f^*$.

The completeness assumption is an extension of the Bellman completeness for value-based hypothesis (Jin et al., 2021), incorporating the notion of the decomposition loss function (DLF) property proposed in Chen et al. (2022b). Our assumption diverges from the one posited in Zhong et al. (2022), where an auxiliary function class $\mathcal{P}(\mathcal{H}) \subseteq \mathcal{G}$ is introduced to enrich choices, accompanied with modifications tailored to accommodate the nuances of the average-reward setting.

**Example 1** (Bellman completeness $\subseteq$ Generalized completeness). Let the discrepancy function be the Bellman discrepancy in (3.1) with $\zeta_t = \{s_t, a_t, r_t, s_{t+1}\}$, and takes the (hypothesis-scheme) Bellman operator, defined as $\mathcal{T}(f) = \{\mathcal{T}_{J_f}(Q_f), J_f\}$ for all $f \in \mathcal{H}$, modified from (2.2). Then,

$$\begin{aligned} l_{f'}(f, g, \zeta_t) - l_{f'}\left(f, \mathcal{T}(f), \zeta_t\right) &= Q_g(s_t, a_t) - \mathcal{T}_{J_f}Q_f(s_t, a_t) \\ &= Q_g(s_t, a_t) - r(s_t, a_t) + \mathbb{E}_{\zeta_t}\big[V_f(s_{t+1})\big] + J_f = \mathbb{E}_{\zeta_t}\big[l_{f'}(f, g, \zeta_t)\big], \end{aligned}$$

where the expectation is taken over the transition state $s_{t+1}$ from $\mathbb{P}(\cdot|s_t, a_t)$.

The preceding example illustrates that the Bellman discrepancy, a frequently employed discrepancy function across problems, satisfies both assumptions. More examples and choices of the discrepancy function for MLE-based algorithms are respectively provided in Appendix B and C.

## 3.2 Relation with Tractable Complexity Metric

To bridge the gap between concrete function approximation instances and the relatively abstract measure AGEC, this section introduces two intermediate metrics: the Eluder dimension (Russo & Van Roy, 2013) and the Average-reward Bellman Eluder (ABE) dimension. In particular, Chen et al. (2022a) employs the Eluder dimension to gauge the complexity of model-based hypothesis classes for infinite-horizon learning. To provide an intuitive complexity of value-based hypothesis classes, we additionally propose the ABE dimension, which is a generalization of the standard BE dimension (Jin et al., 2021). These two metrics provide valuable insights into the nature of AGEC.

**Eluder Dimension**   We start with point-wise $\epsilon$-independence notation (Russo & Van Roy, 2013).

**Definition 4** (Point-wise $\epsilon$-independence). Let $\mathcal{H}$ be a function class defined on $\mathcal{X}$ and consider sequence $\{z, x_1, \ldots, x_n\} \in \mathcal{X}$. We say $z$ is $\epsilon$-independent of $\{x_1, \ldots, x_n\}$ with respect to $\mathcal{H}$ if there exists $f, f' \in \mathcal{H}$ such that $\sqrt{\sum_{i=1}^{n}(f(x_i) - f'(x_i))^2} \leq \epsilon$, but $|f(z) - f'(z)| \geq \epsilon$.

Based on $\epsilon$-independence, the Eluder dimension can be efficiently defined as below.

**Definition 5** (Eluder dimension). Let $\mathcal{H}$ be a function class defined on $\mathcal{X}$. The Eluder dimension $\dim_{\mathrm{E}}(\mathcal{H}, \epsilon)$ is the length of the longest sequence $\{x_1, \ldots, x_n\} \subset \mathcal{X}$ such that there exists $\epsilon' \geq \epsilon$ where $x_i$ is $\epsilon'$-independent of $\{x_1, \ldots, x_{i-1}\}$ for all $i \in [n]$.

The following lemma shows that a model-based hypothesis class with a low Eluder dimension has low AGEC. Motivated by Ayoub et al. (2020); Chen et al. (2022a), we consider the Eluder dimension over function class derived from the model-based hypotheses class $\mathcal{H}$, defined as

$$\mathcal{X}_{\mathcal{H}} := \big\{X_{f,f'}(s, a) = \big(r_f + \mathbb{P}_{f'}V_f\big)(s, a) : f, f' \in \mathcal{H}\big\}.$$

Note that Chen et al. (2022a) considered function class $\mathcal{X}_{\mathcal{H}, \mathcal{V}} := \{\mathbb{P}_f V(s, a) : f \in \mathbb{P}(\mathcal{H}), V \in \mathcal{V}\}$, where $\mathbb{P}(\mathcal{H})$ denotes the hypotheses class over the transition kernel and $\mathcal{V}$ denotes the hypotheses class over the optimal bias function. We remark that definitions over function class based on $(\mathbb{P}_f, V)$ with $(f, V) \in \mathcal{H} \times \mathcal{V}$ and $(\mathbb{P}_f, r_f)$ with $f \in \mathcal{H}$ is almost equivalent and in this paper we focus on $\mathcal{X}_{\mathcal{H}}$ under the latter framework, aligning with the model-based hypothesis (see Definition 2).

**Lemma 1** (Low Eluder dim $\subseteq$ Low AGEC). Consider the discrepancy function

$$l_{f'}(f, g, \zeta_t) = \big(r_g + \mathbb{P}_g V_{f'}\big)(s_t, a_t) - r(s_t, a_t) + V_{f'}(s_{t+1}), \tag{3.3}$$

with $\mathcal{P}(f) = f^*$, and the expectation is taken over $s_{t+1}$ from $\mathbb{P}(\cdot|s_t, a_t)$. Let $d_{\mathrm{E}} = \dim_{\mathrm{E}}(\mathcal{X}_{\mathcal{H}}, \epsilon)$ be the $\epsilon$-Eluder dimension defined over $\mathcal{X}_{\mathcal{H}}$, then we have $d_{\mathrm{G}} \leq 2d_{\mathrm{E}} \cdot \log T$ and $\kappa_{\mathrm{G}} \leq d_{\mathrm{E}}$.

**Average-Reward Bellman Eluder (ABE) Dimension** Before delving into details of the average-reward BE (ABE) dimension, we start with two useful notations, distributional $\epsilon$-independence and distributional Eluder (DE) dimension proposed by Jin et al. (2021), which is a generalization of point-wise $\epsilon$-independence and Eluder dimension defined above (see Definitions 4 and 5).

**Definition 6** (Distributional $\epsilon$-independence). Let $\mathcal{H}$ be a function class defined on $\mathcal{X}$ and sequence $\{v, \mu_1, \ldots, \mu_n\}$ be the probability measures over $\mathcal{X}$. We say $v$ is $\epsilon$-independent of $\{\mu_1, \ldots, \mu_n\}$ with respect to $\mathcal{H}$ if there exists $f \in \mathcal{H}$ such that $\sqrt{\sum_{i=1}^n (\mathbb{E}_{\mu_i}[f])^2} \leq \epsilon$, but $|\mathbb{E}_v[f]| \geq \epsilon$.

**Definition 7** (Distributional Eluder dimension). Let $\mathcal{H}$ be a function class defined on $\mathcal{X}$ and $\Gamma$ be a family of probability measures over $\mathcal{X}$. The distributional Eluder dimension $\dim_{\mathrm{DE}}(\mathcal{H}, \Gamma, \epsilon)$ is the length of the longest sequence $\{\rho_1, \ldots, \rho_n\} \subset \Gamma$ such that there exists $\epsilon' \geq \epsilon$ where $\rho_i$ is $\epsilon'$-independent of the remaining distribution sequence $\{\rho_1, \ldots, \rho_{i-1}\}$ for all $i \in [n]$.

Now we are ready to introduce the average-reward Bellman Eluder (ABE) dimension. It is defined as the distributional Eluder (DE) dimension of average-reward Bellman error in (2.3) over $\mathcal{H}$.

**Definition 8** (ABE dimension). Denote $\mathcal{E}_{\mathcal{H}} = \{\mathcal{E}(f)(s,a) : f \in \mathcal{H}\}$ be the collection of average-reward Bellman errors defined over $\mathcal{S} \times \mathcal{A}$. For any constant $\epsilon > 0$, the $\epsilon$-ABE dimension of given hypotheses class $\mathcal{H}$ is defined as $\dim_{\mathrm{ABE}}(\mathcal{H}, \epsilon) := \dim_{\mathrm{DE}}(\mathcal{E}_{\mathcal{H}}, \mathcal{D}_{\Delta}, \epsilon)$.

The lemma below posits that the value-based hypothesis problem with a low ABE dimension shall have a low AGEC in terms of the Bellman discrepancy.

**Lemma 2** (Low ABE dim $\subseteq$ Low AGEC). Consider the Bellman discrepancy function as defined in (3.1) , and the expectation is taken over the transition state $s_{t+1}$ from $\mathbb{P}(\cdot|s_t, a_t)$. Let $d_{\mathrm{ABE}} = \dim_{\mathrm{ABE}}(\mathcal{H}, \epsilon)$, then we have $d_{\mathrm{G}} \leq 2d_{\mathrm{ABE}} \cdot \log T$ and $\kappa_{\mathrm{G}} \leq d_{\mathrm{ABE}}$.

The Eluder dimension and ABE dimension can capture numerous concrete problems, respectively under model-based and value-based scenarios. Specifically, the Eluder dimension incorporates rich model-based problems like linear mixture AMDPs (Wu et al., 2022), and the ABE dimension can characterize tabular AMDPs (Jaksch et al., 2010), linear AMDPs (Wei et al., 2021), AMDPs with Bellman Completeness, generalized linear AMDPs, and kernel AMDPs, where the latter three problems are newly proposed for AMDPs. Details about the concrete examples are deferred to Appendix C. Combining these facts and Lemmas 1 and 2, we can conclude that AGEC serves as a unified complexity measure, as it encompasses all of these tractable AMDP models illustrated above.

## 4 LOCAL-FITTED OPTIMIZATION WITH OPTIMISM

To solve AMDPs with low AGEC value (see Definition 3), we propose the algorithm Local-fitted Optimization with OPtimism (LOOP), whose pseudocode is given in Algorithm 1. At a high level, LOOP is a modified version of the classical fitted Q-iteration (Szepesvári, 2010) with optimistic planning and lazy policy updates. That is, the policies are only updated when a certain criterion is met (Line 2). When this is the case, LOOP performs three main steps:

- **Optimistic planning** (Line 4.1): Compute the most optimistic $f_t \in \mathcal{H}$ within $\mathcal{B}_t$ that maximizes the corresponding average-reward $J_t$ by solving a constrained optimization problem.

- **Construct confidence set** (Line 4.2): Construct the confidence set $\mathcal{B}_t$ for optimization using $\mathcal{D}_{t-1}$, where all $f_t \in \mathcal{H}$ satisfying $\mathcal{L}_{\mathcal{D}_{t-1}}(f_t, f_t) - \inf_{g \in \mathcal{G}} \mathcal{L}_{\mathcal{D}_{t-1}}(f_t, g) \leq \beta$ is included. Here, $\beta > 0$ defines the radius, corresponding to the log covering number of the hypothesis class $\mathcal{H}$.

- **Execute Policy and Update** $\Upsilon_t$ (Line 8-10): Choose the greedy policy $\pi_t = \pi_{f_t}$ as the exploration policy. Execute policy, collect data, and update trigger $\Upsilon_t = \mathcal{L}_{\mathcal{D}_t}(f_t, f_t) - \inf_{g \in \mathcal{G}} \mathcal{L}_{\mathcal{D}_t}(f_t, g)$.

Note that both the confidence set $\mathcal{B}_t$ and the update condition $\Upsilon_t$ are constructed upon the (cumulative) squared discrepancy, which is crucial to the algorithmic design. It takes the form

$$\mathcal{L}_{\mathcal{D}_t}(f, f) - \inf_{g \in \mathcal{G}} \mathcal{L}_{\mathcal{D}_t}(f, g), \quad \text{where } \mathcal{L}_{\mathcal{D}_t}(f, g) = \sum_{(f_i, \zeta_i) \in \mathcal{D}_t} \|l_{f_i}(f, g, \zeta_i)\|_2^2, \quad (4.1)$$

where $f_i$ and $\zeta_i = (s_i, a_i, s_{i+1})$ are drawn from $\mathcal{D}_t$, and $l_{f'}(f, g, \zeta)$ is the discrepancy function, which varies across different RL problems. The sum of squared discrepancy serves as an empirical estimation of the in-sample training error (see Definition 3). Besides, we highlight two key designs:

---

**Algorithm 1** Local-fitted Optimization with Optimism - $\text{LOOP}(\mathcal{H}, \mathcal{G}, T, \delta)$

---

**Parameter:** Initial $s_1$, span $\text{sp}(V^*)$, optimistic parameter $\beta = c \log \left( T \mathcal{N}_{\mathcal{H} \cup \mathcal{G}}^2 (1/T)/\delta \right) \cdot \text{sp}(V^*)$

**Initialize:** Draw $a_1 \sim \text{Unif}(\mathcal{A})$ and set $\tau_0 \leftarrow 0, \Upsilon_0 \leftarrow 0, \mathcal{B}_0 \leftarrow \emptyset, \mathcal{D}_0 \leftarrow \emptyset$.

1: **for** $t = 1, \ldots, T$ **do**
2:      **if** $t = 1$ or $\Upsilon_{t-1} \geq 4\beta$ **then**
3:          Set $\tau_t = t$.
4:          Solve optimization problem $f_t = \text{argmax}_{f_t \in \mathcal{B}_t} J_{f_t}$, where

$$\mathcal{B}_t = \left\{ f \in \mathcal{H} : \mathcal{L}_{\mathcal{D}_{t-1}}(f, f) - \inf_{g \in \mathcal{G}} \mathcal{L}_{\mathcal{D}_{t-1}}(f, g) \leq \beta \right\}, \qquad (4.2)$$

5:          Compute $Q_t = Q_{f_t}$, $V_t = V_{f_t}$ and $J_t = J_{f_t}$.
6:      **else**
7:          Retain $(f_t, J_t, V_t, Q_t, \tau_t) = (f_{t-1}, J_{t-1}, V_{t-1}, Q_{t-1}, \tau_{t-1})$.
8:      Take $a_t = \text{argmax}_{a \in \mathcal{A}} Q_t(s_t, a)$.
9:      Observe $r_t = r(s_t, a_t)$ and transition state $s_{t+1}$.
10:     Update $\mathcal{D}_t = \mathcal{D}_{t-1} \cup \{(s_t, a_t, r_t, s_{t+1}; f_t)\}$ and $\Upsilon_t = \mathcal{L}_{\mathcal{D}_t}(f_t, f_t) - \inf_{g \in \mathcal{G}} \mathcal{L}_{\mathcal{D}_t}(f_t, g)$.

---

- **Consistent control over discrepancy**: The construction of confidence set $\mathcal{B}_t$ ensures that the cumulative squared discrepancy is controlled at level $\beta$ in each step. To see this, suppose $\tau_t = t$, i.e., policy switches at the $t$-th step, then (4.2) ensures that $\mathcal{L}_{\mathcal{D}_{t-1}}(f_t, f_t) - \inf_{g \in \mathcal{G}} \mathcal{L}_{\mathcal{D}_{t-1}}(f_t, g) \leq \beta$. Otherwise, it we do not switch policy at step $t$, then we must have $\Upsilon_{t-1} = \mathcal{L}_{\mathcal{D}_{t-1}}(f_t, f_t) - \inf_{g \in \mathcal{G}} \mathcal{L}_{\mathcal{D}_{t-1}}(f_t, g) \leq 4\beta$, as hypothesis $f_t = f_{t-1}$ remains unchanged.

- **Lazy policy update**: The regret decomposition in (5.1) elucidates that each policy switch incurs an additional cost of $|V_{t+1}(s_{t+1}) - V_t(s_{t+1})|$ in regret at each step (see (5.3)). This underscores the necessity of implementing lazy updates to achieve sublinear regret. Within the LOOP framework, policy updates occur adaptively, triggered only when a substantial increase in cumulative discrepancy surpassing $3\beta$ has occurred since the last update. Intuitively, a policy switch occurs when there is a notable infusion of new information from newly collected data. Importantly, such gap is pivotal as it provides the theoretical foundation for the implementation of lazy updates, leveraging the problem's transferability structure (see (ii), Definition 3). Here, LOOP employs a threshold of $4\beta$, considering inherent uncertainty and estimation errors between the minimizer $g$ and $\mathcal{P}(f_t)$, ensuring that the out-of-sample error will exceed $\beta$ under the updating rule.

Similar to previous works in general function approximation (Jin et al., 2021; Du et al., 2021), our algorithm lacks a computationally efficient solution for constrained optimization problems. Instead, our focus is on the sample efficiency, as guaranteed by the theorem below.

**Theorem 3** (Regret). Under Assumptions 1-4, there exists constant $c$ such that for any $\delta \in (0, 1)$ and horizon $T$, with probability at least $1 - 5\delta$, the regret of LOOP satisfies that

$$\text{Reg}(T) \leq \mathcal{O}\left( \text{sp}(V^*) \cdot d\sqrt{T\beta} \right),$$

where $\beta = c \log \left( T \mathcal{N}_{\mathcal{H} \cup \mathcal{G}}^2 (1/T)/\delta \right) \cdot \text{sp}(V^*)$ and $d = \max\{\sqrt{d_{\mathrm{G}}}, \kappa_{\mathrm{G}}\}$. Here, $(d_{\mathrm{G}}, \kappa_{\mathrm{G}}) = \text{AGEC}(\mathcal{H}, \{l_f\}, 1/\sqrt{T})$ are AGEC defined in Definition 3, $\mathcal{G}$ is the auxiliary function class defined in Definition 4, and $\mathcal{N}_{\mathcal{H} \cup \mathcal{G}}(\cdot)$ denotes the covering number as defined in Definition 17.

Theorem 3 asserts that both value-based and model-based problems with low AGEC are tractable. Our algorithm LOOP achieves a $\tilde{\mathcal{O}}(\sqrt{T})$ regret and the multiplicative factor depends on span $\text{sp}(V^*)$, problem complexity $\max\{\sqrt{d_{\mathrm{G}}}, \kappa_{\mathrm{G}}\}$ and the log covering number. The proof sketch is provided in Section 5 and the detailed proof is deferred to Appendix D.

## 5 PROOF OVERVIEW OF REGRET ANALYSIS

In this section, we present the proof sketch of Theorem 3. In Section 4, we elucidated the construction of the confidence set and the circumstances in which updates are performed. Here, we delve into the theoretical analysis to substantiate the necessity of such designs.

**Optimism and Regret Decomposition** In LOOP, we apply an optimization based method to ensure the optimism $J_t \geq J^*$ at each step $t \in [T]$. Based on the optimistic algorithm, we propose a new regret decomposition method motivated by the standard performance difference lemma (PDL) in episodic setting (Jiang et al., 2017), following the form as below:

$$\text{Reg}(T) \leq \underbrace{\sum_{t=1}^{T} \mathcal{E}(f_t)(s_t, a_t)}_{\text{Bellman error}} + \underbrace{\sum_{t=1}^{T} \mathbb{E}_{s_{t+1} \sim \mathbb{P}(\cdot|s_t, a_t)}[V_t(s_{t+1})] - V_t(s_t)}_{\text{Realization error}}. \qquad (5.1)$$

**Step 1: Bound over Bellman Error** The control over the Bellman error is achieved through the design of a confidence set and update condition that combinely controls the empirical squared discrepancy. Note that the construction of the confidence set filters $f_t$ with a limited sum of empirical squared discrepancy. Note that $\mathcal{L}_{\mathcal{D}_{t-1}}(f_t, f_t) - \inf_{g \in \mathcal{G}} \mathcal{L}_{\mathcal{D}_{t-1}}(f_t, g)$ can be regarded as an empirical overestimation of the squared discrepancy, controlled at $\mathcal{O}(\beta)$ regardless of updating. Then,

$$\text{In-sample training error} = \sum_{i=1}^{t-1} \|\mathbb{E}_{\zeta_i}[l_{f_i}(f_t, f_t, \zeta_i)]\|_2^2 \lesssim \beta \quad \forall t \in [T], \qquad (5.2)$$

with high probability, and $\beta$ is pre-determined optimistic parameter depends on horizon $T$ and the log $\rho$-covering number. Recall that the dominance coefficient $d_{\text{G}}$ regulates that Bellman error $\lesssim$ $\left[ d_{\text{G}} \sum_{t=1}^{T} \left( \sum_{i=1}^{t-1} \|\mathbb{E}_{\zeta_i}[l_{f_i}(f_t, f_t, \zeta_i)]\|_2^2 \right) \right]^{1/2}$, thus we have Bellman error $\leq \mathcal{O}(\sqrt{d_{\text{G}} \beta T})$.

**Step 2: Bound over Realization error** The realization error is small if the switching cost is low as the concentration arguments indicated that with high probability it holds

$$\text{Realization error} \leq \underbrace{\text{sp}(V^*) \cdot \mathcal{N}(T)}_{\text{Switching cost}} + \underbrace{\mathcal{O}\big(\text{sp}(V^*) \cdot \sqrt{T \log(1/\delta)}\big)}_{\text{Azuma-Hoeffding term}}, \qquad (5.3)$$

where $\mathcal{N}(t)$ denote Switching cost defined as $\mathcal{N}(T) = \#\{t \in [T] : \tau_t \neq \tau_{t-1}\}$. Motivated by the recent work of Xiong et al. (2023), the main idea of low-switching control is summarized below. The key step is that the minimizer $g$ is a good approximator of $\mathcal{P}(f_t)$ such that

$$0 \leq \mathcal{L}_{\mathcal{D}_t}(f_t, \mathcal{P}(f_t)) - \inf_{g \in \mathcal{G}} \mathcal{L}_{\mathcal{D}_t}(f_t, g) \leq \beta, \quad \forall t \in [T] \qquad (5.4)$$

with high probability based on the minimization and the definition of optimistic parameter $\beta$. In the following analysis, we assume that (5.4) holds. Suppose that an update occurs at step $t + 1$, then it implies that $\mathcal{L}_{\mathcal{D}_t}(f_t, f_t) - \inf_{g \in \mathcal{G}} \mathcal{L}_{\mathcal{D}_t}(f_t, g) > 4\beta$ and the latest update at step $\tau_t$ ensures that $\mathcal{L}_{\mathcal{D}_{\tau_t - 1}}(f_{\tau_t}, f_{\tau_t}) - \inf_{g \in \mathcal{G}} \mathcal{L}_{\mathcal{D}_{\tau_t - 1}}(f_{\tau_t}, g) \leq \beta$. Combine (5.4) with the arguments above, we have

$$\text{(i). } \mathcal{L}_{\mathcal{D}_t}(f_t, f_t) - \mathcal{L}_{\mathcal{D}_t}\big(f_t, \mathcal{P}(f_t)\big) > 3\beta, \quad \text{(ii). } \mathcal{L}_{\mathcal{D}_{\tau_t - 1}}(f_{\tau_t}, f_{\tau_t}) - \mathcal{L}_{\mathcal{D}_{\tau_t - 1}}\big(f_{\tau_t}, \mathcal{P}(f_{\tau_t})\big) \leq \beta. \qquad (5.5)$$

Based on the concentration argument and (i), (ii) in (5.5), the out-sample training error between two updates is lower bounded by $\sum_{i=\tau_t}^{t} \|\mathbb{E}_{\zeta_i}[l_{f_i}(f_i, f_i, \zeta_i)]\|_2^2 > \beta$. Let $b_1, \ldots, b_{\mathcal{N}(T)+1}$ be the sequence of updated steps, take summation over the $T$ steps and then we have

$$\mathcal{N}(T)\beta \leq \sum_{u=1}^{\mathcal{N}(T)} \sum_{t=b_u}^{b_{u+1}-1} \|\mathbb{E}_{\zeta_t}[l_{f_t}(f_t, f_t, \zeta_t)]\|_2^2 = \sum_{t=1}^{T} \|\mathbb{E}_{\zeta_t}[l_{f_t}(f_t, f_t, \zeta_t)]\|_2^2 \leq \mathcal{O}(\kappa_{\text{G}} \cdot \beta \log T), \quad (5.6)$$

where the first inequality follows arguments above and the second is based on the definition of transferability (see Definition 3) given $\sum_{i=1}^{t-1} \|\mathbb{E}_{\zeta_i}[l_{f_i}(f_t, f_t, \zeta_i)]\|_2^2 \leq \mathcal{O}(\beta)$ for all $t \in [T]$. Thus, we have $\mathcal{N}(T) \leq \mathcal{O}(\kappa_{\text{G}} \log T)$ and the Realization error is bounded by $\mathcal{O}\big(\kappa_{\text{G}} \cdot \text{sp}(V^*) \log T\big)$. Please refer to Lemma 11 in Appendix D.4 for a formal statement and detailed techniques.

# 6 CONCLUSION

This work studies the infinite-horizon average-reward MDPs under general function approximation. To address the unique challenges of AMDPs, we introduce a new complexity metric — average-reward generalized eluder coefficient (AGEC) and a unified algorithm named Local-fitted Optimization with OPtimism (LOOP). We posit that our work paves the way for future work, including developing more general frameworks for AMDPs and new algorithms with sharper regret bounds.

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

## A BACKGROUNDS AND TECHNICAL NOVELTIES

**Notations.** For any integer $n \in \mathbb{N}^+$, we take the convention to use $[n] = \{1, \ldots, n\}$. Consider two non-negative sequences $\{a_n\}_{n \geq 0}$ and $\{b_n\}_{n \geq 0}$, if $\limsup a_n/b_n < \infty$, then we write it as $a_n = \mathcal{O}(b_n)$. Else if $\limsup a_n/b_n = 0$, then we write it as $a_n = o(b_n)$. And we use $\tilde{\mathcal{O}}$ to omit the logarithmic terms. Denote $\Delta(\mathcal{X})$ be the probability simplex over the set $\mathcal{X}$. Denote by $\sup_x |v(x)|$ the supremum norm of a given function. $x \wedge y$ stands for $\min\{x, y\}$ and $x \vee y$ stands for $\max\{x, y\}$. Given any continuum $\mathcal{S}$, let $|\mathcal{S}|$ be the cardinality. Given two distributions $P, Q \in \Delta(\mathcal{X})$, the TV distance of the two distributions is defined as $\mathrm{TV}(P, Q) = \frac{1}{2}\mathbb{E}_{x \sim P}[|\mathrm{d}Q(x)/\mathrm{d}P(x) - 1|]$.

**Infinite-horizon Average-reward MDPs.** Pioneering works by Auer et al. (2008) and Bartlett & Tewari (2012) laid foundation for model-based algorithms operating within the online framework with sub-linear regret. In recent years, the pursuit of improved regret guarantees has led to the emergence of a multitude of new algorithms. In tabular case, these advancements include numerous model-based approaches (Ouyang et al., 2017; Fruit et al., 2018; Zhang & Ji, 2019; Ortner, 2020) and model-free algorithms (Abbasi-Yadkori et al., 2019; Wei et al., 2020; Hao et al., 2021; Lazic et al., 2021; Zhang & Xie, 2023). In the context of function approximation, POLITEX (Abbasi-Yadkori et al., 2019), a variant of the regularized policy iteration, is the first model-free algorithm with linear value-function approximation, and achieves $\tilde{\mathcal{O}}(T^{\frac{3}{4}})$ regret for the ergodic MDP. The work by Hao et al. (2021) followed the same setting and improved the results to $\tilde{\mathcal{O}}(T^{\frac{2}{3}})$ with an adaptive approximate policy iteration (AAPI) algorithm. Wei et al. (2021) proposed an optimistic Q-learning algorithm FOPO for the linear function approximation, and achieve a near-optimal $\tilde{\mathcal{O}}(\sqrt{T})$ regret. On another line of research, Wu et al. (2022) delved into the linear function approximation under the framework of linear mixture model, which is mutually uncoverable concerning linear MDPs (Wei et al., 2021), and proposed UCRL2-VTR based on the value-targeted regression (Ayoub et al., 2020). Recent work of Chen et al. (2022a) expanded the scope of research by addressing the general function approximation problem in average-reward RL and proposed the SIM-TO-REAL algorithm, which can be regarded as an extension to UCRL2-VTR. In comparison to the works mentioned, our algorithm, LOOP, not only addresses all the problems examined in those studies but also extends its applicability to newly identified models. See Table 1 for a summary.

**Function Approximation in Finite-horizon MDPs.** In the pursuit of developing sample-efficient algorithms capable of handling large state spaces, extensive research efforts have converged on the linear function approximation problems within the finite-horizon setting. See Yang & Wang (2019); Wang et al. (2019); Jin et al. (2020); Ayoub et al. (2020); Cai et al. (2020); Zhou et al. (2021a;b); Zhou & Gu (2022); Agarwal et al. (2022); He et al. (2022); Zhong & Zhang (2023); Zhao et al. (2023); Huang et al. (2023); Li & Sun (2023) and references therein. Furthermore, Wang et al. (2020) studied RL with general function approximation and adopted the eluder dimension (Russo & Van Roy, 2013) as a complexity measure. Before this, Jiang et al. (2017) considered a substantial subset of problems with low Bellman ranks. Building upon these foundations, Jin et al. (2021) combined both the Eluder dimension and Bellman error, thereby broadening the scope of solvable problems under the concept of the Bellman Eluder (BE) dimension. In a parallel line of research, Sun et al. (2019) proposed the witness ranking focusing on the low-rank structures, and Du et al. (2021) extended it to encompass more scenarios with the bilinear class. Besides, Foster et al. (2021; 2023) provided a unified framework, decision estimation coefficient, for interactive decision making. The work of Chen et al. (2022b) extended the value-based GOLF (Jin et al., 2021) with the introduction of the discrepancy loss function to handle the broader admissible Bellman characterization (ABC) class. More recently, Zhong et al. (2022); Liu et al. (2023b) proposed a unified framework measured by generalized eluder coefficient (GEC), an extension to Dann et al. (2021) that captures almost all known tractable problems. All these works are restricted to the finite-horizon regime, and their complexity measure and algorithms are not applicable in the infinite-horizon average-reward setting.

**Low-Switching Cost Algorithms.** Addressing low-switching cost problems in bandit and reinforcement learning has seen notable progress. Abbasi-Yadkori et al. (2011) first proposed an algorithm for linear bandits with $\mathcal{O}(\log T)$ switching cost. Subsequent research extended this to tabular MDPs, including works of Bai et al. (2019); Zhang et al. (2020). A significant stride was made by Kong et al. (2021), who introduced importance scores to handle low-switching cost scenarios in general function approximation with complexity measured by eluder dimension (Russo & Van Roy,

2013). Recently, Xiong et al. (2023) introduced the eluder condition (EC) class, offering a comprehensive framework to address all tractable low-switching cost problems above. In the context of average-reward RL, Wei et al. (2021); Wu et al. (2022); Chen et al. (2022a); Hu et al. (2022) developed low-switching algorithms to control the regret under linear structure or model-based class, leaving a unifying framework for both value-based and model-based problems an open problem.

**Further Elaboration on Our Contributions and Technical Novelties.** Compared to episodic MDPs or discounted MDPs, AMDPs present unique challenges that prevent a straightforward extension of existing algorithms and analyses from these well-studied domains. One notable distinction is a different regret notion in average-reward RL due to a different form of the Bellman optimality equation. Furthermore, such a difference is coupled with the challenge of exploration in the context of general function approximation. To effectively bound this regret, we introduce a new regret decomposition approach within the context of general function approximation (refer to (5.1) and (5.3)). This regret decomposition suggests that the total regret can be controlled by the cumulative Bellman error and the switching cost. Inspired by this, we propose an optimistic algorithm with lazy updates in the general function approximation setting, which uses the residue of the loss function as the indicator for deciding when to conduct policy updates. Such a lazy policy update scheme adaptively divides the total of $T$ steps into $\mathcal{O}(\log T)$ epochs, which is significantly different from (OLSVI.FH; Wei et al., 2021) that reduces the infinite-horizon setting to the finite-horizon setting by splitting the whole learning procedure into several $H$-length epoch, where $H$ typically chosen as $\Theta(\sqrt{T})$ (Wei et al., 2021). We remark that such an adaptive lazy updating design and corresponding analysis are pivotal in achieving the optimal $\tilde{\mathcal{O}}(\sqrt{T})$ rate, as opposed to the $\tilde{\mathcal{O}}(T^{3/4})$ regret in (OLSVI.FH; Wei et al., 2021). Moreover, our approach is an extension to the existing lazy update approaches for average-reward setting (Wei et al., 2021; Wu et al., 2022) that leverages the postulated linear structure and is not applicable to problems with general function approximation. Furthermore, to accommodate the average-reward term, we introduce a new complexity measure AGEC, which characterizes the exploration challenge in general function approximation. Compared with Zhong et al. (2022), our additional transferability restriction is tailored for the infinite-horizon setting and plays a crucial role in analyzing the low-switching error. Despite this additional transferability restriction, AGEC can still serve as a unifying complexity measure in the infinite-horizon average-reward setting, like the role of GEC in the finite-horizon setting. Specifically, AGEC captures a rich class of tractable AMDP models, including all previously recognized AMDPs, including all known tractable AMDPs, and some newly identified AMDPs. See Table 1 for a summary.

**Discussion about distribution families** Beyond the singleton distribution family $\mathcal{D}_\Delta$ taken in this paper, there exists a notable distribution family $\mathcal{D}_\mathcal{H} = \{\mathcal{D}_{\mathcal{H},t}\}_{t\in[T]}$, proposed in Jin et al. (2021), where $\mathcal{D}_{\mathcal{H},t}$ characterizes probability measure over $\mathcal{S} \times \mathcal{A}$ obtained by executing different policies induced by $f_1, \ldots, f_{t-1} \in \mathcal{H}$, measures the detailed distribution under sequential policies. However, in this paper, we exclude the consideration of $\mathcal{D}_\mathcal{H}$ for two principal reasons. First, evaluations of average-reward RL focus on the difference between *observed* rewards $r(s_t, a_t)$ and optimal average reward $J^*$ — as opposed to the expected value $V_h^\pi$ (i.e *expected* sum of reward) under specific policy and optimal value at step $h \in [H]$ in episodic setting — rendering the introduction of $\mathcal{D}_\mathcal{H}$ unnecessary. Second, in infinite settings, the measure of such distribution becomes highly intricate and impractical given *different* policy induced by $f_1, \ldots, f_T$ over a potentially infinite $T$-steps. As a comparison, in the episode setting a *fixed* policy induced by $f_t$ is executed over a finite $H$-step.

# B ALTERNATIVE CHOICES OF DISCREPANCY FUNCTION

Note that there is another line of research that addresses model-based problems using Maximum Likelihood Estimator (MLE)-based approaches (Liu et al., 2023a; Zhong et al., 2022), as opposed to the value-targeted regression with reward function known. We remark that these MLE-based approaches can be also incorporated within our framework through the use of the discrepancy function:

$$l_{f'}(f, g, \zeta_t) = \frac{1}{2}|\mathbb{P}_g(s_{t+1}|s_t, a_t)/\mathbb{P}_{f^*}(s_{t+1}|s_t, a_t) - 1|, \tag{B.2}$$

where the trajectory is $\zeta_t = (s_t, a_t, s_{t+1})$ with expectation taken over the next transition state $s_{t+1}$ from $\mathbb{P}(\cdot|s_t, a_t)$ such that $\mathbb{E}_{\zeta_t}[l_{f'}(f, g, \zeta_t)] = \text{TV}(\mathbb{P}_f(\cdot|s_t, a_t), \mathbb{P}_{f^*}(\cdot|s_t, a_t))$. To accommodate the discrepancy function in (B.2), we introduce a natural variant of AGEC defined below.

---

**Algorithm 2** MLE-based Local-fitted Optimization with Optimism - MLE-LOOP($\mathcal{H}, T, \delta$)

---

**Parameter:** Initial $s_1$, span $\mathrm{sp}(V^*)$, optimistic parameter $\beta = c \log \left( T \mathcal{N}_{\mathcal{H}}(1/T)/\delta \right)$
**Initialize:** Draw $a_1 \sim \mathrm{Unif}(\mathcal{A})$ and set $\tau_0 \leftarrow 0, \Upsilon_0 \leftarrow 0, \mathcal{B}_0 \leftarrow \emptyset, \mathcal{D}_0 \leftarrow \emptyset$.
 1: **for** $t = 1, \ldots, T$ **do**
 2:      **if** $t = 1$ or $\Upsilon_{t-1} \geq 3\sqrt{\beta t}$ **then**
 3:          Update $\tau_t = t$.
 4:          Solve optimization problem $f_t = \mathrm{argmax}_{f_t \in \mathcal{B}_t} J_{f_t}$, where

$$\mathcal{B}_t = \left\{ f \in \mathcal{H} : \mathcal{L}_{\mathcal{D}_{t-1}}(f,f) - \inf_{g \in \mathcal{G}} \mathcal{L}_{\mathcal{D}_{t-1}}(f,g) \leq \beta \right\}, \tag{B.1}$$

 5:          Update $Q_t = Q_{f_t}, V_t = V_{f_t}, J_t = J_{f_t}$ and $g_t = \inf_{g \in \mathcal{G}} \mathcal{L}_{\mathcal{D}_{t-1}}(f_t, g)$.
 6:      **else**
 7:          Retain $(f_t, g_t, J_t, V_t, Q_t, \tau_t) = (f_{t-1}, g_{t-1}, J_{t-1}, V_{t-1}, Q_{t-1}, \tau_{t-1})$.
 8:      Execute $a_t = \mathrm{argmax}_{a \in \mathcal{A}} Q_t(s_t, a)$.
 9:      Collect reward $r_t = r(s_t, a_t)$ and transition state $s_{t+1}$.
10:      Update $\mathcal{D}_t = \mathcal{D}_{t-1} \cup \{(s_t, a_t, r_t, s_{t+1})\}, \Upsilon_t = \sum_{s,a \in \mathcal{D}_t} \mathrm{TV}\left( \mathbb{P}_{f_t}(\cdot|s,a), \mathbb{P}_{g_t}(\cdot|s,a) \right)$.

---

**Definition 9** (MLE-AGEC). Given hypothesis class $\mathcal{H}$, the MLE-discrepancy function $\{l_f\}_{f \in \mathcal{H}}$ in (B.2) and constant $\epsilon > 0$, the MLE-based average-reward generalized eluder coefficients MLE-AGEC($\mathcal{H}, \{l_f\}, \epsilon$) is defined as the smallest coefficients $\kappa_{\mathrm{G}}$ and $d_{\mathrm{G}}$ such that following two conditions hold with absolute constants $C_1 > 0$:

(i) (MLE-Bellman dominance) There exists constant $d_{\mathrm{G}} > 0$ such that

$$\sum_{t=1}^{T} \mathcal{E}(f_t)(s_t, a_t) \leq d_{\mathrm{G}} \cdot \mathrm{sp}(V^*) \sum_{t=1}^{T} \|\mathbb{E}_{\zeta_i}[l_{f_t}(f_t, f_t, \zeta_i)]\|_1.$$

(ii) (MLE-Transferability) There exists constant $\kappa_{\mathrm{G}} > 0$ such that for hypotheses $f_1, \ldots, f_T \in \mathcal{H}$, if it holds that $\sum_{i=1}^{t-1} \|\mathbb{E}_{\zeta_i}[l_{f_i}(f_t, f_t, \zeta_i)]\|_1 \leq \sqrt{\beta t}$ for all $t \in [T]$, then we have

$$\sum_{t=1}^{T} \|\mathbb{E}_{\zeta_t}[l_{f_t}(f_t, f_t, \zeta_t)]\|_1 \leq \mathrm{poly}(\log T)\sqrt{\kappa_{\mathrm{G}} \cdot \beta T} + C_1 \cdot \mathrm{sp}(V^*)^2 \min\{\kappa_{\mathrm{G}}, T\} + 2T\epsilon^2.$$

The main difference between the MLE-based variant (see Definition 9) and the original AGEC (see Definition 3) is that the coefficients are defined over the $\ell_1$-norm rather than the $\ell_2$-norm, and a similar condition is considered in Liu et al. (2023a); Xiong et al. (2023). Now, we are ready to introduce the algorithm for the alternative discrepancy function in (B.2) and please see Algorithm 2 for complete pseudocode. The main modification lies in the construction of confidence set and update condition $\Upsilon_t$. Here, the confidence set now follows

$$\mathcal{B}_t = \{f_t \in \mathcal{H} : \mathcal{L}_{\mathcal{D}_{t-1}}(f_t, f_t) - \inf_{g \in \mathcal{G}} \mathcal{L}_{\mathcal{D}_{t-1}}(f_t, g) \leq \beta\}, \quad \mathcal{L}_{\mathcal{D}}(f,g) = - \sum_{(s,a,s') \in \mathcal{D}} \log \mathbb{P}_g(s'|s,a).$$

In comparison, the update condition follows that $\Upsilon_t = \sum_{(s,a) \in \mathcal{D}_t} \mathrm{TV}\left( \mathbb{P}_{f_t}(\cdot|s,a), \mathbb{P}_{g_t}(\cdot|s,a) \right)$. Unlike the standard LOOP algorithm, the the confidence set and update condition in the MLE-based varint no longer shares the same construction. Following the literature of MLE-based algorithms, we adopt the bracket number to approximation the cardinality of the function class.

**Definition 10** ($\rho$-bracket). Let $\rho > 0$ and $\mathcal{F}$ is a set of functions defined over $\mathcal{X}$. Under $\ell_1$-norm, a set of functions $\mathcal{V}_\rho(\mathcal{F})$ is an $\rho$-bracket of $\mathcal{F}$ if for any $f \in \mathcal{F}$, there exists a function $f' \in \mathcal{F}$ such that the following two properties hold: (i) $f'(x) \geq f(x)$ for all $x \in \mathcal{X}$, and (ii) $\|f - f'\|_1 \leq \rho$. The bracketing number $\mathcal{B}_{\mathcal{F}}(\rho)$ is the cardinality of the smallest $\rho$-bracket needed to cover $\mathcal{F}$.

The theoretical guarantee is provided below.

**Theorem 4** (Cumulative regret). Under Assumptions 1-2 and the discepancy function in (B.2) with self-completeness such that $\mathcal{G} = \mathcal{H}$, there exists constant $c$ such that for any $\delta \in (0,1)$ and time horizon $T$, with probability at least $1 - 4\delta$, the regret of MLE-LOOP satisfies that

$$\text{Reg}(T) \leq \mathcal{O}\Big(\text{sp}(V^*) \cdot d\sqrt{T\beta}\Big),$$

where $\beta = c \log \big(T\mathcal{B}_{\mathcal{H}}(1/T)/\delta\big) \cdot \text{sp}(V^*)$, $d = d_{\text{G}}\sqrt{\kappa_{\text{G}}}$. Here, $(d_{\text{G}}, \kappa_{\text{G}}) = \text{MLE-AGEC}(\mathcal{H}, \{l_f\}, 1/\sqrt{T})$ denote MLE-AGEC defined in Definition 9 and $\mathcal{B}_{\mathcal{H}}(\cdot)$ denotes the bracketing number.

The proof of Theorem 4 is similar to that of Theorem 3, and can be found in Appendix H.1.

## C  CONCRETE EXAMPLES

In this section, we present concrete examples of problems for AMDP. We remark that the understanding of function approximation problems under the average-reward setting is quite limited, and to our best knowledge, existing works have primarily focused on linear approximation (Wei et al., 2021; Wu et al., 2022) and model-based general function approximation (Chen et al., 2022a). Here, we introduce a variety of function classes with low AGEC. Beyond the examples considered in existing work, these newly proposed function classes are mostly natural extensions from their counterpart the finite-horizon episode setting (Jin et al., 2020; Zanette et al., 2020; Du et al., 2021; Domingues et al., 2021), which can be extended to the average-reward problems with moderate justifications.

### C.1  LINEAR FUNCTION APPROXIMATION AND VARIANTS

**Linear function approximation**   Consider the linear FA, which encompasses a wide range of concrete problems with state-action bias function linear in a $d$-dimensional feature mapping. Specifically, a linear function class $\mathcal{H}$ is defined as $\mathcal{H} = \{Q(\cdot,\cdot) = \langle \omega, \phi(\cdot,\cdot)\rangle, J \in \mathcal{J}_{\mathcal{H}} | \|\omega\|_2 \leq \frac{1}{2}\text{sp}(V^*)\sqrt{d}\}$, where the feature satisfies that $\|\phi\|_{2,\infty} \leq \sqrt{2}$ with first coordinate fixed to 1. We remark that such scaling is without loss of generality as justified in Lemma 19. To begin with, we first introduce two specific problems: linear AMDP and AMDP with linear Bellman completion.

**Definition 11** (Linear AMDP, Wei et al. (2021)). There exists a known feature mapping $\phi : \mathcal{S} \times \mathcal{A} \mapsto \mathbb{R}^d$, an unknown $d$-dimensional signed measures $\mu = (\mu_1, \ldots, \mu_d)$ over $\mathcal{S}$, and an unknown reward parameter $\theta \in \mathbb{R}^d$, such that the transition kernel the reward function can be written as

$$\mathbb{P}(\cdot|s,a) = \langle \phi(s,a), \mu(\cdot)\rangle, \quad r(s,a) = \langle \phi(s,a), \theta\rangle. \tag{C.1}$$

for all $(s,a) \in \mathcal{S} \times \mathcal{A}$. Without loss of generality, we assume that the feature mapping $\phi$ satisfies that $\|\phi\|_{2,\infty} \leq \sqrt{2}$ with first coordinate fixed to 1, $\|\theta\|_2 \leq \sqrt{d}$ and $\|\mu(\mathcal{S})\|_2 \leq \sqrt{d}$, where we denote $\mu(\mathcal{S}) = (\mu_1(\mathcal{S}), \ldots, \mu_d(\mathcal{S}))$ and $\mu_i(\mathcal{S}) = \int_{\mathcal{S}} \mathrm{d}\mu_i(s)$ be the total measure of $\mathcal{S}$.

We remark that the scaling on the feature mapping can help in overcoming the gap between the episodic setting and the average-reward one by ensuring the linear structure of Q- and V-value function under optimality (Wei et al., 2021). To illustrate the necessity, note that

$$Q^*(s,a) = r(s,a) + \mathbb{E}_{s' \sim \mathbb{P}(s,a)}[V^*(s')] - J^* = \phi(s,a)^\top \left(\theta - J^* \mathbf{e}_1 + \int_{\mathcal{S}} V^*(s')\mathrm{d}\mu(s')\right),$$

where denote $\mathbf{e}_1 = (1, 0, \ldots, 0) \in \mathbb{R}^d$. Next, we provide the AMDPs with linear Bellman completion, modified from Zanette et al. (2020), which is a more general setting than linear AMDPs.

**Definition 12** (Linear Bellman completion). There exists a known feature mapping $\phi : \mathcal{S} \times \mathcal{A} \mapsto \mathbb{R}^d$ such that for all $(s,a) \in \mathcal{S} \times \mathcal{A}, \omega \in \mathcal{W}_{\mathcal{H}}$ and $J \in \mathcal{J}_{\mathcal{H}}$, we have

$$\langle \mathcal{T}(\omega, J), \phi(s,a)\rangle := r(s,a) + \mathbb{E}_{s' \sim \mathbb{P}(\cdot|s,a)} \left[\max_{a' \in \mathcal{A}}\Big\{\omega^\top \phi(s',a')\Big\}\right] - J, \tag{C.2}$$

**Generalized linear function approximation**   To introduce the nonlinearity beyond linear FA, we extend by incorporating a link function. In generalized linear FA, the hypotheses class is defined as

$\mathcal{H} = \{Q(\cdot, \cdot) = \sigma\left(\omega^\top \phi(\cdot, \cdot)\right), J \in \mathcal{J}_{\mathcal{H}} | \|\omega\|_2 \leq \sqrt{d}\}$, where $\|\phi(s, a)\|_{2,\infty} \leq 1$ and $\sigma : \mathbb{R} \mapsto \mathbb{R}$ is an $\alpha$-bi-Lipschitz function with $\|\sigma\|_\infty \leq \frac{1}{2}\mathrm{sp}(V^*)$. We say $\sigma$ is $\alpha$-bi-Lipschitz continuous if

$$\frac{1}{\alpha} \cdot |x - x'| \leq |\sigma(x) - \sigma(x')| \leq \alpha \cdot |x - x'|, \quad \forall x, x' \in \mathbb{R}. \tag{C.3}$$

We remark that the generalized linear function class $\mathcal{H}$ degenerates to the standard linear function class $\mathcal{H}$ if we choose $\sigma(x) = x$. Modified from Wang et al. (2019) for the episodic setting, we define AMDPs with generalized linear Bellman completion as follows.

**Definition 13** (Generalized linear Bellman completion). There exists a known feature mapping $\phi : \mathcal{S} \times \mathcal{A} \mapsto \mathbb{R}^d$ such that for all $(s, a) \in \mathcal{S} \times \mathcal{A}$, $\omega \in \mathcal{W}_{\mathcal{H}}$ and $J \in \mathcal{J}_{\mathcal{H}}$, we have

$$\sigma\left(\mathcal{T}(\omega, J)^\top \phi(\cdot, \cdot)\right) := r(s, a) + \mathbb{E}_{s' \sim \mathbb{P}(\cdot | s, a)} \left[ \max_{a' \in \mathcal{A}} \left\{ \sigma\left(\omega^\top \phi(s', a')\right) \right\} \right] - J. \tag{C.4}$$

The proposition below states that (generalized) linear function classes have low AGEC.

**Proposition 5** (Linear FA $\subset$ Low AGEC). Consider linear function class $\mathcal{H}_{\mathrm{Lin}}$ and generalized linear function class $\mathcal{H}_{\mathrm{Glin}}$ with a $d$-dimensional feature mapping $\phi : \mathcal{S} \times \mathcal{A} \mapsto \mathbb{R}^d$, if the problem follows one of Definitions 11-13, then it have low AGEC under Bellman discrepancy in (3.1):

$$d_{\mathrm{G}} \leq \mathcal{O}\left(d \log\left(\mathrm{sp}(V^*)\sqrt{d}\epsilon^{-1}\right) \log T\right), \quad \kappa_{\mathrm{G}} \leq \mathcal{O}\left(d \log\left(\mathrm{sp}(V^*)\sqrt{d}\epsilon^{-1}\right)\right).$$

**Linear $Q^*/V^*$ AMDP** Moreover, we consider the linear $Q^*/V^*$ AMDPs, which is modified from the one in Du et al. (2021) under the episodic setting.

**Definition 14** (Linear $Q^*/V^*$ AMDP). There exists known feature mappings $\phi : \mathcal{S} \times \mathcal{A} \mapsto \mathbb{R}^{d_1}$, $\psi : \mathcal{S} \mapsto \mathbb{R}^{d_2}$, and unknown vectors $\omega^* \in \mathbb{R}^{d_1}$, $\theta^* \in \mathbb{R}^{d_2}$ such that optimal value functions follow

$$Q^*(s, a) = \langle \phi(s, a), \omega^* \rangle, \quad V^*(s') = \langle \psi(s'), \theta^* \rangle,$$

for all $(s, a, s') \in \mathcal{S} \times \mathcal{A} \times \mathcal{S}$. Without loss of generality, we assume that features $\|\phi\|_{2,\infty} \leq \sqrt{2}$ and $\|\psi\|_{2,\infty} \leq \sqrt{2}$ with first coordinate fixed to 1, and $\|\omega^*\|_2 \leq \frac{1}{2}\mathrm{sp}(V^*)\sqrt{d_1}$, $\|\theta^*\|_2 \leq \frac{1}{2}\mathrm{sp}(V^*)\sqrt{d_2}$.

The proposition below states that linear $Q^*/V^*$ also has low AGEC.

**Proposition 6** (Linear $Q^*/V^* \subset$ Low AGEC). Linear $Q^*/V^*$ AMDPs with coupled $(d_1, d_2)$-dimensional feature mappings $\phi : \mathcal{S} \times \mathcal{A} \mapsto \mathbb{R}^{d_1}$ and $\psi : \mathcal{S} \mapsto \mathbb{R}^{d_2}$ have low AGEC such that

$$d_{\mathrm{G}} \leq \mathcal{O}\left(d^+ \log\left(\mathrm{sp}(V^*)\sqrt{d^+}\epsilon^{-1}\right) \log T\right), \quad \kappa_{\mathrm{G}} \leq \mathcal{O}\left(d^+ \log\left(\mathrm{sp}(V^*)\sqrt{d^+}\epsilon^{-1}\right)\right),$$

where denote $d^+ = d_1 + d_2$ as the sum of dimensions of features.

The proposition above asserts that in linear $Q^*/V^*$ AMDPs, with additional structural information in state bias function, add $\tilde{\mathcal{O}}(d_2)$ in complexity from the AGEC perspective. We remark that linear FA, generalized linear FA, and linear $Q^*/V^*$ AMDPs are typical value-based problems. The proof of this proposition relies on ABE dimension as an intermediate, and then uses Lemma 2.

## C.2 KERNEL FUNCTION APPROXIMATION

In this subsection, we first introduce the notion of effective dimension. With this notion, we prove a useful proposition that any kernel function class with a low effective dimension has low AGEC. Consider kernel FA, a natural extension to linear FA from $d$-dimensional Euclidean space $\mathbb{R}^d$ to a decomposable kernel Hilbert space $\mathcal{K}$. Formally, a kernel function class is defined as $\mathcal{H} = \{Q(\cdot, \cdot) = \langle \phi(\cdot, \cdot), \omega \rangle_{\mathcal{K}}, J \in \mathcal{J}_{\mathcal{H}} | \|\omega\|_{\mathcal{K}} \leq \mathrm{sp}(V^*)R\}$, where the feature mapping $\phi : \mathcal{S} \times \mathcal{A} \mapsto \mathcal{K}$ satisfies that $\|\phi\|_{\mathcal{K},\infty} \leq 1$. To measure the complexity of problems in a Hilbert space $\mathcal{K}$ with a potentially infinite dimension, we introduce the $\epsilon$-effective dimension below.

**Definition 15** ($\epsilon$-effective dimension). Consider a set $\mathcal{Z}$ with the possibly infinite elements in a given separable Hilbert space $\mathcal{K}$, the $\epsilon$-effective dimension, denoted by $\dim_{\mathrm{eff}}(\mathcal{Z}, \epsilon)$, is defined as the length $n$ of the longest sequence satisfying the condition below:

$$\sup_{\mathbf{z}_1, \ldots, \mathbf{z}_n \in \mathcal{Z}} \left\{ \frac{1}{n} \log \det\left(\mathbf{I} + \frac{1}{\epsilon^2} \sum_{t=1}^n \mathbf{z}_i \mathbf{z}_i^\top \right) \leq \frac{1}{e} \right\}.$$

Here, the concept of $\epsilon$-effective dimension is inspired by the measurement of maximum information gain (Srinivas et al., 2009) and is later introduced as a complexity measure of Hilbert space in Du et al. (2021); Zhong et al. (2022). Similar to Jin et al. (2021), we augment the assumption by requiring the $\mathcal{H}$ to be self-complete under average-reward Bellman operator, i.e., $\mathcal{G} = \mathcal{H}$. Next, the proposition below demonstrates that kernel FA has low AGEC.

**Proposition 7** (Kernel FA $\subset$ Low AGEC). Under the self-completeness, kernel FA with function class $\mathcal{H}_{\mathrm{Ker}}$ concerning a known feature mapping $\phi : \mathcal{S} \times \mathcal{A} \mapsto \mathcal{K}$ have low AGEC such that

$$d_{\mathrm{G}} \le \dim_{\mathrm{eff}}\big(\mathcal{X}, \epsilon/2\mathrm{sp}(V^*)R\big) \log T, \quad \kappa_{\mathrm{G}} \le \dim_{\mathrm{eff}}\big(\mathcal{X}, \epsilon/2\mathrm{sp}(V^*)R\big),$$

where denote $\mathcal{X} = \{\phi(s,a) : (s,a) \in \mathcal{S} \times \mathcal{A}\}$ as the collection of feature mappings.

The proposition above shows that the kernel FA with a low $\epsilon$-effective dimension over the Hilbert space also has low AGEC. As a special case of kernel FA, if we choose $\mathcal{K} = \mathbb{R}^d$, then we can prove that the RHS in the proposition above is upper bounded by $\tilde{\mathcal{O}}(d)$.

### C.3 LINEAR MIXTURE AMDP

In this subsection, we focus on the average-reward linear mixture problem considered in Wu et al. (2022). In this context, the hypotheses function class is defined as $\mathcal{H} = \{\mathbb{P}(s'|s,a) = \langle \theta, \phi(s,a,s')\rangle$,
$r(s,a) = \langle \theta, \psi(s,a)\rangle |\|\theta\|_2 \le 1\}$ with known feature mappings $\phi : \mathcal{S} \times \mathcal{A} \times \mathcal{S} \mapsto \mathbb{R}^d$, $\psi : \mathcal{S} \times \mathcal{A} \mapsto \mathbb{R}^d$, and an unknown parameter $\theta \in \mathbb{R}^d$. The problem is defined as below.

**Definition 16** (Linear mixture AMDPs, Wu et al. (2022)). There exists a known feature mapping $\phi : \mathcal{S} \times \mathcal{A} \times \mathcal{S} \mapsto \mathbb{R}^d$, $\psi : \mathcal{S} \times \mathcal{A} \mapsto \mathbb{R}^d$, and an unknown vector $\theta \in \mathbb{R}^d$, it holds that

$$\mathbb{P}(s'|s,a) = \langle \theta, \phi(s,a,s')\rangle, \quad r(s,a) = \langle \theta, \psi(s,a)\rangle,$$

for all $(s,a,s') \in \mathcal{S} \times \mathcal{A} \times \mathcal{S}$. Without loss of generality, we assume $\|\phi\|_{2,\infty} \le \sqrt{d}$, $\|\psi\|_{2,\infty} \le \sqrt{d}$.

Now we show that the linear mixture problem is tractable under the framework of AGEC.

**Proposition 8** (Linear mixture $\subset$ Low AGEC). Consider linear mixture problem with hypotheses class $\mathcal{H}$ and $d$-dimensional feature mappings $(\phi, \psi)$. If we choose discrepancy function as

$$l_{f'}(f, g, \zeta_t) = \theta_g^\top \left( \psi(s_t, a_t) + \int_{\mathcal{S}} \phi(s_t, a_t, s') V_{f'}(s')\mathrm{d}s' \right) - r(s_t, a_t) - V_{f'}(s_{t+1}), \quad \text{(C.5)}$$

and takes $\mathcal{H} = \mathcal{G}$ with operator following $\mathcal{P}(f) = f^*$ for all $f \in \mathcal{H}$, it has low AGEC such that

$$d_{\mathrm{G}} \le \mathcal{O}\left( d\log\left(\mathrm{sp}(V^*)T/\sqrt{d}\epsilon\right)\right), \quad \kappa_{\mathrm{G}} \le \mathcal{O}\left( d\log\left(\mathrm{sp}(V^*)T/\sqrt{d}\epsilon\right)\right).$$

The proposition posits that AGEC can capture the linear mixture AMDP, based on a modified version of the Bellman discrepancy function in (2.2). In contrast to the linear FA discussed in Appendix C.1, the presence of the average-reward term in this model-based problem does not impose any additional computational or statistical burden, and there is no need for structural assumptions on feature mappings, such as a fixed first coordinate, considering discrepancy in (C.5).

## D PROOF OF MAIN RESULTS FOR LOOP

### D.1 PROOF OF THEOREM 3

*Proof of Theorem 3.* Note that the regret can be decomposed as

$$\mathrm{Reg}(T) = \sum_{t=1}^{T}\left( J^* - r(s_t, a_t)\right) \le \sum_{t=1}^{T}\left( J_t - r(s_t, a_t)\right) \qquad \text{(optimism)}$$

$$\overset{(a)}{=} \sum_{t=1}^{T} \mathcal{E}(f_t)(s_t, a_t) - \sum_{t=1}^{T} \mathbb{E}_{s_{t+1}\sim\mathbb{P}(\cdot|s_t,a_t)}\left[ Q_t(s_t,a_t) - \max_{a\in\mathcal{A}} Q_t(s_{t+1}, a)\right]$$

$$\overset{(b)}{=} \underbrace{\sum_{i=1}^{T} \mathcal{E}(f_t)(s_t, a_t)}_{\text{Bellman error}} + \underbrace{\sum_{t=1}^{T}\left[ \mathbb{E}_{s_{t+1}\sim\mathbb{P}(\cdot|s_t,a_t)}[V_t(s_{t+1})] - V_t(s_t)\right]}_{\text{Realization error}}, \qquad \text{(D.1)}$$

where step $(a)$ and step $(b)$ follow the definition of the Bellman optimality operator and the greedy policy. Below, we will present the bound of Bellman error and Realization error respectively.

**Step 1: Bound over Bellman error**  Recall that the of confidence set ensures that $\mathcal{L}_{\mathcal{D}_{t-1}}(f_t, f_t)$ $- \inf_{g \in \mathcal{G}} \mathcal{L}_{\mathcal{D}_{t-1}}(f_t, g) \leq \mathcal{O}(\beta)$ across all steps. Using the concentration arguments, we can infer

$$\sum_{i=1}^{t-1} \|\mathbb{E}_{\zeta_i}[l_{f_i}(f_t, f_t, \zeta_i)]\|_2^2 \leq \mathcal{O}(\beta), \tag{D.2}$$

with high probability and the formal statements are deferred to Lemma 10 in Appendix D.3. In the following arguments, we assume the above event holds. Take $\epsilon = 1/\sqrt{T}$, recall the definition of dominance coefficient $d_\mathrm{G}$ in $\mathrm{AGEC}(\mathcal{H}, \mathcal{J}, l, \epsilon)$ and it directly indicates that

$$\text{Bellman error} = \sum_{t=1}^{T} \mathcal{E}(f_t)(s_t, a_t) \leq \left[d_\mathrm{G} \sum_{t=1}^{T} \sum_{i=1}^{t-1} \|\mathbb{E}_{\zeta_i}[l_{f_i}(f_t, f_t, \zeta)]\|_2^2\right]^{1/2} + \mathcal{O}\left(\mathrm{sp}(V^*)\sqrt{d_\mathrm{G}T}\right),$$

and thus the Bellman error can be upper bounded by $\mathcal{O}\left(\mathrm{sp}(V^*)\sqrt{d_\mathrm{G}\beta T}\right)$.

**Step 2: Bound over Realization error**  To bound Realization error, we use the concentration argument and the upper-boundded switching cost. Note that

$$\text{Realization error} \overset{(c)}{=} \sum_{t=1}^{T} [V_t(s_{t+1}) - V_t(s_t)] + \mathcal{O}(\mathrm{sp}(V^*)\sqrt{T\log(1/\delta)}),$$

$$= \sum_{t=1}^{T} \left[V_{\tau_t}(s_{t+1}) - V_{\tau_{t+1}}(s_{t+1})\right] + \mathcal{O}(\mathrm{sp}(V^*)\sqrt{T\log(1/\delta)}), \tag{Shift}$$

$$= \sum_{t=1}^{T} \left[V_{\tau_t}(s_{t+1}) - V_{\tau_{t+1}}(s_{t+1})\right] \mathbb{1}(\tau_t \neq \tau_{t+1}) + \mathcal{O}(\mathrm{sp}(V^*)\sqrt{T\log(1/\delta)})$$

$$\leq \mathrm{sp}(V^*) \cdot \mathcal{N}(T) + \mathcal{O}(\mathrm{sp}(V^*)\sqrt{T\log(1/\delta)}) \overset{(d)}{\leq} \mathcal{O}(\mathrm{sp}(V^*) \cdot \kappa_\mathrm{G}\sqrt{T\log(1/\delta)}), \tag{D.3}$$

where step $(c)$ directly follows the Azuma-Hoeffding inequality and step $(d)$ is based the fact that $\|V_{\tau_t} - V_{\tau_{t+1}}\|_\infty \leq \mathrm{sp}(V^*)$ and the bounded switching cost such that $\mathcal{N}(T) \leq \mathcal{O}(\kappa_\mathrm{G} \log T)$, where $\kappa_\mathrm{G}$ is the transferability coefficient in AGEC with $\epsilon = 1/\sqrt{T}$. Please refer to Lemma 11 in Appendix D.4 for the detailed statement and proof of the bounded switching cost.

**Step 3: Combine the bounded erroes**  Plugging (D.2) and (D.3) back into (D.1), we have

$$\mathrm{Reg}(T) \leq \text{Bellman error} + \text{Realization error}$$

$$\leq \mathcal{O}\left(\mathrm{sp}(V^*)\sqrt{d_\mathrm{G}\beta T}\right) + \mathcal{O}(\mathrm{sp}(V^*)\kappa_\mathrm{G}\sqrt{T\log(1/\delta)}) = \mathcal{O}\left(\mathrm{sp}(V^*) \cdot d\sqrt{T\beta}\right),$$

where $d = \max\{\sqrt{d_\mathrm{G}}, \kappa_\mathrm{G}\}$ is a function of $(d_\mathrm{G}, \kappa_\mathrm{G}) = \mathrm{AGEC}(\mathcal{H}, \{l_f\}_{f \in \mathcal{H}}, 1/\sqrt{T})$. In the arguments above, the optimistic parameter is chosen as $\beta = c \log\left(T\mathcal{N}_{\mathcal{H} \cup \mathcal{G}}^2(1/T)/\delta\right) \cdot \mathrm{sp}(V^*)$, which takes the upper bound of the optimistic parameters, aligning with the choice in Lemma 9, Lemma 10, and Lemma 11. Then finish the proof of cumulative regret for LOOP in Algorithm 1. $\qquad\square$

## D.2 PROOF OF LEMMA 9

**Lemma 9** (Optimism).  Under Assumptions 1-4, LOOP is an optimistic algorithm such that it ensures $J_t \geq J^*$ for all $t \in [T]$ with probability greater than $1 - \delta$.

*Proof of Lemma 9.*  Denote $\mathcal{V}_\rho(\mathcal{G})$ be the $\rho$-cover of $\mathcal{G}$ and $\mathcal{N}_{\mathcal{G}}(\rho)$ be the size of $\rho$-cover $\mathcal{V}_\rho(\mathcal{G})$. Consider fixed $(i, g) \in [T] \times \mathcal{G}$ and define the auxiliary function

$$X_{i,f_i}(g) := \|l_{f_i}(f^*, g, \zeta_i)\|_2^2 - \|l_{f_i}(f^*, f^*, \zeta_i)\|_2^2, \tag{D.4}$$

where $f^*$ is the optimal hypothesis in value-based problems and the true hypothesis in model-based ones. Let $\mathscr{F}_t$ be the filtration induced by $\{s_1, a_1, \ldots, s_t, a_t\}$ and note that $f_1, \ldots, f_t$ is fixed under the filtration, then we have

$$
\begin{aligned}
\mathbb{E}[X_{i,f_i}(g)|\mathscr{F}_i] &= \mathbb{E}_{\zeta_i}[\|l_{f_i}(f^*, g, \zeta_i)\|_2^2 - \|l_{f_i}(f^*, \mathcal{P}(f^*), \zeta_i)\|_2^2|\mathscr{F}_i] \\
&= \mathbb{E}_{\zeta_i}\Big[\big[l_{f_i}(f^*, g, \zeta_i) - l_{f_i}(f^*, \mathcal{P}(f^*), \zeta_i)\big] \cdot \big[l_{f_i}(f^*, g, \zeta_i) + l_{f_i}(f^*, \mathcal{P}(f^*), \zeta_i)\big]\Big|\mathscr{F}_i\Big] \\
&= \mathbb{E}_{\zeta_i}\Big[\mathbb{E}_{\zeta_i}\big[l_{f_i}(f^*, g, \zeta_i)\big] \cdot \big[l_{f_i}(f^*, g, \zeta_i) + l_{f_i}(f^*, \mathcal{P}(f^*), \zeta_i)\big]\Big|\mathscr{F}_i\Big] \\
&= \|\mathbb{E}_{\zeta_i}\big[l_{f_i}(f^*, g, \zeta_i)\big]\|_2^2,
\end{aligned}
$$

where the equation follows the definition of generalized completeness (see Assumption 4):

$$
\begin{cases}
\mathbb{E}_{\zeta_i}[l_{f'}(f, g, \zeta)] = l_{f'}(f, g, \zeta) - l_{f'}(f, \mathcal{P}(f), \zeta), \\
\mathbb{E}_{\zeta_i}[l_{f'}(f, g, \zeta)] = \mathbb{E}_{\zeta_i}\big[l_{f'}(f, g, \zeta) + l_{f'}(f, \mathcal{P}(f), \zeta)\big].
\end{cases}
$$

Similarly, we can obtain that the second moment of the auxiliary function is bounded by

$$
\mathbb{E}[X_{i,f_i}(g)^2|\mathscr{F}_i] \leq \mathcal{O}\Big(\mathrm{sp}(V^*)^2 \|\mathbb{E}_{\zeta_i}[l_{f_i}(f^*, g, \zeta_i)]\|_2^2\Big),
$$

By Freedman's inequality (see Lemma 18), with probability greater than $1 - \delta$ it holds that

$$
\begin{aligned}
&\left|\sum_{i=1}^t X_{i,f_i}(g) - \sum_{i=1}^t \|\mathbb{E}_{\zeta_i}[l_{f_i}(f^*, g, \zeta_i)]\|_2^2\right| \\
&\leq \mathcal{O}\left(\sqrt{\log(1/\delta) \cdot \mathrm{sp}(V^*)^2 \sum_{i=1}^t \|\mathbb{E}_{\zeta_i}[l_{f_i}(f^*, g, \zeta_i)]\|_2^2} + \log(1/\delta)\right).
\end{aligned}
$$

By taking union bound over $[T] \times \mathcal{V}_\rho(\mathcal{G})$, for any $(t, \phi) \in [T] \times \mathcal{V}_\rho(\mathcal{G})$ we have $-\sum_{i=1}^t X_{i,f_i}(\phi) \leq \mathcal{O}(\zeta)$, where $\zeta = \mathrm{sp}(V^*)\log(T\mathcal{N}_\mathcal{G}(\rho)/\delta)$ and we use the fact that $\|\mathbb{E}_{\zeta_i}[l_{f_i}(f^*, g, \zeta_i)]\|_2^2$ is non-negative. Recall the definition of $\rho$-cover, it ensures that for any $g \in \mathcal{G}$, there exists $\phi \in \mathcal{V}_\epsilon(\mathcal{G})$ such that $\|g(s, a) - \phi(s, a)\|_1 \leq \rho$ for all $(s, a) \in \mathcal{S} \times \mathcal{A}$. Therefore, for any $g \in \mathcal{G}$ we have

$$
-\sum_{i=1}^t X_{i,f_i}(g) \leq \mathcal{O}\Big(\zeta + t\rho\Big). \tag{D.5}
$$

Combine the (D.5) above and the designed confidence set, then for all $t \in [T]$ it holds that

$$
\mathcal{L}_{\mathcal{D}_{t-1}}(f^*, f^*) - \inf_{g \in \mathcal{G}} \mathcal{L}_{\mathcal{D}_{t-1}}(f^*, g) = -\sum_{i=1}^{t-1} X_{i,f_i}(\tilde{g}) \leq \mathcal{O}\Big(\zeta + t\rho\Big) < \beta, \tag{D.6}
$$

where $\tilde{g}$ is the local minimizer to $\mathcal{L}_{\mathcal{D}_{t-1}}(f^*, g)$, and we take the covering coefficient as $\rho = 1/T$ and optimistic parameter as $\beta = c \log\big(T\mathcal{N}_{\mathcal{H}\cup\mathcal{G}}^2(1/T)/\delta\big) \cdot \mathrm{sp}(V^*)$. Based on (D.6), with probability greater than $1 - \delta$, $f^*$ is a candidate of the confidence set such that $J_t \geq J^*$ for all $t \in [T]$.  □

## D.3  PROOF OF LEMMA 10

**Lemma 10.** For fixed $\rho > 0$ and the optimistic parameter $\beta = c\big(\mathrm{sp}(V^*) \cdot \log\big(T\mathcal{N}_{\mathcal{H}\cup\mathcal{G}}^2(\rho)/\delta\big) + T\rho\big)$ where $c > 0$ is constant large enough, then it holds that

$$
\sum_{i=1}^{t-1} \mathbb{E}_{\zeta_i} \|l_{f_i}(f_t, f_t, \zeta_i)\|^2 \leq \mathcal{O}(\beta), \tag{D.7}
$$

for all $t \in [T]$ with probability greater than $1 - \delta$.

*Proof of Lemma 10.* Denote $\mathcal{V}_\rho(\mathcal{H})$ be the $\rho$-cover of $\mathcal{H}$ and $\mathcal{N}_\mathcal{H}(\rho)$ be the size of $\rho$-cover $\mathcal{V}_\rho(\mathcal{H})$. Consider fixed $(i, f) \in [T] \times \mathcal{H}$ and define the auxiliary function

$$
X_{i,f_i}(f) := \big\|l_{f_i}(f, f, \zeta_i)\big\|_2^2 - \big\|l_{f_i}(f, \mathcal{P}(f), \zeta_i)\big\|_2^2,
$$

Let $\mathscr{F}_t$ be the filtration induced by $\{s_1, a_1, \ldots, s_t, a_t\}$ and note that $f_1, \ldots, f_t$ is fixed under the filtration, then we have

$$
\begin{aligned}
\mathbb{E}[X_{i,f_i}(f)|\mathscr{F}_i] &= \mathbb{E}_{\zeta_i}[\|l_{f_i}(f, f, \zeta_i)\|_2^2 - \|l_{f_i}(f, \mathcal{P}(f), \zeta_i)\|_2^2 |\mathscr{F}_i] \\
&= \mathbb{E}_{\zeta_i}\Big[ [l_{f_i}(f, f, \zeta_i) - l_{f_i}(f, \mathcal{P}(f), \zeta_i)] \cdot [l_{f_i}(f, f, \zeta_i) + l_{f_i}(f, \mathcal{P}(f), \zeta_i)]\Big|\mathscr{F}_i \Big] \\
&= \mathbb{E}_{\zeta_i}\big[l_{f_i}(f, f, \zeta_i)\big] \cdot \mathbb{E}_{\zeta_i}\big[l_{f_i}(f, f, \zeta_i) + l_{f_i}(f, \mathcal{P}(f), \zeta_i)|\mathscr{F}_i\big] \\
&= \big\|\mathbb{E}_{\zeta_i}\big[l_{f_i}(f, f, \zeta_i)\big]\big\|_2^2,
\end{aligned}
$$

where the equation generalized completeness (see Lemma 9). Similarly, we can obtain that the second moment of the auxiliary function is bounded by

$$
\mathbb{E}[X_{i,f_i}(f)^2|\mathscr{F}_i] \leq \mathcal{O}\Big(\mathrm{sp}(V^*)^2 \big\|\mathbb{E}_{\zeta_i}\big[l_{f_i}(f, f, \zeta_i)\big]\big\|_2^2\Big),
$$

By Freedman's inequality in Lemma 18, with probability greater than $1 - \delta$ we have

$$
\Big|\sum_{i=1}^{t} X_{i,f_i}(f) - \sum_{i=1}^{t} \big\|\mathbb{E}_{\zeta_i}\big[l_{f_i}(f, f, \zeta_i)\big]\big\|_2^2\Big|
$$

$$
\leq \mathcal{O}\left(\sqrt{\log(1/\delta) \cdot \mathrm{sp}(V^*)^2 \sum_{i=1}^{t}\big\|\mathbb{E}_{\zeta_i}\big[l_{f_i}(f, f, \zeta_i)\big]\big\|_2^2} + \log(1/\delta)\right).
$$

Define $\zeta = \mathrm{sp}(V^*)\log(T\mathcal{N}_{\mathcal{H}}(\rho)/\delta)$, by taking a union bound over $\rho$-covering of hypothesis set $\mathcal{H}$, we can obtain that with probability greater than $1 - \delta$, for all $(t, \phi) \in [T] \times \mathcal{V}_\rho(\mathcal{H})$ we have

$$
\Big|\sum_{i=1}^{t} X_{i,f_i}(\phi) - \sum_{i=1}^{t}\|\mathbb{E}_{\zeta_i}\big[l_{f_i}(\phi, \phi, \zeta_i)\big]\|_2^2\Big|
$$

$$
\leq \mathcal{O}\left(\sqrt{\zeta \cdot \mathrm{sp}(V^*)^2 \sum_{i=1}^{t}\|\mathbb{E}_{\zeta_i}\big[l_{f_i}(\phi, \phi, \zeta_i)\big]\|_2^2} + \zeta\right). \tag{D.8}
$$

The following analysis assumes that the event above is true. Recall that the LOOP ensures that

$$
\begin{aligned}
\sum_{i=1}^{t-1} X_{i,f_i}(f_t) &= \sum_{i=1}^{t-1}\|l_{f_i}(f_t, f_t, \zeta_i)\|_2^2 - \sum_{i=1}^{t-1}\|l_{f_i}(f_t, \mathcal{P}(f_t), \zeta_i)\|_2^2, \\
&\leq \sum_{i=1}^{t-1}\big\|l_{f_i}(f_t, f_t, \zeta_i)\big\|_2^2 - \inf_{g \in \mathcal{G}}\sum_{i=1}^{t-1}\big\|l_{f_i}(f_t, g, \zeta_i)\big\|_2^2, \\
&= \mathcal{L}_{\mathcal{D}_{t-1}}(f_t, f_t) - \inf_{g \in \mathcal{G}}\mathcal{L}_{\mathcal{D}_{t-1}}(f_t, g) \leq \mathcal{O}(\beta), \tag{D.9}
\end{aligned}
$$

where the last inequality is based on the confidence set and the update condition combined. Note that if the update is executed at time $t$, the confidence set ensures that

$$
\mathcal{L}_{\mathcal{D}_{t-1}}(f_t, f_t) - \inf_{g \in \mathcal{G}}\mathcal{L}_{\mathcal{D}_{t-1}}(f_t, g) \leq \beta,
$$

within the update step $t$. Otherwise, if the update condition is not triggered, we have $f_{\tau_t} = f_t$ and

$$
\Upsilon_{t-1} = \mathcal{L}_{\mathcal{D}_{t-1}}(f_t, f_t) - \inf_{g \in \mathcal{G}}\mathcal{L}_{\mathcal{D}_{t-1}}(f_t, g) \leq 4\beta.
$$

Recall that based on the definition of $\rho$-cover for any $f \in \mathcal{H}$, there exists $\phi \in \mathcal{V}_\rho(\mathcal{H})$ such that $\|g(s, a) - \phi(s, a)\|_1 \leq \rho$ for all $(s, a) \in \mathcal{S} \times \mathcal{A}$, we have the in-sample training error is bounded by

$$
\sum_{i=1}^{t-1}\big\|\mathbb{E}_{\zeta_i}\big[l_{f_i}(f_t, f_t, \zeta_i)\big]\big\|_2^2 \leq \sum_{i=1}^{t-1}\big\|\mathbb{E}_{\zeta_i}\big[l_{f_i}(\phi_t, \phi_t, \zeta_i)\big]\big\|_2^2 + \mathcal{O}(t\rho), \qquad (\rho\text{-approximation})
$$

$$
= \sum_{i=1}^{t-1} X_{i,f_i}(\phi_t) + \mathcal{O}(t\rho + \zeta) \qquad (\text{D.8})
$$

$$
= \sum_{i=1}^{t-1} X_{i,f_i}(f_t) + \mathcal{O}(t\rho + \zeta) \leq \mathcal{O}(T\rho + \zeta + \beta) = \mathcal{O}(\beta), \qquad (\text{D.10})
$$

where the last inequality follows (D.9), and takes $\beta = c\big((\mathrm{sp}(V^*)\log\big(T\mathcal{N}_{\mathcal{H}\cup\mathcal{G}}^2(\rho)/\delta\big) + T\rho\big)$. $\qquad\square$

### D.4 PROOF OF LEMMA 11

**Lemma 11.** Let $\mathcal{N}(T)$ be the switching cost with time horizon $T$, defined as

$$\mathcal{N}(T) = \#\{t \in [T] : \tau_t \neq \tau_{t-1}\}.$$

Given fixed $\rho > 0$ and the optimistic parameter $\beta = c\big(\mathrm{sp}(V^*) \log \big(T\mathcal{N}_{\mathcal{H}\cup\mathcal{G}}^2(\rho)/\delta\big) + T\rho\big)$, where $c > 0$ is large enough constant, then with probability greater than $1 - 2\delta$ we have

$$\mathcal{N}(T) \leq \mathcal{O}\big(\kappa_{\mathrm{G}} \log T + \beta^{-1} T\epsilon^2\big),$$

where $\kappa_{\mathrm{G}}$ is the transferability coefficient with respect to $\mathrm{AGEC}(\mathcal{H}, \{l_{f'}\}, \epsilon)$.

*Proof of Lemma 11.* Denote $\mathcal{V}_\rho(\mathcal{H})$ be the $\rho$-cover of $\mathcal{H}$ and $\mathcal{N}_{\mathcal{H}}(\rho)$ be the size of $\rho$-cover $\mathcal{V}_\rho(\mathcal{H})$.

**Step 1: Bound the difference of discrepancy between the minimizer and $\mathcal{P}(f)$.**

Consider fixed tuple $(i, f, g) \in [T] \times \mathcal{H} \times \mathcal{G}$ and define auxiliary function as

$$X_{i,f_i}(f,g) := \big\|l_{f_i}(f,g,\zeta_i)\big\|_2^2 - \big\|l_{f_i}(f,\mathcal{P}(f),\zeta_i)\big\|_2^2$$

Let $\mathscr{F}_t$ be the filtration induced by $\{s_1, a_1, \ldots, s_t, a_t\}$ and note that $f_1, \ldots, f_t$ is fixed under the filtration, then we have

$$
\begin{aligned}
\mathbb{E}[X_{i,f_i}(f,g)|\mathscr{F}_i] &= \mathbb{E}_{\zeta_i}[\big\|l_{f_i}(f,g,\zeta_i)\big\|_2^2 - \big\|l_{f_i}(f,\mathcal{P}(f),\zeta_i)\big\|_2^2|\mathscr{F}_i] \\
&= \mathbb{E}_{\zeta_i}\Big[\big[l_{f_i}(f,g,\zeta_i) - l_{f_i}(f,\mathcal{P}(f),\zeta_i)\big] \cdot \big[l_{f_i}(f,g,\zeta_i) + l_{f_i}(f,\mathcal{P}(f),\zeta_i)\big]\Big|\mathscr{F}_i\Big] \\
&= \mathbb{E}_{\zeta_i}\big[l_{f_i}(f,g,\zeta_i)\big] \cdot \mathbb{E}_{\zeta_i}\big[l_{f_i}(f,g,\zeta_i) + l_{f_i}(f,\mathcal{P}(f),\zeta_i)\big|\mathscr{F}_i\big] \\
&= \big\|\mathbb{E}_{\zeta_i}\big[l_{f_i}(f,g,\zeta_i)\big]\big\|_2^2,
\end{aligned}
$$

where the equation generalized completeness (see Lemma 9). Similarly, we can obtain that the second moment of the auxiliary function is bounded by

$$\mathbb{E}[X_{i,f_i}(f,g)^2|\mathscr{F}_i] \leq \mathcal{O}\Big(\mathrm{sp}(V^*)^2 \big\|\mathbb{E}_{\zeta_i}\big[l_{f_i}(f,g,\zeta_i)\big]\big\|_2^2\Big),$$

By Freedman's inequality in Lemma 18, with probability greater than $1 - \delta$

$$
\Big| \sum_{i=1}^t X_{i,f_i}(f,g) - \sum_{i=1}^t \big\|\mathbb{E}_{\zeta_i}\big[l_{f_i}(f,g,\zeta_i)\big]\big\|_2^2 \Big|
$$

$$
\leq \mathcal{O}\left( \sqrt{\log(1/\delta) \cdot \mathrm{sp}(V^*)^2 \sum_{i=1}^t \big\|\mathbb{E}_{\zeta_i}\big[l_{f_i}(f,g,\zeta_i)\big]\big\|_2^2} + \log(1/\delta)\right)
$$

Define $\zeta = \mathrm{sp}(V^*)\log(T\mathcal{N}_{\mathcal{H}\cup\mathcal{G}}^2(\rho)/\delta)$, by taking a union bound over $\rho$-covering of hypothesis set $\mathcal{H} \times \mathcal{G}$, with probability greater than $1 - \delta$, for all $(t, \phi, \varphi) \in [T] \times \mathcal{V}_\rho(\mathcal{H}) \times \mathcal{V}_\rho(\mathcal{G})$ it holds

$$
\Big| \sum_{i=1}^t X_{i,f_i}(\phi,\psi) - \sum_{i=1}^t \big\|\mathbb{E}_{\zeta_i}\big[l_{f_i}(\phi,\psi,\zeta_i)\big]\big\|_2^2 \Big|
$$

$$
\leq \mathcal{O}\left( \sqrt{\zeta \cdot \mathrm{sp}(V^*)^2 \sum_{i=1}^t \big\|\mathbb{E}_{\zeta_i}\big[l_{f_i}(\phi,\psi,\zeta_i)\big]\big\|_2^2} + \zeta \right), \tag{D.11}
$$

where $\zeta = \mathrm{sp}(V^*)\log(T\mathcal{N}_{\mathcal{H}\cup\mathcal{G}}^2(\rho)/\delta)$. Note that $\big\|\mathbb{E}_{\zeta_i}\big[l_{f_i}(\phi,\psi,\zeta_i)\big]\big\|_2^2$ is non-negative, then it holds that $-\sum_{i=1}^t X_{i,f_i}(\phi,\varphi) \leq \mathcal{O}(\zeta)$ for all $t \in [T]$. Based on (D.11) and the $\rho$-approximation, we have

$$
-\sum_{i=1}^t X_{i,f_i}(f,g) \leq \mathcal{O}\big(\zeta + t\rho\big), \qquad \forall t \in [T],
$$

for any $(f, g) \in \mathcal{H} \times \mathcal{G}$. Recall that $\beta = c \log(T \mathcal{N}^2_{\mathcal{H} \cup \mathcal{G}}(\rho)/\delta) \mathrm{sp}(V^*)$, for all $t \in [T]$ we have

$$
\mathcal{L}_{\mathcal{D}_t}(f_t, \mathcal{P}(f_t)) - \inf_{g \in \mathcal{G}} \mathcal{L}_{\mathcal{D}_t}(f_t, g) = \sum_{i=1}^{t} \|l_{f_i}(f_t, \mathcal{P}(f_t), \zeta_i)\|_2^2 - \inf_{g \in \mathcal{G}} \sum_{i=1}^{t} \|l_{f_i}(f_t, g, \zeta_i)\|_2^2
$$

$$
= -\sum_{i=1}^{t} X_{i,f_i}(f_t, \tilde{g}) \leq \mathcal{O}\big(\zeta + t\rho\big) \leq \beta. \tag{D.12}
$$

Combine (D.12), and the fact that $g$ is defined as the local minimizer among auxiliary class $\mathcal{G}$ and $\mathcal{P}(f_t) \in \mathcal{G}$, then for all $t \in [T]$ we have the difference of discrepancy bounded by

$$
0 \leq \mathcal{L}_{\mathcal{D}_t}(f_t, \mathcal{P}_{J_t}(f_t)) - \inf_{g \in \mathcal{G}} \mathcal{L}_{\mathcal{D}_t}(f_t, g) \leq \beta. \tag{D.13}
$$

**Step 2: Bound the out-sample training error between updates.**

Consider an update is executed at step $t+1$, it directly implies that $\mathcal{L}_{\mathcal{D}_t}(f_t, f_t) - \inf_{g \in \mathcal{G}} \mathcal{L}_{\mathcal{D}_t}(f_t, g) > 4\beta$, while the latest update at step $\tau_t$ ensures that $\mathcal{L}_{\mathcal{D}_{\tau_t - 1}}(f_{\tau_t}, f_{\tau_t}) - \inf_{g \in \mathcal{G}} \mathcal{L}_{\mathcal{D}_{\tau_t - 1}}(f_{\tau_t}, g) \leq \beta$, where $\tau_t$ is the pointer of the lastest update. Combined the results above with (D.13), we have

$$
\mathcal{L}_{\mathcal{D}_t}(f_t, f_t) - \mathcal{L}_{\mathcal{D}_t}\big(f_t, \mathcal{P}(f_t)\big) > 3\beta, \quad \mathcal{L}_{\mathcal{D}_{\tau_t - 1}}(f_{\tau_t}, f_{\tau_t}) - \mathcal{L}_{\mathcal{D}_{\tau_t - 1}}\big(f_{\tau_t}, \mathcal{P}(f_{\tau_t})\big) \leq \beta. \tag{D.14}
$$

It indicates that the sum of squared empirical discrepancy between two adjacent updates follows

$$
\sum_{i=\tau_t}^{t} \|l_{f_t}(f_t, f_t, \zeta_t)\|_2^2 = \mathcal{L}_{\mathcal{D}_{\tau_t:t}}(f_t, f_t) - \mathcal{L}_{\mathcal{D}_{\tau_t:t}}(f_t, \mathcal{P}_{J_t}(f_t)) > 2\beta, \tag{D.15}
$$

where denote $\mathcal{D}_{\tau_t:t} = \mathcal{D}_t / \mathcal{D}_{\tau_t}$. Based on the similar concentration arguments as Lemma 10, we have the out-sample training error between updates is bounded by $\sum_{i=\tau_t}^{t} \|\mathbb{E}_{\zeta_i}[l_{f_i}(f_i, f_i, \zeta_i)]\|_2^2 > \beta$.

**Step 3: Bound the switching cost under the transferability constraint.**

Denote $b_1, \ldots, b_{\mathcal{N}(T)}, b_{\mathcal{N}(T)+1}$ be the sequence of updated steps such that $\tau_t \in \{b_t\}$ for all $t \in [T]$, and we fix the recorder $b_1 = 1$ and $b_{\mathcal{N}(T)+1} = T + 1$. Note that based on (D.15), the sum of out-sample training error shall have a lower bound such that

$$
\sum_{t=1}^{T} \|\mathbb{E}_{\zeta_t}[l_{f_t}(f_t, f_t, \zeta_t)]\|_2^2 = \sum_{u=1}^{\mathcal{N}(T)} \sum_{t=b_u}^{b_{u+1}-1} \|\mathbb{E}_{\zeta_t}[l_{f_t}(f_t, f_t, \zeta_t)]\|_2^2 \geq \mathcal{N}(T) \cdot \beta. \tag{D.16}
$$

Besides, note that the in-sample training error $\sum_{i=1}^{t-1} \|\mathbb{E}_{\zeta_t}[l_{f_i}(f_t, f_t, \zeta_t)]\|_2^2 \leq \mathcal{O}\big(\beta\big)$ for all $t \in [T]$ and based on the definition of transferability coefficient $\kappa_{\mathrm{G}}$ (see Definition 3), we have

$$
\sum_{t=1}^{T} \|\mathbb{E}_{\zeta_t}[l_{f_t}(f_t, f_t, \zeta_t)]\|_2^2 \leq \mathcal{O}\left(\kappa_{\mathrm{G}} \cdot \beta \log T + \mathrm{sp}(V^*)^2 \min\{\kappa_{\mathrm{G}}, T\} + T\epsilon^2\right) \tag{D.17}
$$

Combine (D.16) and (D.17), it holds $\mathcal{N}(T) \leq \mathcal{O}\big(\kappa_{\mathrm{G}} \log T + \beta^{-1} T \log T \epsilon^2\big)$ and finish the proof. $\square$

# E  PROOF OF RESULTS ABOUT COMPLEXITY MEASURES

In this section, we provide the proof of results about the complexity metrics in Section 3. We remark that the proof highly relies on Lemma 13 and Lemma 14, which are natural extentions to original results in Jin et al. (2020); Zhong et al. (2022) and proofs are provided in Section G.1.

## E.1  PROOF OF LEMMA 1

*Proof of Lemma 1.* Recall that the eluder dimension is defined over the function class following

$$
\mathcal{X}_{\mathcal{H}} := \big\{ X_{f,f'}(s, a) = \big(r_f + \mathbb{P}_{f'} V_f\big)(s, a) : f, f' \in \mathcal{H} \big\},
$$

and for model-based problems, the discrepancy function is chosen as

$$
l_{f'}(f, g, \zeta_t) = \big(r_g + \mathbb{P}_g V_{f'}\big)(s_t, a_t) - r(s_t, a_t) - V_{f'}(s_{t+1}).
$$

**Step 1: Bound over transferability coefficient.**

Start with the transferability coefficient, the condition can be equivalently written as

$$\sum_{i=1}^{t-1} \|\mathbb{E}_{\zeta_i}[l_{f_i}(f_t, f_t, \zeta_i)]\|_2^2 = \sum_{i=1}^{t-1} \|(r_{f_t} + \mathbb{P}_{f_t} V_{f_t} - r_{f^*} + \mathbb{P}_{f^*} V_{f_t})(s_i, a_i)\|_2^2$$

$$= \sum_{i=1}^{t-1} (X_{f_t, f_t} - X_{f_t, f^*})(s_i, a_i)^2 \le \beta, \quad \forall t \in [T]. \tag{E.1}$$

Let $\breve{\mathcal{X}}_{\mathcal{H}} = \{f - f' : f, f' \in \mathcal{X}_{\mathcal{H}}\}$, the generalized pigeon-hole principle (see Lemma 13) indicates that if we take $\Gamma = \mathcal{D}_\Delta$, $\phi_t = X_{f_t, f_t} - X_{f_t, f^*}$, $\Phi = \breve{\mathcal{X}}_{\mathcal{H}}$, $\|\phi_t\|_\infty \le \mathrm{sp}(V^*) + 2$, then it holds that

$$\sum_{i=1}^{t} \|\mathbb{E}_{\zeta_i}[l_{f_i}(f_i, f_i, \zeta_i)]\|^2 = \sum_{i=1}^{t} (X_{f_i, f_i} - X_{f_i, f^*})(s_i, a_i)^2$$

$$\le \dim_{\mathrm{DE}}(\breve{\mathcal{X}}_{\mathcal{H}}, \mathcal{D}_\Delta, \epsilon) \cdot \beta \log t + (\mathrm{sp}(V^*) + 2)^2 \min\{\dim_{\mathrm{DE}}(\breve{\mathcal{X}}_{\mathcal{H}}, \mathcal{D}_\Delta, \epsilon), t\} + t\epsilon^2$$

$$= \dim_{\mathrm{E}}(\mathcal{X}_{\mathcal{H}}, \epsilon) \cdot \beta \log t + (\mathrm{sp}(V^*) + 2)^2 \min\{\dim_{\mathrm{E}}(\mathcal{X}_{\mathcal{H}}, \epsilon), t\} + t\epsilon^2, \tag{E.2}$$

given condition that (E.1) holds for all $t \in [T]$, where the last equation uses $\dim_{\mathrm{DE}}(\breve{\mathcal{X}}_{\mathcal{H}}, \mathcal{D}_\Delta, \epsilon) = \dim_{\mathrm{E}}(\mathcal{X}_{\mathcal{H}}, \epsilon)$. Denote $d_{\mathrm{E}} = \dim_{\mathrm{E}}(\mathcal{X}_{\mathcal{H}}, \epsilon)$, then we have $\kappa_{\mathrm{G}} \le d_{\mathrm{E}}$ based on (E.2).

**Step 2: Bound over dominance coefficient.**

Based on Lemma 14 and $\mathcal{E}(f_t)(s_t, a_t) = \mathbb{E}_{\zeta_t}[l_{f_t}(f_t, f_t, \zeta_t)]$ based on definition, it holds that

$$\sum_{t=1}^{T} \|\mathbb{E}_{\zeta_t}[l_{f_t}(f_t, f_t, \zeta_t)]\|_2^2 = \sum_{t=1}^{T} [(X_{f_t, f_t} - X_{f_t, f^*})(s_t, a_t)]^2$$

$$\le \left[2d_{\mathrm{DE}} \log T \sum_{t=1}^{T} \sum_{i=1}^{t-1} [(X_{f_t, f_t} - X_{f_t, f^*})(s_i, a_i)]^2\right]^{1/2} + (\mathrm{sp}(V^*) + 2) \min\{d_{\mathrm{DE}}, T\} + T\epsilon$$

$$= \left[2d_{\mathrm{E}} \log T \sum_{t=1}^{T} \sum_{i=1}^{t-1} \|\mathbb{E}_{\zeta_i}[l_{f_i}(f_t, f_t, \zeta_i)]\|_2^2\right]^{1/2} + (\mathrm{sp}(V^*) + 2) \min\{d_{\mathrm{E}}, T\} + T\epsilon, \tag{E.3}$$

by taking $\Gamma = \mathcal{D}_\Delta$, $\phi_t = X_{f_t, f_t} - X_{f_t, f^*}$, $\Phi = \breve{\mathcal{X}}_{\mathcal{H}}$, $\|\phi_t\|_\infty \le \mathrm{sp}(V^*) + 2$, and $1 + \log T \le 2 \log T$. $\square$

## E.2 PROOF OF LEMMA 2

*Proof of Lemma 2.* Consider the Bellman discrepancy function, defined as

$$l_{f'}(f, g, \zeta_t) = Q_g(s_t, a_t) - r(s_t, a_t) - V_f(s_{t+1}) + J_g,$$

and the expectation is taken over $s_{i+1}$ from $\mathbb{P}(\cdot|s_i, a_i)$ such that $\mathbb{E}_{\zeta_i}[l_{f_i}(f_t, f_t, \zeta_i)] = \mathcal{E}(f_t)(s_i, a_i)$.

**Step 1: Bound over transferability coefficient.**

First, we're going to demonstrate the transferability. Note that the generalized pigeon-hole principle (see Lemma 13) directly indicates that, given

$$\sum_{i=1}^{t-1} \|\mathcal{E}(f_t)(s_i, a_i)\|_2^2 = \sum_{i=1}^{t-1} \|\mathbb{E}_{\zeta_i}[l_{f_i}(f_t, f_t, \zeta_i)]\|_2^2 \le \beta, \quad \forall t \in [T],$$

if we take $\phi_t = \mathcal{E}(f_t)$, $\Phi = \mathcal{E}_{\mathcal{H}}$ and $\Gamma = \mathcal{D}_\Delta$, then for all $t \in [T]$ we have

$$\sum_{i=1}^{t} \|\mathcal{E}(f_i)(s_i, a_i)\|_2^2 \le d_{\mathrm{ABE}} \cdot \beta \log t + (\mathrm{sp}(V^*) + 2)^2 \min\{d_{\mathrm{ABE}}, t\} + t\epsilon^2, \tag{E.4}$$

and thus we upper bound $\kappa_{\mathrm{G}} \le d_{\mathrm{ABE}} := \dim_{\mathrm{ABE}}(\mathcal{H}, \epsilon)$.

**Step 2: Bound over dominance coefficient.**

Based on Lemma 14 and $\mathbb{E}_{\zeta_i}[l_{f_i}(f_t, f_t, \zeta_i)] = \mathcal{E}(f_t)(s_i, a_i)$, it holds that

$$\sum_{t=1}^{T} \mathcal{E}(f_t)(s_t, a_t) \leq \left[2d_{\text{ABE}} \log T \sum_{t=1}^{T} \sum_{i=1}^{t-1} \|\mathcal{E}(f_t)(s_i, a_i)\|_2^2\right]^{1/2} + (\text{sp}(V^*) + 2) \min\{d_{\text{ABE}}, T\} + T\epsilon,$$

where we take $\phi_t = \mathcal{E}(f_t)$, $\Phi = \mathcal{E}_{\mathcal{H}}$, $\Gamma = \mathcal{D}_{\Delta}$, $\|\mathcal{E}(f_t)\|_{\infty} \leq \text{sp}(V^*) + 2$, and $1 + \log T \leq 2 \log T$. □

## F PROOF OF RESULTS FOR CONCRETE EXAMPLES

In this section, we provide detailed proofs of results for concrete examples in Appendix C.

### F.1 PROOF OF PROPOSITION 5

*Proof of Proposition 5.* To show that linear FA has low AGEC, we first prove that it is captured by ABE dimension $\dim_{\text{ABE}}(\mathcal{H}, \epsilon)$, and then apply Lemma 2 (low ABE dim $\subseteq$ low AGEC). Suppose that there exists $\epsilon' \geq \epsilon$, $\{\delta_{s_i, a_i}\}_{i \in [m]} \subseteq \mathcal{D}_{\Delta}$, and $\{f_i\}_{i \in [m]} \subseteq \mathcal{H}$ with length $m \in \mathbb{N}$, such that

$$\sqrt{\sum_{i=1}^{t-1} \left[\mathcal{E}(f_t)(s_i, a_i)\right]^2} \leq \epsilon', \quad \left|\mathcal{E}(f_t)(s_t, a_t)\right| > \epsilon', \quad \forall t \in [m]. \tag{F.1}$$

Based on Definitions 7-8, $\dim_{\text{ABE}}(\mathcal{H}, \epsilon)$ is the largest $m$. Following this, we provide detailed discussion about linear AMDPs, AMDPs with linear Bellmen completion, and AMDPs with generalized linear completion (see Definition 11-13). Denote $Q_t(s, a) = \phi(s, a)^{\top} \omega_t$ for all $(s, a, t) \in \mathcal{S} \times \mathcal{A} \times [m]$.

**(i). Linear AMDPs.** As defined in Definition 11, for any $f_t \in \mathcal{H}$, it holds that

$$\mathcal{E}(f_t)(s_i, a_i) = \phi(s_i, a_i)^{\top} \omega_t - \phi(s_i, a_i)^{\top} \theta - \phi(s_i, a_i)^{\top} \int_S V_t(s) \mathrm{d}\mu(s) + J_t$$

$$= \phi(s_i, a_i)^{\top} \left(\omega_t - \theta + \int_S V_t(s) \mathrm{d}\mu(s) + J_t \mathbf{e}_1\right). \tag{F.2}$$

**(ii). AMDPs with linear Bellmen completion.** As a natural extension to the linear AMDPs, linear Bellmen completeness (see Definition 12) suggests that the Bellman error follows

$$\mathcal{E}(f_t)(s_i, a_i) = \phi(s_i, a_i)^{\top} \omega_t - \phi(s_i, a_i)^{\top} \mathcal{T}(\omega_t, J_t) = \phi(s_i, a_i)^{\top} (\omega_t - \mathcal{T}(\omega_t, J_t)). \tag{F.3}$$

**(iii). AMDPs with generalized linear completion.** Moreover, AMDPs with generalized linear completion further extends the standard linear FA by introducing link functions. Note that

$$\mathcal{E}(f_t)(s_i, a_i) = \sigma\big(\phi(s_i, a_i)^{\top} \omega_t\big) - \sigma\big(\phi(s_i, a_i)^{\top} \mathcal{T}(\omega_t, J_t)\big) \tag{F.4}$$

and based on the $\alpha$-bi-Lipschitz continuity condition in (C.3), it holds that

$$\frac{1}{\alpha} \cdot \phi(s_i, a_i)^{\top} (\omega_t - \mathcal{T}(\omega_t, J_t)) \leq \mathcal{E}(f_t)(s_i, a_i) \leq \alpha \cdot \phi(s_i, a_i)^{\top} (\omega_t - \mathcal{T}(\omega_t, J_t)). \tag{F.5}$$

Based on the Lemma 15 and arguments in (i), (ii) and(iii), we are ready to provide a unified proof for linear FA. By substituting the arguments (F.2), (F.3) and (F.5) into (F.1), then

$$\sqrt{\sum_{i=1}^{t-1} [\langle\phi(s_i, a_i), \omega_t - \mathcal{T}(\omega_t, J_t)\rangle]^2} \leq \alpha\epsilon', \quad |\langle\phi(s_t, a_t), \omega_t - \mathcal{T}(\omega_t, J_t)\rangle| > \frac{\epsilon'}{\alpha}, \quad \forall t \in [m].$$
$$\tag{F.6}$$

Here, we take $\alpha = 1$ for standard linear FA , and let $\alpha$ be the Lipschitz constant for generalized linear FA. Based on Lemma 15, if we take $\phi_t = \phi(s_t, a_t)$, $\psi_t = \omega_t - \mathcal{T}(\omega_t, J_t)$, $B_{\phi} = \sqrt{2}$, $B_{\psi} = \text{sp}(V^*)\sqrt{d}$, $\varepsilon = \epsilon$, $c_1 = \alpha$, $c_2 = \alpha^{-1}$, then $m \leq \mathcal{O}\big(d \log(\text{sp}(V^*)\sqrt{d}/\epsilon)\big)$. As the ABE dimension is defined as the length of the longest sequence satisfying (F.6), thus

$$\dim_{\text{ABE}}(\mathcal{H}, \epsilon) \leq \mathcal{O}\big(d \log\big(\text{sp}(V^*)\sqrt{d}/\epsilon\big)\big).$$

Based on Lemma 2, $d_{\text{G}} \leq \mathcal{O}\big(d \log\big(\text{sp}(V^*)\sqrt{d}\epsilon^{-1}\big) \log T\big)$ and $\kappa_{\text{G}} \leq \mathcal{O}\big(d \log\big(\text{sp}(V^*)\sqrt{d}\epsilon^{-1}\big)\big)$. □

### F.2 PROOF OF PROPOSITION 6

*Proof of Proposition 6.* For all $f_t \in \mathcal{H}$, the Bellman error can be written as

$$
\begin{aligned}
\mathcal{E}(f_t)(s_i, a_i) &= \phi(s_i, a_i)^\top \omega_t - \mathbb{E}[\psi(s_{i+1})]^\top \theta_t + J_t - r(s_i, a_i) \\
&= \phi(s_i, a_i)^\top \omega_t - \mathbb{E}[\psi(s_{i+1})]^\top \theta_t + J_t - (Q^*(s_i, a_i) - \mathbb{E}[V^*(s_{i+1})] - J^*) \\
&= \begin{bmatrix} \phi(s_i, a_i) \\ \mathbb{E}[\psi(s_{i+1})] \end{bmatrix}^\top \left( \begin{bmatrix} \omega_t - \omega^* \\ \theta^* - \theta_t \end{bmatrix} + (J_t - J^*) \cdot \mathbf{e}_1 \right),
\end{aligned}
\tag{F.7}
$$

where the second equation results from Bellman optimality equation in (2.1). Following a similar argument in the proof of Proposition 5, we can show that the linear $Q^*/V^*$ AMDPs have a low ABE dimension with an $(d_1 + d_2)$-dimensional compound feature mapping equivalently based on (F.7). Based on Lemma 2 and write $d^+ = d_1 + d_2$, then we have

$$
d_{\mathrm{G}} \le \mathcal{O}\big(d^+ \log \big(\mathrm{sp}(V^*)\sqrt{d^+}\epsilon^{-1}\big) \log T\big), \quad \kappa_{\mathrm{G}} \le \mathcal{O}\big(d^+ \log \big(\mathrm{sp}(V^*)\sqrt{d^+}\epsilon^{-1}\big)\big). \qquad \square
$$

### F.3 PROOF OF PROPOSITION 7

*Proof of Proposition 7.* Similar to linear FA, we will show that kernel FA is captured by by ABE dimension $\dim_{\mathrm{ABE}}(\mathcal{H}, \epsilon)$, and then apply Lemma 2 (low ABE dim $\subseteq$ low AGEC). Suppose that there exists $\epsilon' \ge \epsilon$, $\{\delta_{s_i, a_i}\}_{i \in [m]} \subseteq \mathcal{D}_\Delta$, and $\{f_i\}_{i \in [m]} \subseteq \mathcal{H}$ with length $m \in \mathbb{N}$, such that

$$
\sqrt{\sum_{i=1}^{t-1} \big[\mathcal{E}(f_t)(s_i, a_i)\big]^2} \le \epsilon', \quad \big|\mathcal{E}(f_t)(s_t, a_t)\big| > \epsilon', \quad \forall t \in [m].
\tag{F.8}
$$

Suppose the kernel function class has a finite $\epsilon$-effective dimension concerning the feature mapping $\phi$. The existence of Bellman error $\mathcal{E}(f_t)$ is equivalent to the one of $W_t \in (\mathcal{W} - \mathcal{W})$:

$$
\mathcal{E}(f_t)(\cdot, \cdot) = (Q_{f_t} - \mathcal{T}_{J_t} Q_{f_t})(\cdot, \cdot) = \langle \phi(\cdot, \cdot), \omega_t - \omega_t' \rangle_\mathcal{K} := \langle \phi(\cdot, \cdot), W_t \rangle_\mathcal{K},
\tag{F.9}
$$

where the second equation is based on the self-completeness assumption with kernel FA such that $\mathcal{G} = \mathcal{H}$. Denote $X_t = \phi(s_t, a_t)$, we can rewrite the condition in (F.8) as

$$
\sqrt{\sum_{i=1}^{t-1} (X_i^\top W_t)^2} \le \epsilon', \quad |X_t^\top W_t| > \epsilon', \quad \forall t \in [m].
\tag{F.10}
$$

Let $\Sigma_t = \sum_{i=1}^{t-1} X_i X_i^\top + (\epsilon'^2/4R^2 \cdot \mathrm{sp}(V^*)^2) \cdot \mathbf{I}$, then $\|W_t\|_{\Sigma_t} \le \sqrt{2}\epsilon'$ and $\epsilon' \le \|W_t\|_{\Sigma_t} \|X_t\|_{\Sigma_t^{-1}}$ for all $t \in [m]$ based on Cauchy-Swartz inequlity and $\|\omega_t\|_\mathcal{K} \le \mathrm{sp}(V^*)R$. Thus, $\|X_t\|_{\Sigma_t^{-1}}^2 \ge 0.5$ and

$$
\sum_{t=1}^m \log \left(1 + \|X_t\|_{\Sigma_t^{-1}}^2\right) = \log \left(\frac{\det \Sigma_{m+1}}{\det \Sigma_1}\right) = \log \det \left[\mathbf{I} + \frac{4R^2 \mathrm{sp}(V^*)^2}{\epsilon'^2} \sum_{t=1}^m X_t X_t^\top\right], \quad \text{(F.11)}
$$

based on the matrix determinant lemma. Therefore, (F.10) directly implies that

$$
\frac{1}{e} \le \log \frac{3}{2} \le \frac{1}{m} \log \det \left[\mathbf{I} + \frac{4R^2 \mathrm{sp}(V^*)^2}{\epsilon'^2} \sum_{t=1}^m X_t X_t^\top\right],
$$

and then we have $m \le \dim_{\mathrm{eff}}\big(\mathcal{X}, \epsilon/2\mathrm{sp}(V^*)R\big)$. Recall that the $\epsilon$-effective dimension is the minimum positive integer satisfying the condition. As ABE dimension is defined as the length of the longest sequence satisfying (F.10), thus it holds that

$$
\dim_{\mathrm{ABE}}(\mathcal{H}, \epsilon) \le \dim_{\mathrm{eff}}\big(\mathcal{X}, \epsilon/2\mathrm{sp}(V^*)R\big).
$$

Based on Lemma 2, $d_{\mathrm{G}} \le \dim_{\mathrm{eff}}\big(\mathcal{X}, \epsilon/2\mathrm{sp}(V^*)R\big) \log T$ and $\kappa_{\mathrm{G}} \le \dim_{\mathrm{eff}}\big(\mathcal{X}, \epsilon/2\mathrm{sp}(V^*)R\big)$. $\quad \square$

### F.4 PROOF OF PROPOSITION 8

*Proof of Proposition 8.* Note that expected discrepancy function follows: for any $t \in [T]$

$$\|\mathbb{E}_{\zeta_i}[l_{f_i}(f_t, f_t, \zeta_i)]\|_2 = \theta_t^\top \left( \psi(s_i, a_i) + \int_{\mathcal{S}} \phi(s_i, a_i, s') V_{f_i}(s') \mathrm{d}s' \right) - r(s_i, a_i) - \mathbb{E}_{\zeta_i}[V_{f_i}(s_{i+1})]$$

$$= (\theta_t - \theta^*)^\top \left( \psi(s_i, a_i) + \int_{\mathcal{S}} \phi(s_i, a_i, s') V_{f_i}(s') \mathrm{d}s' \right). \tag{F.12}$$

Let $W_t = \theta_t - \theta^*$, $X_t = \psi(s_i, a_i) + \int_{\mathcal{S}} \phi(s_i, a_i, s') V_{f_i}(s') \mathrm{d}s'$, and $\Sigma_t = \epsilon \mathbf{I} + \sum_{i=1}^{t-1} X_t X_t^\top$. Note

$$\|W_t\|_{\Sigma_t} = \left[ \epsilon\|\theta_t - \theta^*\|_2^2 + \sum_{i=1}^{t-1} \|\mathbb{E}_{\zeta_i}[l_{f_i}(f_t, f_t, \zeta_i)]\|_2^2 \right]^{1/2} \leq 2\sqrt{\epsilon} + \left[ \sum_{i=1}^{t-1} \|\mathbb{E}_{\zeta_i}[l_{f_i}(f_t, f_t, \zeta_i)]\|_2^2 \right]^{1/2} \tag{F.13}$$

where we use $\|\theta_t\| \leq 1$. Based on the elliptical potential lemma (see Lemma 17), we have

$$\sum_{t=1}^{T} \|X_t\|_{\Sigma_t^{-1}} \wedge 1 \leq \sum_{t=1}^{T} 2d \cdot \log\left( 1 + \frac{1}{d} \sum_{t=1}^{T} \|X_t\|_2 \right)$$

$$\leq 2d \cdot \log\left( 1 + (1 + \mathrm{sp}(V^*)/2) \cdot T \left( \sqrt{d}\epsilon \right)^{-1} \right) := d(\epsilon), \tag{F.14}$$

where the last inequality results from $\|\phi\|_{2,\infty} \leq \sqrt{d}$, $\|\psi\|_{2,\infty} \leq \sqrt{d}$ and $\|V_f\|_\infty \leq \frac{1}{2}\mathrm{sp}(V^*)$. Combine (F.13) and $\mathbb{1}\left( \|X_t\|_{\Sigma_t^{-1}} \geq 1 \right) \leq \|X_t\|_{\Sigma_t^{-1}} \wedge 1$, it holds that

$$\sum_{t=1}^{T} \mathbb{1}\left( \|X_t\|_{\Sigma_t^{-1}} \geq 1 \right) \leq \sum_{t=1}^{T} \|X_t\|_{\Sigma_t^{-1}} \wedge 1 \leq d(\epsilon). \tag{F.15}$$

**Step 1: Bound over dominance coefficient.**

Note the sum of Bellman errors follows that

$$\sum_{t=1}^{T} \mathcal{E}(f_t)(s_t, a_t) = \sum_{t=1}^{T} \left( (r_{f_t} + \mathbb{P}_{f_t} V_{f_t})(s_t, a_t) - (r_{f^*} + \mathbb{P}_{f^*} V_{f_t})(s_t, a_t) \right)$$

$$= \sum_{t=1}^{T} (\theta_t - \theta^*)^\top \left( \psi(s_t, a_t) + \int_{\mathcal{S}} \phi(s_t, a_t, s') V_{f_t}(s') \mathrm{d}s' \right)$$

$$= \sum_{t=1}^{T} W_t^\top X_t \cdot \left( \mathbb{1}\left( \|X_t\|_{\Sigma_t^{-1}} \leq 1 \right) + \mathbb{1}\left( \|X_t\|_{\Sigma_t^{-1}} > 1 \right) \right)$$

$$\leq \sum_{t=1}^{T} W_t^\top X_t \cdot \mathbb{1}\left( \|X_t\|_{\Sigma_t^{-1}} \leq 1 \right) + (\mathrm{sp}(V^*) + 2) \cdot \min\{d(\epsilon), T\}$$

$$\leq \sum_{t=1}^{T} \|W_t\|_{\Sigma_t} \cdot \left( \|X_t\|_{\Sigma_t^{-1}} \wedge 1 \right) + (\mathrm{sp}(V^*) + 2) \cdot \min\{d(\epsilon), T\}, \tag{F.16}$$

where the first inequality results from (F.15) and the last inequality arises from the Cauchy-Swartz inequality. Combine (F.13) and (F.14), we have

$$\sum_{t=1}^{T} \|W_t\|_{\Sigma_t} \cdot \left( \|X_t\|_{\Sigma_t^{-1}} \wedge 1 \right) \leq \sum_{t=1}^{T} \left( 2\sqrt{\epsilon} + \left[ \sum_{i=1}^{t-1} \|l_{f_i}(f_t, f_t, \zeta_i)\|_2^2 \right]^{1/2} \right) \cdot \left( \|X_t\|_{\Sigma_t^{-1}} \wedge 1 \right)$$

$$\leq \left[ \sum_{t=1}^{T} 4\epsilon \right]^{1/2} \left[ \sum_{t=1}^{T} \|X_t\|_{\Sigma_t^{-1}} \wedge 1 \right]^{1/2} + \left[ \sum_{t=1}^{T} \sum_{i=1}^{t-1} \|l_{f_i}(f_t, f_t, \zeta_i)\|_2^2 \right]^{1/2} \left[ \sum_{t=1}^{T} \|X_t\|_{\Sigma_t^{-1}} \wedge 1 \right]^{1/2}$$

$$\leq 2\sqrt{T\epsilon \cdot \min\{d(\epsilon), T\}} + \left[ d(\epsilon) \sum_{t=1}^{T} \sum_{i=1}^{t-1} \|l_{f_i}(f_t, f_t, \zeta_i)\|_2^2 \right]^{1/2}, \tag{F.17}$$

where the second inequality results from Cauchy-Swartz inequality and the last inequality follows (F.14). Plugging the result back into the (F.16), we conclude that

$$\sum_{t=1}^{T} \mathcal{E}(f_t)(s_t, a_t) \leq \left[ d(\epsilon) \sum_{t=1}^{T} \sum_{i=1}^{t-1} \|\mathbb{E}_{\zeta_i}[l_{f_i}(f_t, f_t, \zeta_i)]\|_2^2 \right]^{1/2}$$
$$+ 2\sqrt{T\epsilon \cdot \min\{d(\epsilon), T\}} + (\mathrm{sp}(V^*) + 2) \min\{d(\epsilon), T\}$$
$$\leq \left[ d(\epsilon) \sum_{t=1}^{T} \sum_{i=1}^{t-1} \|\mathbb{E}_{\zeta_i}[l_{f_i}(f_t, f_t, \zeta_i)]\|_2^2 \right]^{1/2} + (\mathrm{sp}(V^*) + 3) \min\{d(\epsilon), T\} + T\epsilon,$$

$$(F.18)$$

where the last inequality follows AM-GM inequality. Thus, $d_{\mathrm{G}} \leq \mathcal{O}(d \log(\mathrm{sp}(V^*)T/\sqrt{d}\epsilon))$.

**Step 2: Bound over transferability coefficient.**

Given condition that $\sum_{i=1}^{t-1} \|\mathbb{E}[l_{f_i}(f_t, f_t, \zeta_i)]\|_2^2 \leq \beta$ for all $t \in [T]$, we have

$$\sum_{t=1}^{T} \|\mathbb{E}[l_{f_t}(f_t, f_t, \zeta_t)]\|_2^2 = \sum_{t=1}^{T} \left[ (\theta_i - \theta^*)^\top \left( \psi(s_t, a_t) + \int_{\mathcal{S}} \phi(s_t, a_t, s') V_{f_i}(s') \mathrm{d}s' \right) \right]^2$$
$$\leq \sum_{t=1}^{T} (W_t^\top X_t)^2 \cdot \mathbb{1}\left( \|X_t\|_{\Sigma_t^{-1}}^2 \leq 1 \right) + (\mathrm{sp}(V^*) + 2)^2 \min\{d(\epsilon), T\}$$
$$\leq \sum_{i=1}^{T} (\beta + 4\epsilon) \cdot \left( \|X_t\|_{\Sigma_t^{-1}}^2 \wedge 1 \right) + (\mathrm{sp}(V^*) + 2)^2 \min\{d(\epsilon), T\}$$
$$\leq d(\epsilon) \beta \log T + (\mathrm{sp}(V^*)^2 + 4\mathrm{sp}(V^*) + 6) \min\{d(\epsilon), T\} + 2T\epsilon^2,$$

$$(F.19)$$

where we use a variant of (F.14) and (F.15), following that

$$\sum_{t=1}^{T} \mathbb{1}\left( \|X_t\|_{\Sigma_t^{-1}}^2 \geq 1 \right) \leq \sum_{t=1}^{T} \|X_t\|_{\Sigma_t^{-1}}^2 \wedge 1 \leq \sum_{t=1}^{T} \|X_t\|_{\Sigma_t^{-1}} \wedge 1 \leq d(\epsilon),$$

and the last inequality results from a similar proof as (F.17) and (F.18) using Cauchy-Swartz and AM-GM inequality. Thus, we have $\kappa_{\mathrm{G}} \leq \mathcal{O}(d \log(\mathrm{sp}(V^*)T/\sqrt{d}\epsilon))$. □

### F.5 DISCUSSION ABOUT PERFORMANCE ON CONCRETE EXAMPLES

In this subsection, we show performance of LOOP for specific problems. LOOP achieves an $\tilde{\mathcal{O}}(\sqrt{T})$ regret, which is nearly minimax optimal in $T$, in linear AMDP and linear mixture AMDP.

**Linear AMDP** Recall that linear function class is defined as $\mathcal{H} = \left\{ (Q, J) : Q(\cdot, \cdot) = \omega^\top \phi(\cdot, \cdot) \big| \|\omega\|_2 \leq \frac{1}{2}\mathrm{sp}(V^*)\sqrt{d}, J \in \mathcal{J}_{\mathcal{H}} \right\}$. Consider the $\rho$-covering number, note that

$$|Q(s, a) - Q'(s, a)| \leq |(\omega - \omega')^\top \phi(s, a)| \leq \sqrt{2} \cdot \|\omega - \omega'\|_1.$$

Based on the Lemma 20, combine the fact that $|\mathcal{J}_{\mathcal{H}}| \leq 2$ and its $\rho$-covering number $\mathcal{N}_\rho(\mathcal{J}_{\mathcal{H}})$ is at most $2\rho^{-1}$, we can get the log covering number of the hypotheses class $\mathcal{H}$ is upper bounded by

$$\log \mathcal{N}_{\mathcal{H}}(\rho) \leq d \log \left( \mathrm{sp}(V^*) 2^{\frac{3}{2}} d^{\frac{3}{2}} \rho^{-2} \right),$$

$$(F.20)$$

by taking $\alpha = w$, $P = d$, $B = \mathrm{sp}(V^*)\sqrt{d}/2$. Recall that Proposition 5 indicates that

$$d_{\mathrm{G}} \leq \mathcal{O}\left(d \log \left(\mathrm{sp}(V^*)\sqrt{d}\rho^{-1}\right) \log T\right), \quad \kappa_{\mathrm{G}} \leq \mathcal{O}\left(d \log \left(\mathrm{sp}(V^*)\sqrt{d}\rho^{-1}\right)\right).$$

$$(F.21)$$

Combine (F.20), (F.21) and the regret guarantee in Theorem 3, we get

$$\mathrm{Reg}(T) \leq \mathcal{O}\left( \mathrm{sp}(V^*) \max\{d_{\mathrm{G}}, \kappa_{\mathrm{G}}\} \sqrt{T \log \left(T\mathcal{N}_{\mathcal{H} \cup \mathcal{G}}^2(1/T)/\delta\right)\mathrm{sp}(V^*)} \right) \leq \tilde{\mathcal{O}}\left( \mathrm{sp}(V^*)^{\frac{3}{2}} d^{\frac{3}{2}} \sqrt{T} \right).$$

For linear AMDPs, our method achieves $\tilde{\mathcal{O}}(\mathrm{sp}(V^*)^{\frac{3}{2}} d^{\frac{3}{2}} \sqrt{T})$ regret for both linear and generalized linear AMDPs. In comparison, the FOPO algorithm (Wei et al., 2021) achieves the best-known $\tilde{\mathcal{O}}(\mathrm{sp}(V^*) d^{\frac{3}{2}} \sqrt{T})$ regret. Our method incurs an additional constant $\mathrm{sp}(V^*)^{\frac{1}{2}}$ in the regret bound.

**Linear mixture** Recall that the Proposition 8 posits that AGEC of the linear mixture probelm satisfies that $\max\{\sqrt{d_{\mathrm{G}}}, \kappa_{\mathrm{G}}\} \leq \mathcal{O}(d\sqrt{\log T})$. Note that the hypotheses class is defined as

$$\mathcal{H} = \{(\mathbb{P}, r) : \mathbb{P}(\cdot|s, a) = \theta^\top \phi(s, a, \cdot),\ r(s, a) = \theta^\top \psi(s, a)|\ \|\theta\|_2 \leq 1\}.$$

Consider the covering number note that both transition function and reward can be written as

(i). $|(\mathbb{P} - \mathbb{P}')(s'|s, a)| = |(\theta - \theta')^\top \phi(s, a, s')| \leq \sqrt{d} \cdot \|\theta - \theta'\|_1,$

(ii). $|(r - r')(s, a)| = |(\theta - \theta')^\top \psi(s, a)| \leq \sqrt{d} \cdot \|\theta - \theta'\|_1.$

Based on the Lemma 20, the log covering number of $\mathcal{H}_{\mathrm{LM}}$ is upper bounded by

$$\log \mathcal{N}_{\mathcal{H}}(\rho) \leq 2d \log \left(d^{\frac{3}{2}} \rho^{-1}\right),$$

by taking $\alpha = \theta$, $P = d$, $B = 1$. Combine results above and Theorem 3, we get

$$\mathrm{Reg}(T) \leq \mathcal{O}\left(\mathrm{sp}(V^*) \max\{d_{\mathrm{G}}, \kappa_{\mathrm{G}}\} \sqrt{T \log\left(T \mathcal{N}^2_{\mathcal{H} \cup \mathcal{G}}(1/T)/\delta\right) \mathrm{sp}(V^*)}\right) \leq \tilde{\mathcal{O}}\left(\mathrm{sp}(V^*)^{\frac{3}{2}} d^{\frac{3}{2}} \sqrt{T}\right).$$

At our best knowledge, the UCRL2-VTR (Wu et al., 2022) achieves the best $\tilde{\mathcal{O}}(Dd\sqrt{T})$ regret for linear mixture AMDP, where $D$ is the diameter under communicating AMDP assumption and it is provable that $\mathrm{sp}(V^*) \leq D$ (Wang et al., 2022). We remark that the two algorithms are incomparable under different assumptions and both achieve a near minimax optimal regret at $\tilde{\mathcal{O}}(\sqrt{T})$.

# G TECHNICAL LEMMAS

In this section, we provide useful technical lemmas used in later theoretical analysis. Most are directly borrowed from existing works and proof of modified lemmas is provided in Section G.1.

**Lemma 12.** Given function class $\Phi$ defined on $\mathcal{X}$, and a family of probability measures $\Gamma$ over $\mathcal{X}$. Suppose sequence $\{\phi_k\}_{k=1}^K \subset \Phi$ and $\{\mu_k\}_{k=1}^K \subset \Gamma$ satisfy that for all $k \in [K]$, $\sum_{t=1}^{k-1}(\mathbb{E}_{\mu_t}[\phi_k])^2 \leq \beta$. Then, for all $k \in [K]$, we have

$$\sum_{t=1}^k \mathbb{1}\left(\left|\mathbb{E}_{\mu_t}[\phi_t]\right| > \epsilon\right) \leq \left(\frac{\beta}{\epsilon^2} + 1\right) \dim_{\mathrm{DE}}(\Phi, \Pi, \epsilon).$$

*Proof.* See Lemma 43 of Jin et al. (2021) for detailed proof.

**Lemma 13** (Pigeon-hole principle). Given function class $\Phi$ defined on $\mathcal{X}$ with $|\phi(x)| \leq C$ for all $\phi \in \Phi$ and $x \in \mathcal{X}$, and a family of probability measure over $\mathcal{X}$. Suppose sequence $\{\phi_k\}_{k=1}^K \subset \Phi$ and $\{\mu_k\}_{k=1}^K \subset \Gamma$ satisfy that for all $k \in [K]$, it holds $\sum_{t=1}^{k-1}(\mathbb{E}_{\mu_t}[\phi_k])^2 \leq \beta$. Let $d_{\mathrm{DE}} = \dim_{\mathrm{DE}}(\Phi, \Gamma, \epsilon)$ be the DE dimension, then for all $k \in [K]$ and $\epsilon > 0$, we have

$$\sum_{t=1}^k \left|\mathbb{E}_{\mu_t}[\phi_t]\right| \leq 2\sqrt{d_{\mathrm{DE}}\beta k} + \min\{k, d\}C + k\epsilon,$$

and

$$\sum_{t=1}^k \left[\mathbb{E}_{\mu_t}[\phi_t]\right]^2 \leq d_{\mathrm{DE}}\beta \log k + \min\{k, d\}C^2 + k\epsilon^2.$$

*Proof.* See Section G.1.1.

**Lemma 14.** Given function class $\Phi$ defined on $\mathcal{X}$ with $|\phi(x)| \leq C$ for all $\phi \in \Phi$ and $x \in \mathcal{X}$, and a family of probability measure over $\mathcal{X}$. Let $d_{\mathrm{DE}} = \dim_{\mathrm{DE}}(\Phi, \Gamma, \epsilon)$ be the DE dimension, then for all $k \in [K]$ and $\epsilon > 0$, we have

$$\sum_{t=1}^k \left|\mathbb{E}_{\mu_k}[\phi_k]\right| \leq \left[d_{\mathrm{DE}}(1 + \log K) \sum_{k=1}^K \sum_{t=1}^{k-1}(\mathbb{E}_{\mu_t}[\phi_k])^2\right]^{1/2} + \min\{d_{\mathrm{DE}}, k\}C + k\epsilon.$$

*Proof.* See Section G.1.2.

**Lemma 15** (*d*-upper bound). Let $\Phi$ and $\Psi$ be sets of $d$-dimensional vectors and $\|\phi\|_2 \leq B_\phi, \|\psi\|_2 \leq B_\psi$ for any $\phi \in \Phi$ and $\psi \in \Psi$. If there exists set $(\phi_1, \ldots, \phi_m)$ and $(\psi_1, \ldots, \psi_m)$ such that for all $t \in [m]$, $\sqrt{\sum_{k=1}^{t-1} \langle \phi_t, \psi_k \rangle^2} \leq c_1 \varepsilon$ and $|\langle \phi_t, \psi_t \rangle| > c_2 \varepsilon$, where $c_1 \geq c_2 > 0$ is a constant and $\varepsilon > 0$, then the number of elements in set is bounded by $m \leq \mathcal{O}\big(d \log(B_\phi B_\psi / \varepsilon)\big)$.

*Proof.* See Section G.1.3.

**Lemma 16.** For any sequence of positive reals $x_1, \ldots, x_m$, it holds that $\frac{\sum_{i=1}^m x_i}{\sqrt{\sum_{i=1}^m i x_i^2}} \leq \sqrt{1 + \log n}$.

*Proof.* See Lemma 6 in Dann et al. (2021) for detailed proof.

**Lemma 17.** Let $\{x_i\}_{i \in [t]}$ be a sequence of vectors defined over Hilbert space $\mathcal{X}$. Let $\Lambda_0$ be a positive definite matrix and $\Lambda_t = \Lambda_0 + \sum_{i=1}^{t-1} x_t x_t^\top$. It holds that

$$\sum_{i=1}^t \|x_t\|_{\Lambda_t^{-1}}^2 \wedge 1 \leq 2 \log \left( \frac{\det \Lambda_{t+1}}{\det \Lambda_0} \right).$$

*Proof.* See Elliptical Potential Lemma (EPL) in Dani et al. (2008) for a detailed proof.

**Lemma 18** (Freedman's inequality). Let $X_1, \ldots, X_T$ be a real-valued martingale difference sequence adapted to filtration $\{\mathscr{F}_t\}_{t=1}^T$. Assume for all $t \in [T]$ $X_t \leq R$, then for any $\eta \in (0, 1/R)$, with probability greater than $1 - \delta$

$$\sum_{t=1}^T X_t \leq \mathcal{O}\Big(\eta \sum_{t=1}^T \mathbb{E}\big[X_t^2 | \mathscr{F}_t\big] + \frac{\log(1/\delta)}{\eta}\Big),$$

*Proof.* See Lemma 7 in Agarwal et al. (2014) for detailed proof.

**Lemma 19** (Scaling lemma). Let $\phi : \mathcal{S} \times \mathcal{A} \mapsto \mathbb{R}^d$ be a $d$-dimensional feature mapping, there exists an invertible linear transformation $A \in \mathbb{R}^{d \times d}$ such that for any bounded function $f : \mathcal{S} \times \mathcal{A} \mapsto \mathbb{R}$ and $\mathbf{z} \in \mathbb{R}^d$ defined by

$$f(s, a) = \phi(s, a)^\top \mathbf{z},$$

we have $\|A\phi(s, a)\| \leq 1$ and $\|A^{-1} z\| \leq \sup_{s,a} |f| \sqrt{d}$ for all $(s, a) \in \mathcal{S} \times \mathcal{A}$.

*Proof.* See Lemma 8 in Wei et al. (2021) for detailed proof.

In Theorem 3, the proved regret contains the logarithmic term of the $1/T$-covering number of the function classes $\mathcal{N}_\mathcal{H}(1/T)$, which can be regarded as a surrogate cardinality of the function class $\mathcal{H}$. Here, we provide a formal definition of $\rho$-covering and the upper bound of $\rho$-covering number.

**Definition 17** ($\rho$-covering). The $\rho$-covering number of a function class $\mathcal{F}$ is the minimum integer $t$ satisfying that there exists subset $\mathcal{F}' \subseteq \mathcal{F}$ with $|\mathcal{F}'| = t$ such that for any $f \in \mathcal{F}$ we can find a correspondence $f' \in \mathcal{F}'$ that it holds $\|f - f'\|_\infty \leq \rho$.

**Lemma 20** ($\rho$-covering number). Let $\mathcal{F}$ be a function defined over $\mathcal{X}$ that can be parametrized by $\boldsymbol{\alpha} = (\alpha_1, \ldots, \alpha_P) \in \mathbb{R}^P$ with $|\alpha_i| \leq B$ for all $i \in [P]$. Suppose that for any $f, f' \in \mathcal{F}$ it holds that $\sup_{x \in \mathcal{X}} |f(x) - f'(x)| \leq L \|\boldsymbol{\alpha} - \boldsymbol{\alpha}'\|_1$ and let $\mathcal{N}_\mathcal{F}(\rho)$ be the $\rho$-covering number of $\mathcal{F}$, then

$$\log \mathcal{N}_\mathcal{F}(\rho) \leq P \log \Big( \frac{2BLP}{\rho} \Big).$$

*Proof.* See Lemma 12 in Wei et al. (2021) for detailed proof.

## G.1 PROOF OF TECHNICAL LEMMAS

In this subsection, we present the proofs of technical auxiliary lemmas with modifications.

### G.1.1 PROOF OF LEMMA 13

*Proof of Lemma 13.* The first statement is directly from Lemma 41 in Jin et al. (2021), and the second statement follows a similar procedure as below. Note that Lemma 12 suggests that

$$\sum_{t=1}^{k} \mathbb{1}\left( \left[\mathbb{E}_{\mu_t}[\phi_t]\right]^2 > \epsilon^2 \right) \leq \left( \frac{\beta}{\epsilon^2} + 1 \right) \dim_{\mathrm{DE}}(\Phi, \Gamma, \epsilon),$$

and note that the sum of squared expectation can be decomposed as

$$\sum_{t=1}^{k} \left[\mathbb{E}_{\mu_t}[\phi_t]\right]^2 = \sum_{t=1}^{k} \left[\mathbb{E}_{\mu_t}[\phi_t]\right]^2 \mathbb{1}\left( \left[\mathbb{E}_{\mu_t}[\phi_t]\right]^2 > \epsilon^2 \right) + \sum_{t=1}^{k} \left[\mathbb{E}_{\mu_t}[\phi_t]\right]^2 \mathbb{1}\left( \left[\mathbb{E}_{\mu_t}[\phi_t]\right]^2 \leq \epsilon^2 \right)$$

$$\leq \sum_{t=1}^{k} \left[\mathbb{E}_{\mu_t}[\phi_t]\right]^2 \mathbb{1}\left( \left[\mathbb{E}_{\mu_t}[\phi_t]\right]^2 > \epsilon^2 \right) + k\epsilon^2. \tag{G.1}$$

Assume sequence $\left[\mathbb{E}_{\mu_1}[\phi_1]\right]^2, \ldots, \left[\mathbb{E}_{\mu_k}[\phi_k]\right]^2$ are sorted in the decreasing order and consider $t \in [k]$ such that $\left[\mathbb{E}_{\mu_t}[\phi_t]\right]^2 > \epsilon^2$, there exists a constant $\alpha \in (\epsilon^2, \left[\mathbb{E}_{\mu_t}[\phi_t]\right]^2)$ satisfying

$$t \leq \sum_{i=1}^{k} \mathbb{1}\left( \left[\mathbb{E}_{\mu_i}[\phi_i]\right]^2 > \alpha \right) \leq \left( \frac{\beta}{\alpha} + 1 \right) \dim_{\mathrm{DE}}(\Phi, \Gamma, \sqrt{\alpha}) \leq \left( \frac{\beta}{\alpha} + 1 \right) \dim_{\mathrm{DE}}(\Phi, \Gamma, \epsilon),$$

where the last inequality is based on the fact that the DE dimension is monotonically decreasing in terms of $\epsilon$ as proposed in Jin et al. (2021). Denote $d_{\mathrm{DE}} = \dim_{\mathrm{DE}}(\Phi, \Gamma, \epsilon)$ and the inequality above implies that $\alpha \leq d_{\mathrm{DE}}\beta/t - d$. Thus, we have $\left[\mathbb{E}_{\mu_t}[\phi_t]\right]^2 \leq d_{\mathrm{DE}}\beta/t - d$. Beside, based on the definition we also have $\left[\mathbb{E}_{\mu_t}[\phi_t]\right]^2 \leq C^2$ and thus $\left[\mathbb{E}_{\mu_t}[\phi_t]\right]^2 \leq \min\{d_{\mathrm{DE}}\beta/t - d, C^2\}$, then

$$\sum_{t=1}^{k} \left[\mathbb{E}_{\mu_t}[\phi_t]\right]^2 \mathbb{1}\left( \left[\mathbb{E}_{\mu_t}[\phi_t]\right]^2 > \epsilon^2 \right) \leq \min\{d_{\mathrm{DE}}, k\}C^2 + \sum_{t=d+1}^{k} \left( \frac{d_{\mathrm{DE}}\beta}{t - d_{\mathrm{DE}}} \right)$$

$$\leq \min\{d_{\mathrm{DE}}, k\}C^2 + d_{\mathrm{DE}} \cdot \beta \int_0^k \frac{1}{t}\mathrm{d}t$$

$$\leq \min\{d_{\mathrm{DE}}, k\}C^2 + d_{\mathrm{DE}} \cdot \beta \log k. \tag{G.2}$$

Combine (G.1) and (G.2), then finishes the proof. □

### G.1.2 PROOF OF LEMMA 14

We remark that the proof provided in this subsection follows the almost same procedure as Lemma 3.16 in Zhong et al. (2022) with adjustment, and we preserve it for comprehension.

*Proof of Lemma 14.* Denote $d_{\mathrm{DE}} = \dim_{\mathrm{DE}}(\Phi, \Gamma, \epsilon)$, $\widehat{\epsilon}_{t,k} = |\mathbb{E}_{\mu_t}[\phi_k]|$ and $\epsilon_{t,k} = \widehat{\epsilon}_{t,k}\mathbb{1}(\widehat{\epsilon}_{t,k} > \epsilon)$ for $t, k \in [K]$, $\mu_t \in \Gamma$ and $\phi_k \in \Phi$. The proof follows the procedure below. Consider $K$ empty buckets $B_0, \ldots, B_{K-1}$ as initialization, and we examine $\epsilon_{k,k}$ one by one for all $k \in [K]$ as below:

**Case 1** If $\epsilon_{k,k} = 0$, i.e., $\widehat{\epsilon}_{k,k} \leq \epsilon$, then discard it.

**Case 2** If $\epsilon_{k,k} > 0$, i.e., $\widehat{\epsilon}_{k,k} > \epsilon$, at bucket $j$ we add $k$ into $B_j$ if $\sum_{t \leq k-1, t \in B_j}(\epsilon_{t,k})^2 \leq (\epsilon_{k,k})^2$, otherwise we continue with the next bucket $B_{j+1}$.

Denote by $b_k$ the index of bucket that at step $k$ the non-zero $\epsilon_{k,k}$ falls in, i.e. $k \in B_{b_k}$. Based on the rule above, it holds that

$$\sum_{k=1}^{K}\sum_{t=1}^{k-1}(\epsilon_{t,k})^2 \geq \sum_{k=1}^{K} \sum_{0 \leq j \leq b_k-1, b_k \geq 1} \sum_{t \leq k-1, t \in B_j} (\epsilon_{t,k})^2 \geq \sum_{k=1}^{K} b_k \cdot (\epsilon_{k,k})^2,$$

where the first inequality arises from $\{t \in B_j : t \leq k - 1, \, 0 \leq j \leq b_k - 1, b_k \geq 1\} \subseteq [k-1]$ due to the discarding of the $b_k$th bucket, and the second equality directly follows the allocation rule

such that $\sum_{t \leq k-1, t \in B_j} (\epsilon_{t,k})^2 \geq (\epsilon_{k,k})^2$ for any $j \leq b_k - 1$. Recall that based on the definition of distributional eluder (DE) dimension, it is suggested the size $|B_j|$ is no larger than $d_{\mathrm{DE}}$. Then,

$$\sum_{k=1}^{K} b_k (\epsilon_{k,k})^2 = \sum_{j=1}^{K-1} j \sum_{t \in B_j} (\epsilon_{t,t})^2 \qquad \text{(re-summation)}$$

$$\geq \sum_{j=1}^{K-1} \frac{j}{|B_j|} \left( \sum_{t \in B_j} \epsilon_{t,t} \right)^2 \geq \sum_{j=1}^{K-1} \frac{j}{d_{\mathrm{DE}}} \left( \sum_{t \in B_j} \epsilon_{t,t} \right)^2 \qquad (|B_j| \leq d_{\mathrm{DE}})$$

$$\geq (d_{\mathrm{DE}} (1 + \log K))^{-1} \left( \sum_{j=1}^{K-1} \sum_{t \in B_j} \epsilon_{t,t} \right)^2 = (d_{\mathrm{DE}} (1 + \log K))^{-1} \left( \sum_{t \in [K] \setminus B_0} \epsilon_{t,t} \right)^2, \quad \text{(G.3)}$$

where the second inequality follows Lemma 16. Combine the (G.1.2) and (G.3) above, we have

$$\sum_{k=1}^{K} \widehat{\epsilon}_{k,k} \leq \sum_{k=1}^{K} \epsilon_{k,k} + K\epsilon \leq \sum_{t \in [K] \setminus B_0} \epsilon_{t,t} + \min\{d_{\mathrm{DE}}, K\} \|\phi\|_\infty + K\epsilon$$

$$\leq \left[ d_{\mathrm{DE}} (1 + \log K) \sum_{k=1}^{K} \sum_{t=1}^{k-1} (\epsilon_{t,k})^2 \right]^{1/2} + \min\{d_{\mathrm{DE}}, K\} C + K\epsilon$$

$$\leq \left[ d_{\mathrm{DE}} (1 + \log K) \sum_{k=1}^{K} \sum_{t=1}^{k-1} (\widehat{\epsilon}_{t,k})^2 \right]^{1/2} + \min\{d_{\mathrm{DE}}, K\} C + K\epsilon.$$

Substitute the definition $\widehat{\epsilon}_{t,k} = |\mathbb{E}_{\mu_t}[\phi_k]|$ back into the inequality, then finishes the proof. $\qquad \square$

### G.1.3 PROOF OF LEMMA 15

*Proof of Lemma 15.* For notation simplicity, denote $\Lambda_t = \sum_{k=1}^{t-1} \psi_t \psi_t^\top + \frac{\varepsilon^2}{B_\phi^2} \cdot \mathbf{I}$, then for all $t \in [m]$ we have $\|\phi_t\|_{\Lambda_t} \leq \sqrt{\sum_{k=1}^{t-1} (\phi_t^\top \psi_k)^2 + \frac{\varepsilon^2}{B_\phi^2} \|\phi_t\|_2^2} = \sqrt{c_1^2 + 1}\, \varepsilon$ based on the given condition. Using the Cauchy-Swartz inequality and results above, then it holds $\|\psi_t\|_{\Lambda_t^{-1}} \geq |\langle \phi_t, \psi_t \rangle| / \|\phi_t\|_{\Lambda_t} = c_2 / \sqrt{c_1^2 + 1}$. On one hand, the matrix determinant lemma ensures that

$$\det \Lambda_m = \det \Lambda_0 \cdot \prod_{t=1}^{m-1} \left( 1 + \|\psi_t\|_{\Lambda_t^{-1}}^2 \right) \geq \left( 1 + \frac{c_2^2}{1 + c_1^2} \right)^{m-1} \left( \frac{\varepsilon^2}{B_\phi^2} \right)^d. \qquad \text{(G.4)}$$

On the other hand, according to the definition of $\Lambda_t$, we have

$$\det \Lambda_m \leq \left( \frac{\mathrm{Tr}(\Lambda_m)}{d} \right)^d \leq \left( \sum_{k=1}^{t-1} \frac{\|\psi_k\|_2^2}{d} + \frac{\varepsilon^2}{B_\phi^2} \right)^d \leq \left( \frac{B_\psi^2 (m-1)}{d} + \frac{\varepsilon^2}{B_\phi^2} \right)^d. \qquad \text{(G.5)}$$

Combine (G.4) and (G.5), if we take logarithms at both sides, then we have

$$m \leq 1 + d \log \left( \frac{B_\phi^2 B_\psi^2 (m-1)}{d\varepsilon^2} + 1 \right) \bigg/ \log \left( 1 + \frac{c_2^2}{1 + c_1^2} \right).$$

After simple calculations, we can obtain that $m$ is upper bounded by $\mathcal{O}\big(d \log(B_\phi B_\psi / \varepsilon)\big)$. $\qquad \square$

## H SUPPLEMENTARY DISCUSSIONS

### H.1 PROOF SKETCH OF MLE-BASED RESULTS

In this subsection, we provide the proof sketch of Theorem 4. We first introduce several useful lemmas, which is the variant of ones in Appendix D for MLE-based problems, and most have been fully researched in Liu et al. (2022; 2023a); Xiong et al. (2023). As there's no significant technical gap between episodic and average-reward for model-based problems, we only provide a proof sketch.

**Lemma 21** (Akin to Lemma 9). Under Assumptions 1-2, MLE-LOOP is an optimistic algorithm such that it ensures $J_t \geq J^*$ for all $t \in [T]$ with probability greater than $1 - \delta$.

*Proof Sketch of Lemma 21.* See Proposition 13 in Liu et al. (2022) with slight modifications. ☐

**Lemma 22** (Akin to Lemma 10). For fixed $\rho > 0$ and a pre-determined optimistic parameter $\beta = c(\log(T\mathcal{B}_{\mathcal{H}}(\rho)/\delta) + T\rho)$ where constant $c > 0$, it holds that

$$\sum_{i=1}^{t-1} \|\mathbb{E}_{\zeta_i}[l_{f_i}(f_t, f_t, \zeta_i)]\|_1 = \sum_{i=1}^{t-1} \text{TV}\big(\mathbb{P}_{f_t}(\cdot|s_i, a_i), \mathbb{P}_{f^*}(\cdot|s_i, a_i)\big) \leq \mathcal{O}(\sqrt{\beta t}), \quad \text{(H.1)}$$

for all $t \in [T]$ with probability greater than $1 - \delta$.

*Proof Sketch of Lemma 22.* See Proposition 14 in Liu et al. (2022) with slight modifications. ☐

**Lemma 23** (Akin to Lemma 11). Let $\mathcal{N}(T)$ be the switching cost with time horizon $T$, given fixed covering coefficient $\rho > 0$ and pre-determined optimistic parameter $\beta = c(\log(T\mathcal{B}_{\mathcal{H}}(\rho)/\delta) + T\rho)$ where $c$ is a large enough constant, with probability greater than $1 - 2\delta$ we have

$$\mathcal{N}(T) \leq \mathcal{O}\big(\kappa_{\text{G}} \cdot \text{poly}(\log T) + \beta^{-1}T\epsilon^2\big),$$

where $\kappa_{\text{G}}$ is the transferability coefficient with respect to MLE-AGEC$(\mathcal{H}, \{l_{f'}\}, \epsilon)$.

*Proof Sketch of Lemma 23.* The proof is almost the same as Lemma 11.

**Step 1: Bound the difference of discrepancy between the minimizer and $f^*$.**

As proposed in Proposition 14, Liu et al. (2022), $0 \leq \sum_{i=1}^{t} \text{TV}\big(\mathbb{P}_{f^*}(\cdot|s_i, a_i), \mathbb{P}_{g_i}(\cdot|s_i, a_i)\big)^2 \leq \beta$ holds with high probability if the update happens at $t$-th step. Based on the AM-GM inequaliaty, we have

$$0 \leq \sum_{i=1}^{t} \text{TV}\big(\mathbb{P}_{f^*}(\cdot|s_i, a_i), \mathbb{P}_{g_i}(\cdot|s_i, a_i)\big) \leq \sqrt{\beta t}. \quad \text{(H.2)}$$

**Step 2: Bound the expected discrepancy between updates.**

Note that for all $t + 1 \in [T]$, the update happens only if

$$\sum_{i=1}^{t} \text{TV}\big(\mathbb{P}_{f_t}(\cdot|s_i, a_i), \mathbb{P}_{g_i}(\cdot|s_i, a_i)\big) > 3\sqrt{\beta t}. \quad \text{(H.3)}$$

Combine the (H.2) and (H.3) above, and apply the triangle inequality, we have

$$\sum_{i=1}^{t} \text{TV}\big(\mathbb{P}_{f_t}(\cdot|s_i, a_i), \mathbb{P}_{f^*}(\cdot|s_i, a_i)\big)$$

$$\geq \sum_{i=1}^{t} \text{TV}\big(\mathbb{P}_{f_t}(\cdot|s_i, a_i), \mathbb{P}_{g_t}(\cdot|s_i, a_i)\big) - \text{TV}\big(\mathbb{P}_{f^*}(\cdot|s_i, a_i), \mathbb{P}_{g_t}(\cdot|s_i, a_i)\big) \geq 2\sqrt{\beta t}.$$

and the construction of confidence set ensures that $\sum_{i=1}^{\tau_t} \text{TV}\big(\mathbb{P}_{f_t}(\cdot|s_i, a_i), \mathbb{P}_{f^*}(\cdot|s_i, a_i)\big) \leq \sqrt{\beta \tau_t}$ with high probability (Liu et al., 2022, Proposition 14). Recall the definition of MLE-transferability coefficient, then the switching cost can be bounded following the same argument in Lemma 11. ☐

*Proof Sketch of Theorem 4.* Recall that

$$\text{Reg}(T) \leq \underbrace{\sum_{i=1}^{T} \mathcal{E}(f_t)(s_t, a_t)}_{\text{Bellman error}} + \underbrace{\sum_{t=1}^{T} \Big(\mathbb{E}_{s_{t+1} \sim \mathbb{P}(\cdot|s_t, a_t)}[V_t(s_{t+1})] - V_t(s_t)\Big)}_{\text{Realization error}}, \quad \text{(H.4)}$$

where the inequality follows the optimism in Lemma 21. Combine Lemma 22, Lemma 23 and the definition of MLE-AGEC (see Definition 9), then we can finish the proof. ☐

---

**Algorithm 3** Extended Value Iteration (EVI)

---

**Input:** hypothesis $f = (\mathbb{P}_f, r_f)$, desired accuracy level $\epsilon$.
**Initialize:** $V^{(0)}(s) = 0$ for all $s \in \mathcal{S}$, $J^{(0)} = 0$ and counter $i = 0$.
 1: **repeat**
 2:     **for** $s \in \mathcal{S}$ and $a \in \mathcal{A}$ **do**
 3:         Set $Q^{(i)}(s,a) \leftarrow r_f(s,a) + \mathbb{E}_{s' \sim \mathbb{P}_f(s,a)}[V^{(i)}(s')] - J^{(i)}$
 4:         Update $V^{(i+1)}(s) \leftarrow \max_{a \in \mathcal{A}} Q^{(i)}(s,a)$
 5:     Update counter $i \leftarrow i + 1$
 6: **until** $\max_{s \in \mathcal{S}} \{V^{(i+1)}(s) - V^{(i)}(s)\} - \min_{s \in \mathcal{S}} \{V^{(i+1)}(s) - V^{(i)}(s)\} \leq \epsilon$

---

## H.2 EXTENDED VALUE ITERATION (EVI) FOR MODEL-BASED HYPOTHESES

In model-based problems, the discrepancy function sometimes relies on the optimal state bias function $V_f$ and optimal average-reward $J_f$ (see linear mixture model in Section C). In this section, we provide an algorithm, extended value iteration (EVI) proposed in Auer et al. (2008), to output the optimal function and average-reward under given a model-based hypothesis $f = (\mathbb{P}_f, r_f)$. See Algorithm 3 for complete pseudocode. The convergence of EVI is guaranteed by the theorem below.

**Theorem 24.** UnderAssumption 1, there exists a unique centralized solution pair $(Q^*, J^*)$ to the Bellman optimality equation for any AMDP $\mathcal{M}_f$ characterized by hypothesis $f \in \mathcal{H}$. Then, if the extended value iteration (EVI) is stopped under the condition that

$$\max_{s \in \mathcal{S}} \{V^{(i+1)}(s) - V^{(i)}(s)\} - \min_{s \in \mathcal{S}} \{V^{(i+1)}(s) - V^{(i)}(s)\} \leq \epsilon,$$

then the achieved greedy policy $\pi^{(i)}$ is $\epsilon$-optimal such that $J^{\pi^{(i)}}_{\mathcal{M}_f} \geq J^*_{\mathcal{M}_f} + \epsilon$.

*Proof Sketch:* See Theorem 12 in Auer et al. (2008).

