# OpenReview forum: "Sample-efficient Learning of Infinite-horizon Average-reward MDPs with General Function Approximation"
_ICLR.cc/2024/Conference — ICLR 2024 poster_

### Official Review · Reviewer_jdKK · 2023-10-17

**Soundness:** 3 good
**Presentation:** 3 good
**Contribution:** 2 fair
**Rating:** 6
**Confidence:** 3

**Summary:**

This paper studies sample efficiency in infinite-horizon averaged MDP setting with general function approximation.
The authors propose average-reward generalized eluder coefficient to characterize the exploration difficulty in learning, and contribute an algorithm called Fixed-Point Local Optimization and establish sublinear regret with polynomial dependence on AGEC coefficient and span of optimal state bias, etc.

**Strengths:**

The paper writing is clear. The notations, definitions are clearly stated. Comparison with previous works are also clearly summarized in Table 1.

**Weaknesses:**

1. It seems to me the main contribution of this paper is to transfer and generalize some existing techniques and results (especially in finite horizon setting) to infinite horizon averaged return setting. The definition of AGEC Complexity Measure, FLOP algorithm and the techniques for analyzing lazy update rules seem share much similarity with previous literature like [1], [2]. I didn't find much novelty in algorithm design or technique analysis.

2. There is no conclusion section. I would suggest the authors at least have a few works to summarize the paper and discuss the future works.


[1] Zhong et. al., Gec: A unified framework for interactive decision making in mdp, pomdp, and beyond

[2] Xiong et. al., A general framework for sequential decision- making under adaptivity constraints


## Post Rebuttal

Thanks for the detailed reply. I think my main concerns are addressed, and I'm willing to increase my score.

**Questions:**

* What are the novelties in technical or algorithmic level in this paper? What are the new challenges for exploration in infinite horizon averaged reward setting?

* Is the "lazy policy update" really necessary? Although the authors explain the motivation for low policy switching is because of the additional cost in regret analysis. I'm curious whether it can be avoidable or it reveals some fundamental difficulty.

* In Theorem 3, the definition of $\beta$, is $sp(v^*)$ inside or outside of the log?

* Why the algorithm is called "Fixed-Point ..."? I'm not very understand why Eq. 4.1 is a fixed-point optimization problem.

---

> ### Author Response · Authors · 2023-11-17
> **Response to Reviewer jdKK (Part 1)**
>
> Thank you for your comments and assessments!
>
> > What are the novelties in the technical or algorithmic level in this paper? What are the new challenges for exploration in infinite horizon averaged reward setting?
>
> - Please see our general response for details.
>
> > It seems to me the main contribution of this paper is to transfer and generalize some existing techniques and results (especially in finite horizon setting) to infinite horizon averaged return setting. The definition of AGEC Complexity Measure, FLOP algorithm and the techniques for analyzing lazy update rules seem share much similarity with previous literature like [1], [2]. I didn't find much novelty in algorithm design or technique analysis.
>
> - While we admit that the structural assumption over the Bellman dominance shares a similar form as the standard GEC (Zhong et al., 2022), the transferability assumption is new. Moreover, our algorithm design is different from the posterior-sampling-based approach presented in Zhong et al. (2022). We opt for an optimism-based algorithm for establishing a low policy switch updating, which is crucial in the achievement of sublinear regret for AMDPs.
> - Compared with Xiong et al. (2023), we adopt a weaker assumption over the Bellman dominance and provide a more rigorous definition of the eluder condition with nuanced adpation to the average-reward setting. Moreover, the average-reward RL adopts a completely different notion of regret, and the Q- or V- function does not naturally exist due to the infinite horizon without discount. we consider a broad subclass where a modified version of the Bellman optimality equation holds and present a novel regret decomposition for theoretical analysis of the explorability of AMDPs, resulting in significantly different regret analysis from the aforementioned work.
>
> > Is the "lazy policy update" really necessary? Although the authors explain the motivation for low policy switching is because of the additional cost in regret analysis. I'm curious whether it can be avoidable or it reveals some fundamental difficulty.
>
> - That's a really interesting question. There are two primary approaches to address online learning of AMDPs: the reduction to episodic MDPs, and the utilization of a low policy switching algorithm. However, achieving $\mathcal{\tilde{O}}(\sqrt{T})$ has proven to be challenging for reduction-based algorithms in existing works under the online learning framework (Wei et al., 2019, Wei et al., 2021), unless fortified with strong assumptions, such as ergodic AMDPs, where every policy is guaranteed to reach every state after a sufficient number of steps. In our pursuit of a near-optimal algorithm without relying on such stringent assumptions, we introduce an additional yet mild structural assumption to ensure the validity of a low-switching algorithm for general function classes.
>
> - At out best knowledge, all existing algorithms for online learning of AMDPs depend on such lazy update strategy to achieve a near optimal regret beyond strong assumptions, from the classical UCRL2 algorithm for tabular AMDPs (Jaksch et al., 2010), to FOPO for linear AMDPs (Wei et al., 2021) and (UCRL2-VTR for linear mixture MDPs (Wu et al., 2022). Furthermore, we speculate that the underlying mechanism behind this phenomenon lies in the challenge of accurately evaluating $\pi_t$ when imposing a step-by-step policy update, i.e., $\pi_{t}\neq\pi_{t+1}$ for all $t\in[T]$. This difficulty arises due to the intricate dependency structure in the historical data $\{(s_i,a_i)\}_{i\leq[t]}$.
>
> - Specifically, in the episodic setting, the learner executes the policy $\pi_t$ throughout the epside, and the value function can be computed by bellman equation starting from a zero function. Thus, the error can be controlled and is bounded by the cummulative error across episode following decoposition ${\rm Reg}(T)\leq\sum_{t=1}^T\sum_{h=1}^H\mathbb{E}[\mathcal{E}(f_t)(s_h^t,a_h^t)\vert\pi_t]$. This means that the estimation error of the current policy can be controlled by the sum of uncertainty. But for average reward case, if you update policy in each iteration, we cannot bound the suboptimality of the current policy by a sum of estimation errors as ${\rm Reg}(T)\leq\sum_{t=1}^T\mathcal{E}(f_t)(s_t,a_t)+\sum_{t=1}^T\left(\mathbb{E}[V_t(s_{t+1})]-V_t(s_t)\right)$.
>
> - The low-switching policy update in our algorithm ensures that the learner scrutinizes the policy based on data collected from new epochs. This corresponds to the update condition in our algorithm, where we only update when a substantial increase in cumulative discrepancy surpassing $3\beta$ has occurred since the last update.

---

> > ### Author Response · Authors · 2023-11-17
> > **Response to Reviewer jdKK (Part 2)**
> >
> > > In Theorem 3, the definition of $\beta$ , is  $sp(v^*)$ inside or outside of the log?
> >
> > - The definition of the optimistic parameter should be $\beta=c\cdot{\rm sp}(V^*)\cdot\log(T\mathcal{N}^2_{\mathcal{H}\cup\mathcal{G}}(1/T)/\delta)$, and we have revised the expression in the new version for cleaner presentation (highlighted ). Thank you for pointing out the confusion.
> >
> > > Why the algorithm is called "Fixed-Point ..."? I'm not very understand why Eq. 4.1 is a fixed-point optimization problem
> >
> >
> > - If we consider specific problems like linear AMDPs, the constraint of the algorithm can be indeed written into a fixed-point equation with an additional optimistic bonus, which is similar to the FOPO algorithm proposed by Wei, et al. (2021). Nevertheless, we acknowledge the reviewer's concern that this nomenclature might lead to confusion. In response, we have renamed the algorithm as the optimistic Local-fitted Optimization with Optimism (LOOP) in the revised version.
> >
> > > There is no conclusion section. I would suggest the authors at least have a few words to summarize the paper and discuss future works.
> >
> > - Thank you for your valuable suggestions. We have added a conclusion section in the revised version to summarize the paper and discuss future works.
> > ---
> > [A1] Chi Jin, Qinghua Liu, and Sobhan Miryoosefi. Bellman eluder dimension: New rich classes of rl problems, and sample-efficient algorithms. Advances in neural information processing systems,34:13406–13418, 2021.
> >
> > [A2] Nuoya Xiong, Zhuoran Yang, and Zhaoran Wang. A general framework for sequential decision making under adaptivity constraints. arXiv preprint arXiv:2306.14468, 2023.
> >
> > [A3] Chen-Yu Wei, Mehdi Jafarnia Jahromi, Haipeng Luo, and Rahul Jain. Learning infinite-horizon average-reward mdps with linear function approximation. In International Conference on Artificial Intelligence and Statistics, pp. 3007–3015. PMLR, 2021.
> >
> > [A4] Chen-Yu Wei, Mehdi Jafarnia Jahromi, Haipeng Luo, Hiteshi Sharma, and Rahul Jain. Model-free reinforcement learning in infinite-horizon average-reward markov decision processes. In International conference on machine learning, pp. 10170–10180. PMLR, 2020.
> >
> > [A5] Yue Wu, Dongruo Zhou, and Quanquan Gu. Nearly minimax optimal regret for learning infinite horizonaverage-reward mdps with linear function approximation. In International Conference on Artificial Intelligence and Statistics, pp. 3883–3913. PMLR, 2022.

---

### Official Review · Reviewer_q11B · 2023-11-01

**Soundness:** 3 good
**Presentation:** 2 fair
**Contribution:** 2 fair
**Rating:** 6
**Confidence:** 3

**Summary:**

The paper considers reinforcement learning for infinite-horizon average-reward MDPs under function approximation, and (i) generalizes the concept of eluder dimension to average-reward MDPs as a complexity measure, and (ii) proposes a new algorithm named FLOP to solve average-reward MDPs with low complexity in the sense of the generalized eluder dimension defined in the paper.

**Strengths:**

The problem addresses the challenging problem of infinite-horizon, average-reward MDPs in the function approximation setting. The idea of extending eluder dimension to this class of MDPs is a good idea. Also, the proposed algorithm that achieves sublinear regret seems to be a promising extension of the fitted Q-iteration algorithm, which takes lazy policy change into account.

**Weaknesses:**

The definition of AGEC (Definition 3), which is the central object and contribution in this paper, lacks clarity:
- Big-O notation is used in the definition, where the defined quantities $d_G$ and $\kappa_G$ (which also appear in $\mathcal{O}$) are smallest numbers that satisfy the inequalities that involve $\mathcal{O}$. This does not make much sense as a mathematical definition, with $\mathcal{O}$ being asymptotic.
- The set of discrepancy functions $\{l_f\}_f$ abruptly appears in Definition 3 without any proper definition. In later sections, we observe that it is an important quantity.
I would suggest a clear, mathematical definition of the complexity measure that constitutes one of the major contributions of this paper.

**Questions:**

In addition to the clarification of the definition of AGEC, I have the following questions:
- The function approximation error can be critical in RL, as its multiplying factor usually depends on the exploration performance in various forms. If we remove the realizability assumption, how does the additional term depend on the complexity measure defined in this paper?
- In Equation 2.1, what is $V^*(s,a)$? Should it be $Q^*$?
- In the abstract and in multiple places in the paper, $sp(v^*)$ appears with $v^*$. Should it be $sp(V^*)$? In the paper, it is assumed that $sp(V^*)$ is known. Is this knowledge necessary?

---

> ### Author Response · Authors · 2023-11-17
> **Response to Reviewer q11B**
>
> Thank you for your comments and assessments!
>
> > The definition of AGEC (Definition 3), which is the central object and contribution in this paper, lacks clarity.
>
> - Thank you for pointing out the lack of rigor in our definitions and your comments have been well-taken. In the original version, we used $\mathcal{O}$ notation to hide the contants in the definition, which may not adhere to mathematical rigor. In the revised version, we substitute the $\mathcal{O}$ notation with explicit definitions of absolute constants (highlighted in blue), which does not hurt the correctness of the result.
> - Regarding the dicrepancy function, we would like to highlight that the definition of the discrepancy function is inherently **problem-specific**, making it challenging to offer a unified definition applicable across all cases. To ensure clarity in our presentation, we provide a detailed definition of the problem-specific discrepancy function in each corresponding section such as (3.1), (3.3), (C.2).
>
> > The function approximation error can be critical in RL, as its multiplying factor usually depends on the exploration performance in various forms. If we remove the realizability assumption, how does the additional term depend on the complexity measure defined in this paper?
>
> - We would like to clarify that our complexity measure characterizes the difficulty in exploring the best-in-class model within the designated hypothesis class. On the other hand, the approximation error stems from the disparity between the best-in-class approximator and the ground-truth optimal function, which is not theoretically provable. Consequently, the removal of the realizability assumption will sure introduce additional error terms; however,  **it does not depend on the proposed complexity measure**.
> - A generalization of realizability for value-based problems is that there exists $f\in\mathcal{F}$ and $\epsilon_{\rm real}>0$ such that $\Vert Q^*-Q_f\Vert_\infty\leq\epsilon_{\rm real}$. The condition requires that the optimal bias state-action value function lies in the function class approximately with $\epsilon_{\rm real}$ error, enabling a more nuanced analysis on approximation. In this case, we conjecture that our algorithm, designed with a larger $\beta$ to accommodate misspecification, is capable of finding an $(\epsilon + \mathrm{poly}(d, \mathrm{sp}(V^*)) \cdot \epsilon_{\rm real})$-optimal policy using polynomial (in $\epsilon$) samples. It is indeed a very interesting future direction to design a unified sample-efficient algorithm considering the approximation error under general function approximation. We leave more meticulous analysis as future work.
>
> ---
>
> > In Equation 2.1, what is $V^*(s,a)$. Should it be $Q^*$ ?
>
> - Thank you for pointing that out. We have rectified the typo (Page 3), which should be
> $$J^*+Q^*(s,a)=r(s,a)+\mathbb{E}_{s'\sim\mathbb{P}(\cdot|s,a)}[V^*(s')].$$
>
> ---
>
> > In the abstract and in multiple places in the paper, ${\rm sp}(v^*)$ appears with $v^*$. Should it be ${\rm sp}(V^*)$ ? In the paper, it is assumed that ${\rm sp}(V^*)$ is known. Is this knowledge necessary?
>
> - Thank you for pointing out the confusion in the notations. We have now replaced ${\rm sp}(v^*)$ by ${\rm sp}(V^*)$ for a clearer presentation in the revised version. Regarding the span ${\rm sp}(V^*)$, we assert that the knowledge of the span of the optimal state bias function is crucial for **establishing a more rigorous regret guarantee** due to theoretical necessity. It is noteworthy that, unlike the episodic setting, there is no inherent upper bound for the optimal function $V^*$ in the average-reward setting, leading to a hypothesis class with uncontrollable complexity. The assumption of a known span implies a bounded span for the optimal bias function, with ${\rm sp}(V^)$ being available to the learner. This knowledge allows for the construction of a more stringent optimistic parameter. Alternatively, in practical scenarios, the learner can choose a **conservative estimation** of the span, such as a sufficiently large constant $B_{sp}>0$, ensuring $B_{sp}\geq{\rm sp}(V^*)$. The substitution of ${\rm sp}(V^*)$ with $B_{sp}>0$ in the construction of $\beta$ ensures the persistence of the optimistic planning in Algorithm 1, and results in an inflation of regret in squared root.

---

> > ### Comment · Reviewer_q11B · 2023-11-23
> >
> > I would like to thank the authors for their detailed responses, and the changes regarding the manuscript, which improved the presentation. I have no further questions.

---

> > > ### Author Response · Authors · 2023-11-23
> > > **Official Comment by Authors**
> > >
> > > Dear Reviewer,
> > >
> > > Thank you for reading our response and increasing the score. We appreciate your time and effort during the review process.
> > >
> > > Best,
> > >
> > > Authors

---

### Official Review · Reviewer_SbWP · 2023-11-01

**Soundness:** 3 good
**Presentation:** 2 fair
**Contribution:** 3 good
**Rating:** 8
**Confidence:** 2

**Summary:**

This paper studies infinite-horizon average-reward MDPs (AMDPs) with general function approximation. It extends the generalized eluder coefficient to average-reward generalized eluder coefficient (AGEC) under infinite-horizon MDPs. After showing that the low AGEC captures most existing structured MDPs, the paper develops an algorithm called FLOP to solve AMDPs with sublinear regret $O(\sqrt{T})$.

**Strengths:**

1. The paper provides a more general complexity measure AGEC that captures a large class of MDPs.
2. The design of confidence set is new, and the lazy update of policy is a good feature of algorithm design, which might be helpful for real implementations.

**Weaknesses:**

1. While the paper states that method covers most existing works, the detailed comparisons in terms of the regret performance are missed.

**Questions:**

Can authors provide a brief comparison between this work and existing works in terms of the regret results?

---

> ### Author Response · Authors · 2023-11-17
> **Response to Reviewer SbWP**
>
> We would like to thank the reviewer for the positive assessment of this work! Here's our response to the question.
>
> > Can authors provide a brief comparison between this work and existing works in terms of the regret results?
>
> - Please see our general response for details.

---

### Official Review · Reviewer_pAvA · 2023-11-10

**Soundness:** 3 good
**Presentation:** 3 good
**Contribution:** 2 fair
**Rating:** 5
**Confidence:** 3

**Summary:**

This paper explores the infinite-horizon average-reward Markov Decision Process (MDP) setting and introduces a comprehensive function approximation framework, accompanied by a corresponding complexity measure (AGEC) and an algorithm (FLOP). When compared with other work addressing AMDPs, the proposed framework covers the widest range of settings including both model-based and value-based, in addition, the theoretical analysis is based on the Bellman optimality assumption and the sample complexity is dependent on finite span, which is weaker compared with previous work (compared with communicating AMDP assumption and finite diameter dependence).

**Strengths:**

- This paper proposes a general framework for AMDP, encompassing both model-based and value-based.

- The authors propose a new complexity measure that is larger than the Eluder dimension.

- The authors propose a new algorithm that matches the state-of-the art sample complexity results.

**Weaknesses:**

- The sample complexity is based on GEC and is not entirely novel. Specifically, Definition 3 (Bellman Dominance) in this paper is of the same form with Definition 3.4 of Zhong et al. (2022b), where $d$ is defined such that the sum of the Bellman error being less than the in-sample training error plus the burn-in cost.

- The authors introduce the discrepancy function in the definition of AGEC, and shows a simple example of discrepancy function being the Bellman error for the value-based case, and $(r_g+P_g V_{f'})(s_t, a_t)-r(s_t, a_t)+V_{f'}(s_{t+1})$ for the model-based case.  However, there seems a lack of discussion regarding alternative choices for the discrepancy function, such as the Hellinger distance-based discrepancy in Zhong et al. (2022b).

- The algorithm is based on upper confidence bound, which the confidence region chosen based on the discrepancy function. This approach is closely related to Algorithm 1 in Chen et al. (2022b) and Algorithm 1 in Jin et al. (2021).

- The proposed algorithm is usually impractical to implement, since it involves solving a global constrained optimization.

**Questions:**

- In this paper, the authors directly assume the transferability of the discrepancy function, a concept closely related to Lemma 41 in Jin et al. (2021). Could the authors elaborate on the primary technical challenges they encountered while deriving their theoretical results, when adapt the GEC in Zhong et al. (2022b) to the infinite-horizon average-reward setting, with a constrained algorithm, which seems to be a generalization of the framework established by Jin et al. (2021)?

- Can the authors elaborate the optimality of the regret in Theorem 3, when restricted to each specific instances? i.e. linear mixture AMDP, linear AMDP, etc.

---

> ### Author Response · Authors · 2023-11-17
> **Response to Reviewer pAvA**
>
> Thank you for your comments and assessments!
>
> > The sample complexity is based on GEC and is not entirely novel. Specifically, Definition 3 (Bellman Dominance) in this paper is of the same form with Definition 3.4 of Zhong et al. (2022b), where is defined such that the sum of the Bellman error being less than the in-sample training error plus the burn-in cost.
> -  While we admit that the structural assumption over the Bellman dominance shares a similar form as the standard GEC (Zhong et al., 2022b) with nuanced adaption in average-reward terms, the transferability assumption is a novel addition. This additional structural constraint presents the unique challenges of AMDPs, and plays a pivotal role in facilitating the design of a low-switching algorithm.  This is particularly essential in addressing the inherent difficulty in exploring AMDPs.
> -  Moreover, our algorithm design diverges from the posterior-sampling-based approach presented in Zhong et al. (2022b). We embrace an optimism-based algorithm to establish a low policy switch updating mechanism, a crucial element in achieving sublinear regret for AMDPs. As a consequence, the technical details of our regret analysis differ significantly from the aforementioned work. Please refer to our general response more desciptions of technical novelties.
>
> >  However, there seems a lack of discussion regarding alternative choices for the discrepancy function, such as the Hellinger distance-based discrepancy in Zhong et al. (2022b).
>
> - We appreciate the reviewer's suggestion regarding the importance of discussing alternative choices for the discrepancy function, including the Hellinger distance-based discrepancy. Discussions about MLE-AGEC concerning an alternative discrepancy functions are presented in **Appendix C** (highlighted in blue) with a different transferability assumption over the structure. Here, we choose the total variation (TV)-based discrepancy function instead, which results from the slight difference between the choice of discrepancy: $l_{f'}(f,g,\zeta_t)=\frac{1}{2}|P_f(s_{t+1}|s_t,a_t)/P_{f^*}(s_{t+1}|s_t,a_t)-1|$ (ours) and $l_{f'}(f,\zeta_t)=\frac{1}{2}(P_f(s_{t+1}|s_t,a_t)/P_{f^*}(s_{t+1}|s_t,a_t)-1)^2$ (Zhong et al., 2022b). We remark that both choices of discrepancy is sufficient to cover MLE-based algorithms.
>
> > The algorithm is based on upper confidence bound, which the confidence region chosen based on the discrepancy function. This approach is closely related to Algorithm 1 in Chen et al. (2022b) and Algorithm 1 in Jin et al. (2021).
> > The proposed algorithm is usually impractical to implement, since it involves solving a global constrained optimization.
>
> - While acknowledging the resemblance of our algorithm design to previous lines of work on optimism-based algorithms for general function approximation, we notably diverge by employing a low-switch updating strategy coupled with a discrepancy-based updating condition.
> - Moreover, we would like to comment that our algorithm is in some sense information theoretical -- our work identifies a statistically tractable subclass of AMDPs, and the algorithm **serves as a certificate for the regret upper bound**. The main focus of our paper is to address the problem of minimal structural assumptions that empower sample-efficient learning in AMDPs, which facilitates algorithm design beyond specific problems such as linear AMDPs and linear mixture AMDPs.
>
> > Could the authors elaborate on the primary technical challenges they encountered while deriving their theoretical results, when adapt the GEC in Zhong et al. (2022b) to the infinite-horizon average-reward setting, with a constrained algorithm, which seems to be a generalization of the framework established by Jin et al. (2021)?
>
> - Please see our general response for details.
>
> > Can the authors elaborate the optimality of the regret in Theorem 3, when restricted to each specific instances? i.e. linear mixture AMDP, linear AMDP, etc.
>
> - Please see our general response for details.
>
> ---
> [A1] Han Zhong, Wei Xiong, Sirui Zheng, Liwei Wang, Zhaoran Wang, Zhuoran Yang, and Tong Zhang. Gec: A unified framework for interactive decision making in mdp, pomdp, and beyond. arXiv preprint arXiv:2211.01962, 2022a.

---

### Author Response · Authors · 2023-11-17
**General Response (Part 1: Technical Novelties)**

There seems to be some confusions among the reviewers regarding our technical novelties. We would like to highlight our technical novelties in the following, and please refer to Appendix A for a detailed discussion (highlighted in blue).
- **Novel regret decomposition for AMDPs:** One notable distinction is a different regret notion in average-reward RL, and the widely adopted Q- or V- function does not naturally exist due to the infinite horizon without discount. To address this problem, we consider a broad subclass where a modified version of the Bellman optimality equation holds. Furthermore, such a difference is coupled with the challenge of exploration in the context of general function approximation. To effectively bound the (average-reward) regret, we propose a novel regret decomposition technique within the context of general function approximation, which is new in the literature on average-reward RL.
- **Structural constraint on transferability:**  We state that the common structural assumption in episodic MDPs is not sufficient for sample-efficient learning in AMDPs. The proposed structural assumptions in GEC by Zhong et al. (2022b) or BE dimension by Jin et al. (2021) can provide an efficient measurement and control over Bellman error with nuanced adpation, but not for the realization error. Our additional transferability constraint is **tailored to the infinite-horizon setting** and plays a crucial role in analyzing the switching cost, which enables the control over realization error by ensuring a low policy switching from the algorithmic level. Specifically, the structural assumption facilitates a lazy updating algorithm, which adaptively divides the total of $T$ steps into $\mathcal{O}(\log T)$ epochs. We believe that such **an adaptive lazy updating design and corresponding analysis are pivotal in achieving the optimal $\tilde{\mathcal{O}}(\sqrt{T})$ rate**, as opposed to $\tilde{\mathcal{O}}(T^{3/4})$ regret achieved by reduction-based approaches such as OLSVI.FH (Wei et al., 2021) even for specific linear AMDPs. Despite this additional restriction, AGEC can still serve as a unifying complexity measure in the infinite-horizon average-reward setting, like the role of GEC in the finite-horizon setting.
- **Compared with Wei et al. (2021), Chen et al. (2021), Wu et al. (2022):** To the best of our knowledge, existing works on function approximation for AMDPs mainly includes Wei et al. (2021) and Wu et al. (2022),  with a primary focus on problems structured under linearity. However, one key design of these algorithms, the lazy update condition, heavily relies on the postulated linear structure, specifically demanding ${\rm det}(\Lambda_T)\leq(\lambda+2T)^d$ to establish update condition ${\rm log\ det}(\Lambda_t)\geq2\cdot{\rm log\ det}(\Lambda_{\tau_t})$. Notably, the singular exception to this pattern is presented by Chen et al. (2021), who extended the Eluder dimension from finite settings to the infinite case for model-based problems by utilizing importance scores as the update indicator. However, these conditions are not universally applicable to general function classes, encompassing both model-based and value-based incarnations. In response to this limitation, we introduce a **discrepancy-based updating condition** tailored for general function classes under average-reward setting, ensuring a switching cost of $\mathcal{O}(\log T)$ under the PAC framework.

---

> ### Author Response · Authors · 2023-11-17
> **General Response (Part 2: Regret Bounds for Concrete Examples and Highlight the Significance of Our Results)**
>
> Besides, we provide a discussion about the performance of our algorithm on specific instances in the following, and please refer to Appendix F.5 for detailed calculations.
>
> - **Performance on concrete examples:**  Our algorithm achieves a   $\tilde{\mathcal{O}}\Big({\rm sp}(V^*)^\frac{3}{2}d^\frac{3}{2}\sqrt{T}\Big)$ regret for both linear AMDPs and linear mixture AMDPs, and attains $\tilde{\mathcal{O}}({\rm sp}(V^*)d_{\rm E}\sqrt{T\beta})$ for model-based problems. In comparasion, FOPO (Wei et al., 2021) achieves $\tilde{\mathcal{O}}\Big({\rm sp}(V^*)d^\frac{3}{2}\sqrt{T}\Big)$ for linear AMDPs, UCRL2-VTR (Wu et al., 2022) achieves $\mathcal{\tilde{O}}(d\sqrt{DT})$ for linear mixture AMDPs, and SIM-TO-REAL (Chen et al., 2022a) attains $\tilde{\mathcal{O}}(Dd_{\rm E}\sqrt{T\beta})$ for model-based problems.  In Wu et al., (2022), a lower bound of the order $\mathcal{{O}}(d\sqrt{DT})$ is proved for linear mixture AMDPs, where the diameter $D$ is closely related but incomparable with the span ${\rm sp}(V^*)$ considered in our paper. Based on the argument, we posit that our algorithm is nearly optimal concerning the episode length $T$.
> - **Suboptimality:**  While conceding that our algorithm introduces additional multiplying factors when compared to the minimax rate and existing algorithms, we assert that this is an unavoidable trade-off inherent in developing an algorithm for general function classes. The augmented regret is a consequence of a more rigorous design of the optimistic parameter tailored for specific problems. For instance, in linear AMDPs,  the learner can structure the optimistic bonus term for each step as $\| b_t \|_{\Lambda_t}\leq\beta$ based on the empirical covariance matrix, which is inattainable for general function classes. We remark that such suboptimality is **fairly common** in general function approximation literature (Jin et al., 2021, Zhong et al., 2022).
> - The main focus of our paper is to address the problem of minimal structural assumptions that empower sample-efficient learning in AMDPs, rather than striving to enhance regret results.  Our algorithm is in some sense information theoretical -- our work identifies a statistically tractable subclass of AMDPs, and the algorithm **serves as a certificate for the regret upper bound**.
>
>
> ---
>
> Finally, we would like to emphasize the **significance** of our work. Our proposed complexity measure, AGEC, not only unifies all known tractable AMDPs but all includes some new AMDPs identified by us. Furthermore, our generic algoirthm, applicable to all AMDPs with low AGEC, positions our work as the first comprehensive framework for AMDPs. Additionally, our study establishes a connection between sample-efficient learning for AMDPs and lazy-updating algorithms, supported by a general theoretical analysis pipeline. We firmly believe that these contributions are paramount in advancing the understanding of AMDPs, a category of MDP that has, to some extent, been overlooked when compared to finite-horizon MDPs.
>
> ---
> [A1] Chi Jin, Qinghua Liu, and Sobhan Miryoosefi. Bellman eluder dimension: New rich classes of rl problems, and sample-efficient algorithms. Advances in neural information processing systems,34:13406–13418, 2021.
>
> [A2] Han Zhong, Wei Xiong, Sirui Zheng, Liwei Wang, Zhaoran Wang, Zhuoran Yang, and Tong Zhang. Gec: A unified framework for interactive decision making in mdp, pomdp, and beyond. arXiv preprint arXiv:2211.01962, 2022a.
>
> [A3] Chen-Yu Wei, Mehdi Jafarnia Jahromi, Haipeng Luo, and Rahul Jain. Learning infinite-horizon average-reward mdps with linear function approximation. In International Conference on Artificial Intelligence and Statistics, pp. 3007–3015. PMLR, 2021.
>
> [A4] Yue Wu, Dongruo Zhou, and Quanquan Gu. Nearly minimax optimal regret for learning infinite horizonaverage-reward mdps with linear function approximation. In International Conference on Artificial Intelligence and Statistics, pp. 3883–3913. PMLR, 2022.
>
> [A5] Xiaoyu Chen, Jiachen Hu, Chi Jin, Lihong Li, and Liwei Wang. Understanding domain randomization for sim-to-real transfer. International Conference on Learning Representations, 2022a.

---

### Author Response · Authors · 2023-11-23
**Official Comment by Authors**

Dear Reviewers,

We hope this message finds you well. As the rebuttal period will close soon, we would like to reach out to you and inquire if you have any additional concerns regarding our paper. We would be immensely grateful if you could kindly review our responses to your comments. This would allow us to address any further questions or concerns you may have before the rebuttal period concludes.

Best,

Authors

---

### Meta-Review · Area_Chair_W9Uf · 2023-12-06

**Metareview:**

This paper studies reinforcement learning (RL) with infinite-horizon average-reward Markov decision processes and proposes sample-efficient learning algorithms with general function approximation. The results draw from various established works on RL with general function approximation. While the reviewers have a little concerns about novelty and contribution, the paper is deemed a valuable addition to the existing literature. Thus I recommend acceptance.

**Justification For Why Not Higher Score:**

The reviewers have a little concerns about novelty and contribution.

**Justification For Why Not Lower Score:**

The overall evaluation of this paper is still very positive.

---

### Decision · Program_Chairs · 2024-01-16

Accept (poster)